# Generalization error in high-dimensional perceptrons: Approaching Bayes error with convex optimization

**Benjamin Aubin**
Université Paris-Saclay
CNRS, CEA
Institut de physique théorique
91191, Gif-sur-Yvette, France.

**Florent Krzakala**
IdePHICS laboratory
École Polytechnique Fédérale de Lausanne
1015, Lausanne, Switzerland

**Yue M. Lu**
John A. Paulson
School of Engineering and Applied Sciences
Harvard University
Cambridge, MA 02138, USA

**Lenka Zdeborová**
SPOC laboratory
École Polytechnique Fédérale de Lausanne
1015, Lausanne, Switzerland

## Abstract

We consider a commonly studied supervised classification of a synthetic dataset whose labels are generated by feeding a one-layer neural network with random i.i.d inputs. We study the generalization performances of standard classifiers in the high-dimensional regime where $\alpha = n/d$ is kept finite in the limit of a high dimension $d$ and number of samples $n$. Our contribution is three-fold: First, we prove a formula for the generalization error achieved by $\ell_2$ regularized classifiers that minimize a convex loss. This formula was first obtained by the heuristic replica method of statistical physics. Secondly, focussing on commonly used loss functions and optimizing the $\ell_2$ regularization strength, we observe that while ridge regression performance is poor, logistic and hinge regression are surprisingly able to approach the Bayes-optimal generalization error extremely closely. As $\alpha \to \infty$ they lead to Bayes-optimal rates, a fact that does not follow from predictions of margin-based generalization error bounds. Third, we design an optimal loss and regularizer that provably leads to Bayes-optimal generalization error.

## 1 Introduction

High-dimensional statistics, where the ratio $\alpha = n/d$ is kept finite while the dimensionality $d$ and the number of samples $n$ grow, often display interesting non-intuitive features. Asymptotic generalization performances for such problems in the so-called *teacher-student* setting, with synthetic data, have been the subject of intense investigations spanning many decades [1–6]. To understand the effectiveness of modern machine learning techniques, and also the limitations of the classical statistical learning approaches [7, 8], it is of interest to revisit this line of research. Indeed, this direction is currently the subject to a renewal of interests, as testified by some very recent, yet already rather influential papers [9–13]. The present paper subscribes to this line of work and studies high-dimensional classification within one of the simplest models considered in statistics and machine learning: convex linear estimation with data generated by a teacher *perceptron* [14]. We will focus on the generalization abilities in this problem, and compare the performances of Bayes-optimal estimation to the more standard *Empirical Risk Minimization* (ERM). We then compare the results with the prediction of standard generalization bounds that illustrate in particular their limitation even in this simple, yet non-trivial, setting.

**Synthetic data model —** We consider a supervised machine learning task, whose dataset is generated by a single layer neural network, often named a *teacher* [1–3], that belongs to the Generalized Linear Model (GLM) class. Therefore, we assume the $n$ samples are generated according to

$$\mathbf{y} = \varphi_{\mathrm{out}^\star} \left( \frac{1}{\sqrt{d}} \mathbf{X} \mathbf{w}^\star \right) \qquad \Leftrightarrow \qquad \mathbf{y} \sim P_{\mathrm{out}^\star} \left( . | \frac{1}{\sqrt{d}} \mathbf{X} \mathbf{w}^\star \right) , \qquad (1)$$

where $\mathbf{w}^\star \in \mathbb{R}^d$ denotes the ground truth vector drawn from a probability distribution $P_{\mathrm{w}^\star}$ with second moment $\rho_{d,\mathrm{w}^\star} \equiv \frac{1}{d} \mathbb{E} \left[ \|\mathbf{w}^\star\|_2^2 \right]$ and $\varphi_{\mathrm{out}^\star} : \mathbb{R} \mapsto \mathbb{R}$ represents a component-wise deterministic or stochastic activation function equivalently associated to a distribution $P_{\mathrm{out}^\star}$. The input data matrix $\mathbf{X} = (\mathbf{x}_\mu)_{\mu=1}^n \in \mathbb{R}^{n \times d}$ contains i.i.d Gaussian vectors, i.e $\forall \mu \in [1:n]$, $\mathbf{x}_\mu \sim \mathcal{N} (\mathbf{0}, \mathrm{I}_d)$. Even though the framework we use and the theorems and results we derive are valid for a rather generic channel in eq. (1) — including regression problems — we will mainly focus the presentation on the commonly considered perceptron case: a binary classification task with data given by a sign activation function $\varphi_{\mathrm{out}^\star} (\mathbf{z}) = \mathrm{sign} (\mathbf{z})$, with a Gaussian weight distribution $P_{\mathrm{w}^\star}(\mathbf{w}^\star) = \mathcal{N}_{\mathrm{w}^\star} (\mathbf{0}, \rho_{d,\mathrm{w}^\star} \mathrm{I}_d)$. The $\pm 1$ labels are thus generated as

$$\mathbf{y} = \mathrm{sign} \left( \frac{1}{\sqrt{d}} \mathbf{X} \mathbf{w}^\star \right) , \quad \text{with} \quad \mathbf{w}^\star \sim \mathcal{N}_{\mathrm{w}^\star} (\mathbf{0}, \rho_{d,\mathrm{w}^\star} \mathrm{I}_d) . \qquad (2)$$

This particular setting was extensively studied in the past [1, 15] and is interesting in the sense it does not show a computational-to-statistical gap. Yet, our analysis and the set of equations presented in SM. III.15 are valid more generically to any other ground truth distributions $P_{\mathrm{out}^\star}$ and $P_{\mathrm{w}^\star}$. Finally, the isotropic Gaussian hypothesis of the input vectors X can be relaxed to non-isotropic Gaussian.

**Empirical Risk Minimization —** The workhorse of machine learning is Empirical Risk Minimization (ERM), where one minimizes a *loss function* in the corresponding high-dimensional parameter space $\mathbb{R}^d$. To avoid overfitting of the training set one often adds a *regularization term* $r$. ERM then corresponds to estimating $\hat{\mathbf{w}}_{\mathrm{erm}} = \mathrm{argmin}_{\mathbf{w}} [\mathcal{L} (\mathbf{w}; \mathbf{y}, \mathrm{X})]$ where the regularized training loss $\mathcal{L}$ is defined by, using the notation $z_\mu (\mathbf{w}, \mathbf{x}_\mu) \equiv \frac{1}{\sqrt{d}} \mathbf{x}_\mu^\mathsf{T} \mathbf{w}$,

$$\mathcal{L} (\mathbf{w}; \mathbf{y}, \mathrm{X}) = \sum_{\mu=1}^n l \left( y_\mu, z_\mu (\mathbf{w}, \mathbf{x}_\mu) \right) + r (\mathbf{w}) . \qquad (3)$$

The goal of the present paper is to discuss the generalization performance of these estimators for the classification task (2) in the high-dimensional limit. We focus our analysis on commonly used loss functions $l$, namely the square $l^{\mathrm{square}}(y, z) = \frac{1}{2}(y - z)^2$, logistic $l^{\mathrm{logistic}}(y, z) = \log(1 + \exp(-yz))$ and hinge losses $l^{\mathrm{hinge}}(y, z) = \max(0, 1 - yz)$. We will mainly illustrate our results for the $\ell_2$ regularization $r(\mathbf{w}) = \lambda \|\mathbf{w}\|_2^2 / 2$, where we introduced a regularization strength hyper-parameter $\lambda$, even though a similar rigorous analysis can be conducted for any separable and convex regularizer.

**Related works —** The above learning problem has been extensively studied in the statistical physics community using the heuristic replica method [1–3, 14, 15]. Due to the interest in high-dimensional statistics, they have experienced a resurgence in popularity in recent years. In particular, rigorous works on related problems are much more recent. The authors of [10] established rigorously the replica-theory predictions for the Bayes-optimal generalization error. Here we focus on standard ERM estimation and compare it to the information theoretic baseline results obtained in [10]. Authors of [16] analyzed rigorously M-estimators for the regression case where data are generated by a linear-activation teacher. Here we analyze classification with a more general and non-linear teacher, focusing in particular on the sign-teacher. The case of max-margin loss was studied in [17] with a technically closely related proof, but with a focus on the over-parametrized regime, thus not addressing the questions that we focus on. A range of unregularized losses was also analyzed for a sigmoid teacher (that is very similar to a sign-teacher) again in the context of the double-descent behavior in [18, 19]. Here we focus instead on the regularized case as it drastically improves generalization performances of the ERM and that allows us to compare with the Bayes-optimal estimation as well as to standard generalization bounds. Our proof, as in the above mentioned works and [20], is based on Gordon's Gaussian Min-max inequalities [16, 21], including in particular the effect of the regularization.

**Main contributions —** Our first main contribution is to provide, in Sec. 2, the rigorous high-dimensional asymptotics of the classification generalization performances of ERM with the loss

given by (3), for any convex loss $l$ and a $\ell_2$ regularization. Note that for the sake of conciseness, we focus on this latter case, but the proof is performed in the more general regression case and can be easily extended to any convex separable regularization and to non-isotropic Gaussian inputs. Additionally, we provide a proof of the equivalence between the results of our paper and the ones initially obtained by the replica method, which is of additional interest given the wide range of application of these heuristics statistical-physics techniques in machine learning and computer science [22, 23]. In particular, the replica predictions in [15, 24–26] follow from our results. Another approach that originated in physics are the so-called TAP equations [27–29] that lead to the so-called Approximate Message Passing algorithm for solving linear and generalized linear problems with Gaussian matrices [30, 31]. This algorithm can be analyzed with the so-called *state evolution* method [32], and it is widely believed (and in fact proven for linear problems [4, 33]) that the fixed-point of the state evolution gives the optimal error in high-dimensional convex optimization problems. The state evolution equations are in fact equivalent to the one given by the replica theory and therefore our results vindicate this approach as well. We also demonstrate numerically that these asymptotic results are very accurate even for moderate system sizes, and they have been performed with the `scikit-learn` library [34].

Secondly, and more importantly, we provide in Sec. 3 a detailed analysis of the generalization error for standard losses such as square, hinge (or equivalently support vector machine) and logistic, as a function of the regularization strength $\lambda$ and the number of samples per dimension $\alpha$. We observe, in particular, that while the ridge regression never closely approaches the Bayes-optimal performance, the logistic regression with optimized $\ell_2$ regularization gets extremely close to optimal. And so does, to a lesser extent, the hinge regression and the max-margin estimator to which the unregularized logistic and hinge converge [35]. It is quite remarkable that these canonical losses are able to approach the error of the Bayes-optimal estimator for which, in principle, the marginals of a high-dimensional probability distribution need to be evaluated. Notably, all the later losses give —for a *good choice* of the regularization strength $\lambda$— generalization errors scaling as $\Theta\left(\alpha^{-1}\right)$ for large $\alpha$, just as the Bayes-optimal generalization error [10]. This is found to be at variance with the prediction of Rademacher and max-margin-based bounds that predict instead a $\Theta\left(\alpha^{-1/2}\right)$ rate [36, 37], which therefore appear to be vacuous in the high-dimensional regime. Notice that we reproduce the Rademacher complexity results of [38], which deal exactly with the same setting, only to bring to light interesting conclusions on the ERM estimation.

Third, in Sec. 4, we design a custom (non-convex) loss and regularizer from the knowledge of the ground truth distributions $P_{\text{out}^\star}, P_{\text{w}^\star}$ that provably gives a plug-in estimator that efficiently achieves Bayes-optimal performances, including the optimal $\Theta\left(\alpha^{-1}\right)$ rate for the generalization error. Our construction is related to the one discussed in [39–41], but is not restricted to convex losses.

## 2 Main technical results

In the formulas that arise for this statistical estimation problem, the correlations between the estimator $\hat{\mathbf{w}}$ and the ground truth vector $\mathbf{w}^\star$ play a fundamental role and we thus define two scalar overlap parameters to measure the statistical reconstruction:

$$m \equiv \frac{1}{d}\,\mathbb{E}_{\mathbf{y},\mathbf{X}}\,\left[\hat{\mathbf{w}}^\mathsf{T}\mathbf{w}^\star\right]\,, \qquad\qquad q \equiv \frac{1}{d}\,\mathbb{E}_{\mathbf{y},\mathbf{X}}\,\left[\|\hat{\mathbf{w}}\|_2\right]^2\,. \qquad (4)$$

In particular, the generalization error of the estimator $\hat{\mathbf{w}}(\alpha) \in \mathbb{R}^d$ obtained by performing Empirical Risk Minimization (ERM) on the training loss $\mathcal{L}$ in eq. (3) with $n = \alpha d$ samples

$$e_{\text{g}}^{\text{erm}}(\alpha) \equiv \mathbb{E}_{y,\mathbf{x}}\mathbb{1}\left[y \neq \hat{y}\left(\hat{\mathbf{w}}(\alpha);\mathbf{x}\right)\right]\,, \qquad (5)$$

where $\hat{y}\left(\hat{\mathbf{w}}(\alpha);\mathbf{x}\right)$ denotes the predicted label, has both at finite $d$ and in the asymptotic limit an explicit expression depending only on the above overlaps $m$ and $q$:

**Proposition 2.1** (Generalization error of classification)**.** *In our synthetic binary classification task, the generalization error of ERM (or equivalently the test error) is given by*

$$e_{\text{g}}^{\text{erm}}\left(\alpha\right) = \frac{1}{\pi}acos\left(\sqrt{\eta}\right)\,, \qquad \textit{with } \eta \equiv \frac{m^2}{\rho_{d,\text{w}^\star}\,q} \qquad \textit{and} \qquad \rho_{d,\text{w}^\star} \equiv \frac{1}{d}\,\mathbb{E}\left[\|\mathbf{w}^\star\|_2^2\right]. \qquad (6)$$

*Proof.* The proof, shown in SM. II, is a simple computation based on Gaussian integration. $\qquad\square$

To obtain the generalization performances, it thus remains to obtain the asymptotic values of $m$, $q$ (and thus of $\eta$), in the limit $d \to \infty$. With the $\ell_2$ regularization, these values are characterized by a set of fixed point equations given by the next theorems. For any $\tau > 0$, let us first recall the definitions of the Moreau-Yosida regularization and the proximal operator of a convex loss function $(y, z) \mapsto \ell(y \cdot z)$:

$$\mathcal{M}_\tau(z) = \min_x \left\{ \ell(x) + \frac{(x - z)^2}{2\tau} \right\}, \qquad \mathcal{P}_\tau(z) = \mathrm{argmin}_x \left\{ \ell(x) + \frac{(x - z)^2}{2\tau} \right\}. \qquad (7)$$

**Theorem 2.2** (Gordon's min-max fixed point - Binary classification with $\ell_2$ regularization). *As $n, d \to \infty$ with $n/d = \alpha = \Theta(1)$, the overlap parameters $m, q$ and the prior's second moment $\rho_{d,w^\star}$ concentrate to*

$$m \underset{d \to \infty}{\longrightarrow} \sqrt{\rho_{w^\star}} \mu^*, \qquad q \underset{d \to \infty}{\longrightarrow} (\mu^*)^2 + (\delta^*)^2, \qquad \rho_{d,w^\star} \underset{d \to \infty}{\longrightarrow} \rho_{w^\star}, \qquad (8)$$

*where parameters $\mu^*$ and $\delta^*$ are solutions of*

$$(\mu^*, \delta^*) = \arg\min_{\mu, \delta \geq 0} \sup_{\tau > 0} \left\{ \frac{\lambda(\mu^2 + \delta^2)}{2} - \frac{\delta^2}{2\tau} + \alpha \mathbb{E}_{g,s} \mathcal{M}_\tau[\delta g + \mu s \varphi_{\mathrm{out}^\star}(\sqrt{\rho_{w^\star}} s)] \right\}, \qquad (9)$$

*and $g, s$ are two i.i.d standard normal random variables. The solutions $(\mu^*, \delta^*)$ of (9) can be reformulated as a set of fixed point equations*

$$\mu^* = \frac{\alpha}{\lambda\tau^* + \alpha} \mathbb{E}_{g,s}[s \cdot \varphi_{\mathrm{out}^\star}(\sqrt{\rho_{w^\star}} s) \cdot \mathcal{P}_{\tau^*}(\delta^* g + \mu^* s \varphi_{\mathrm{out}^\star}(\sqrt{\rho_{w^\star}} s))],$$

$$\delta^* = \frac{\alpha}{\lambda\tau^* + \alpha - 1} \mathbb{E}_{g,s}[g \cdot \mathcal{P}_{\tau^*}(\delta^* g + \mu^* s \varphi_{\mathrm{out}^\star}(\sqrt{\rho_{w^\star}} s))], \qquad (10)$$

$$(\delta^*)^2 = \alpha \mathbb{E}_{g,s}[((\delta^* g + \mu^* s \varphi_{\mathrm{out}^\star}(\sqrt{\rho_{w^\star}} s)) - \mathcal{P}_{\tau^*}(\delta^* g + \mu^* s \varphi_{\mathrm{out}^\star}(\sqrt{\rho_{w^\star}} s)))^2].$$

The proof, shown in SM. III.1 for regression, and consequently valid for the classification particular case, is an application of the Gordon's comparison inequalities. Let us mention that the theorem focuses on the case of classification for i.i.d isotropic Gaussian input data and $\ell_2$ regularization as this case was extensively studied in past works. However, similar techniques can be generalized to handle convex and separable regularization functions (see, *e.g.*, [42]), and Gaussian inputs with more general covariance matrices [43].

This set of fixed point equations can be finally mapped to the ones obtained equivalently by the heuristic *replica* method from statistical physics (whose heuristic derivation is shown in SM. IV) as well as the state evolution of the Approximate Message-Passing (AMP) algorithm [28, 31, 44]. Notice that the main reason why we rely on Convex Gaussian Min-lax Theory (CGMT) to make this set of equation rigorous is that we do not know how to prove that the AMP state evolution corresponds to the solution of ERM. Only after having the CGMT proof in hand, it follows that the SE of AMP gives the same equations than the CGMT. As a result, their validity for this convex estimation problem is rigorously established by the following theorem:

**Corollary 2.3** (Equivalence Gordon-replicas). *As $n, d \to \infty$ with $n/d = \alpha = \Theta(1)$, the overlap parameters $m, q$ concentrate to the fixed point of the following set of equations:*

$$m = \alpha \Sigma \rho_{w^\star} \cdot \mathbb{E}_{y,\xi} \left[ \mathcal{Z}_{\mathrm{out}^\star} \times f_{\mathrm{out}^\star}(y, \sqrt{\rho_{w^\star}\eta} \xi, \rho_{w^\star}(1 - \eta)) \cdot f_{\mathrm{out}}\left( y, q^{1/2}\xi, \Sigma \right) \right],$$

$$q = m^2/\rho_{w^\star} + \alpha \Sigma^2 \cdot \mathbb{E}_{y,\xi} \left[ \mathcal{Z}_{\mathrm{out}^\star}(y, \sqrt{\rho_{w^\star}\eta} \xi, \rho_{w^\star}(1 - \eta)) \cdot f_{\mathrm{out}}\left( y, q^{1/2}\xi, \Sigma \right)^2 \right], \qquad (11)$$

$$\Sigma = \left( \lambda - \alpha \cdot \mathbb{E}_{y,\xi} \left[ \mathcal{Z}_{\mathrm{out}^\star}(y, \sqrt{\rho_{w^\star}\eta} \xi, \rho_{w^\star}(1 - \eta)) \cdot \partial_\omega f_{\mathrm{out}}\left( y, q^{1/2}\xi, \Sigma \right) \right] \right)^{-1},$$

$$\text{with} \quad \eta \equiv \frac{m^2}{\rho_{w^\star} q}, \qquad\qquad f_{\mathrm{out}}(y, \omega, V) \equiv V^{-1}(\mathcal{P}_V[l(y, .)](\omega) - \omega),$$

$$\mathcal{Z}_{\mathrm{out}^\star}(y, \omega, V) = \mathbb{E}_z \left[ P_{\mathrm{out}^\star}\left( y | \sqrt{V} z + \omega \right) \right], \quad f_{\mathrm{out}^\star}(y, \omega, V) \equiv \partial_\omega \log(\mathcal{Z}_{\mathrm{out}^\star}(y, \omega, V)), \tag{12}$$

*where $\xi, z$ denote two i.i.d standard Gaussian normal random variables, and $\mathbb{E}_y$ the continuous or discrete sum over all possible values $y$ according to $P_{\mathrm{out}^\star}$.*

For clarity, the proof is again left in SM. III.3. Notice that the equivalent sets of equations (10)-(11) have been made rigorous only for binary classification and regression in SM. III.1 with $\ell_2$ regularization. However, the replica's prediction of the fixed point equations for the whole GLM class (classification and regression) are provided for generic convex and separable loss and regularizer (different than $\ell_2$) in SM. III.2 and contain instead six equations. These heuristic equations are nonetheless believed to hold true and the generic Gordon's min-max framework can be easily generalized to this case.

**Bayes optimal baseline —** Finally, we shall compare the ERM performances to the Bayes-optimal generalization error. Being the information-theoretically best possible estimator, we will use it as a reference baseline for comparison. The expression of the Bayes-optimal generalization was derived in [25] and proven in [10] and we recall here the result:

**Theorem 2.4** (Bayes Asymptotic performance, from [10]). *For the model* (1) *with* $P_{w^\star}(w^\star) = \mathcal{N}_{w^\star}\left(\mathbf{0}, \rho_{d,w^\star}\mathrm{I}_d\right)$ *with* $\rho_{d,w^\star} \xrightarrow[d\to\infty]{} \rho_{w^\star}$, *the Bayes-optimal generalization error is quantified by two scalar parameters* $q_{\mathrm{b}}$ *and* $\hat{q}_{\mathrm{b}}$ *that verify asymptotically the set of fixed point equations*

$$q_{\mathrm{b}} = \frac{\hat{q}_{\mathrm{b}}}{1+\hat{q}_{\mathrm{b}}}, \quad \hat{q}_{\mathrm{b}} = \alpha\mathbb{E}_{y,\xi}\left[\mathcal{Z}_{\mathrm{out}^\star}\left(y, q_{\mathrm{b}}^{1/2}\xi, \rho_{w^\star} - q_{\mathrm{b}}\right) \cdot f_{\mathrm{out}^\star}\left(y, q_{\mathrm{b}}^{1/2}\xi, \rho_{w^\star} - q_{\mathrm{b}}\right)^2\right], \quad (13)$$

*and reads*

$$e_{\mathrm{g}}^{\mathrm{bayes}}(\alpha) = \frac{1}{\pi}acos\left(\sqrt{\eta_{\mathrm{b}}}\right) \qquad with \qquad \eta_{\mathrm{b}} = \frac{q_{\mathrm{b}}}{\rho_{w^\star}}. \quad (14)$$

## 3 Generalization errors

We now move to the core of the paper and analyze the set of fixed point equations (10), or equivalently (11), leading to the generalization performances given by (6), for common classifiers on our synthetic binary classification task. As already stressed, even though the results are valid for a wide range of separable convex loss and regularizers, we focus on estimators based on ERM with $\ell_2$ regularization $r(\mathbf{w}) = \lambda\|\mathbf{w}\|_2^2/2$, and with square loss (ridge regression) $l^{\mathrm{square}}(y,z) = \frac{1}{2}(y-z)^2$, logistic loss (logistic regression) $l^{\mathrm{logistic}}(y,z) = \log(1 + \exp(-yz))$ or hinge loss (SVM) $l^{\mathrm{hinge}}(y,z) = \max(0, 1-yz)$. In particular, we study the influence of the hyper-parameter $\lambda$ on the generalization performances and the different large $\alpha$ behavior generalization rates in the high-dimensional regime, and compare with the Bayes results. We show the solutions of the set of fixed point equations eqs. (11) in Figs. 1a, 1b, 1c respectively for ridge, hinge and logistic $\ell_2$ regressions. Let us mention that the analytic solution of the set of equations (10) is provided only in the ridge case, for which its quadratic loss allows to derive and fully solve the equations (see SM. V.3), and also for logistic and hinge losses in the regime of vanishing $\lambda \to 0$ (see SM. V.4). Unfortunately, in general the set of equations has no analytical closed form expression and needs therefore to be evaluated numerically. It is in particular the case for logistic and hinge for finite $\lambda$, whose Moreau-Yosida regularization eq. (7) is, yet, analytical. However, note that (10) are fixed point equations on scalar variables so that their numerical resolution posed no problem. The non-trivial part of the rigorous analysis, performed in Thm. 2.2, is the reduction of the high-dimensional problem to the scalar fixed point equations.

First, to highlight the accuracy of the theoretical predictions, we compare in Figs. 1a-1c the ERM asymptotic ($d \to \infty$) generalization error with the performances of numerical simulations ($d = 10^3$, averaged over $n_s = 20$ samples) of ERM of the training loss eq. (3). Presented for a wide range of number of samples $\alpha$ and of regularization strength $\lambda$, we observe a perfect match between theoretical predictions and numerical simulations so that the error bars are barely visible and have been therefore removed. This shows that the asymptotic predictions are valid even with very moderate sizes. As an information theoretical baseline, we also show the Bayes-optimal performances (black) given by the solution of eq. (13).

**Ridge estimation—** As we might expect the square loss gives the worst performances. For low values of the generalization, it leads to an interpolation-peak at $\alpha = 1$. The limit of vanishing regularization $\lambda \to 0$ leads to the *least-norm* or *pseudo-inverse* estimator $\hat{\mathbf{w}}_{\mathrm{pseudo}} = (\mathrm{X}^\intercal\mathrm{X})^{-1}\mathrm{X}^\intercal\mathbf{y}$. The corresponding generalization error presents the largest interpolation-peak and achieves a maximal generalization error $e_{\mathrm{g}} = 0.5$. These are well known observations, discussed as early as in [24,26,45], that are object of a renewal of interest under the name *double descent*, following a recent series of papers [11,46–52]. This double descent behavior for the pseudo-inverse is shown in Fig. 1a with

a yellow line. On the contrary, larger regularization strengths do not suffer this peak at $\alpha = 1$, but their generalization error performance is significantly worse than the Bayes-optimal baseline for larger values of $\alpha$. Indeed, as we might expect, for a large number of samples, a large regularization biases wrongly the training. However, even with optimized regularizations, performances of the ridge estimator remains far away from the Bayes-optimal performance.

**Hinge and logistic estimation—** Both these losses, which are the classical ones used in classification problems, improve drastically the generalization error. First of all, let us notice that they do not display a double-descent behavior. This is due to the fact that our results are illustrated in the noiseless case and that our synthetic dataset is always linearly separable. Optimizing the regularization, our results in Fig. 1b-1c show both hinge and logistic ERM-based classification approach very closely the Bayes error. This might be an interesting message for practitioners, even though showing it in more realistic settings would be preferable. To offset these results, note that performances of logistic regression on non-linearly separable data are however very poor, as illustrated by our analysis of a *rectangle door* teacher (see SM. V.6).

**Max-margin estimation—** As discussed in [35], both the logistic and hinge estimator converge, for vanishing regularization $\lambda \to 0$, to the *max-margin* solution. Taking the $\lambda \to 0$ limit in our equations, we thus obtain the *max-margin* estimator performances. While this is not what gives the best generalization error (as can be seen in Fig.1c the logistic with an optimized $\lambda$ has a lower error), the max-margin estimator gives very good results, and gets very close to the Bayes-error.

**Optimal regularization—** Defining the regularization value that optimizes the generalization as

$$\lambda^{\mathrm{opt}}(\alpha) = \mathrm{argmin}_\lambda e_{\mathrm{g}}^{\mathrm{erm}}(\alpha, \lambda) \ . \tag{15}$$

we show in Figs. 1b-1c that both optimal values $\lambda^{\mathrm{opt}}(\alpha)$ (dashed-dotted orange) for logistic and hinge regression decrease to 0 as $\alpha$ grows and more data are given. Somehow surprisingly, we observe in particular that the generalization performances of logistic regression with optimal regularization are *extremely close* to the Bayes performances. The difference with the optimized logistic generalization error is barely visible by eye, so that we explicitly plotted the difference, which is roughly of order $10^{-5}$.

Ridge regression Fig. 1a shows a singular behavior: there exists an optimal value (purple) which is moreover independent of $\alpha$ achieved for $\lambda^{\mathrm{opt}} \simeq 0.5708$. This value was first found numerically and confirmed afterwards semi-analytically in SM. V.3.

**Generalization rates at large $\alpha$—** Finally, we turn to the very instructive behavior at large values of $\alpha$ when a large amount of data is available. First, we notice that the Bayes-optimal generalization error, whose large $\alpha$ analysis is performed in SM. V.1, decreases as $e_{\mathrm{g}}^{\mathrm{bayes}} \underset{\alpha \to \infty}{\sim} 0.4417\alpha^{-1}$. Compared to this optimal value, ridge regression gives poor performances in this regime. For any value of the regularization $\lambda$ — and in particular for both the pseudo-inverse case at $\lambda = 0$ and the optimal estimator $\lambda^{\mathrm{opt}}$ — its generalization performances decrease much slower than the Bayes rate, and goes only as $e_{\mathrm{g}}^{\mathrm{ridge}} \underset{\alpha \to \infty}{\sim} 0.2405\alpha^{-1/2}$ (see SM. V.3 for the derivation). Hinge and logistic regressions present a radically different, and more favorable, behavior. Fig. 1b-1c show that keeping $\lambda$ finite when $\alpha$ goes to $\infty$, does not yield the Bayes-optimal rates. However the max-margin solution (that corresponds to the $\lambda \to 0$ limit of these estimators) gives extremely good performances $e_{\mathrm{g}}^{\mathrm{logistic,hinge}} \underset{\lambda \to 0}{\sim} e_{\mathrm{g}}^{\mathrm{max-margin}} \underset{\alpha \to \infty}{\sim} 0.500\alpha^{-1}$ see derivation in SM. V.4). This is the same rate as the Bayes one, only that the constant is slightly higher. However, we do not know whether there is a general criteria that would distinguish when the decay is $\Theta(\alpha^{-1})$ or $\Theta(\alpha^{-1/2})$. Providing such a generic criteria is definitely a line of research we would like to investigate in the future. Moreover, let us point out the work [53] that discusses fast convergence rates $\Theta(n^{-1})$ for the hinge loss, whose analysis for only very large $\alpha$ and Lipshitz functions does not directly apply to our setting.

**Comparison with VC and Rademacher statistical bounds—** Given the fact that both the max-margin estimator and the optimized logistic achieve optimal generalization rates going as $\Theta\left(\alpha^{-1}\right)$, it is of interest to compare those rates to the prediction of statistical learning theory bounds. Statistical learning analysis (see e.g. [36, 37, 54]) relies to a large extent on the *Vapnik-Chervonenkis* dimension (VC) analysis and on the so-called *Rademacher complexity*. The uniform convergence result states

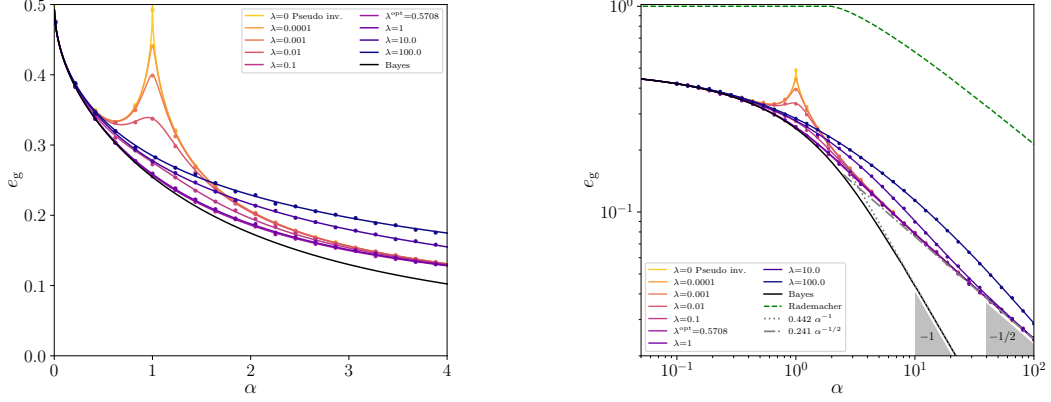

(a) Ridge regression: square loss with $\ell_2$ regularization. Interpolation-peak, at $\alpha = 1$, is maximal for the pseudo-inverse estimator $\lambda = 0$ (yellow line) that reaches $e_{\mathrm{g}} = 0.5$.

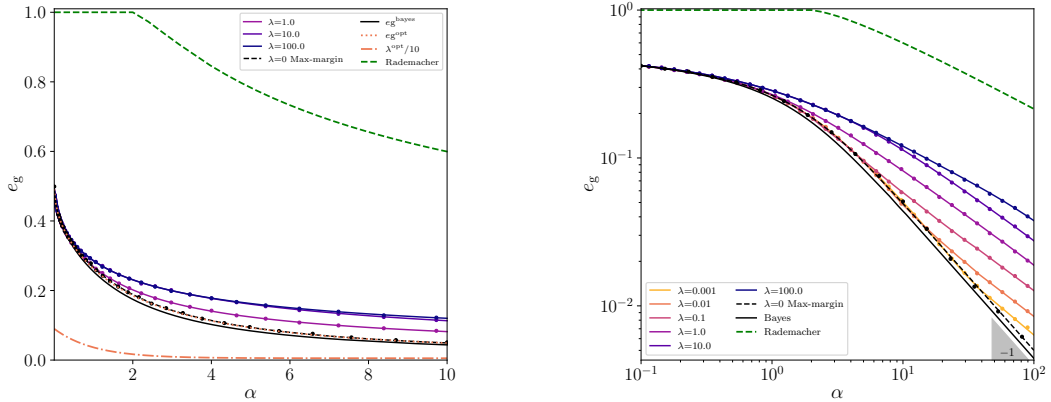

(b) Hinge regression: hinge loss with $\ell_2$ regularization. For clarity the rescaled value of $\lambda^{\mathrm{opt}}/10$ (dotted-dashed orange) is shown as well as its generalization error $e_{\mathrm{g}}^{\mathrm{opt}}$ (dotted orange) that is slightly below and almost indistinguishable of the max-margin performances (dashed black).

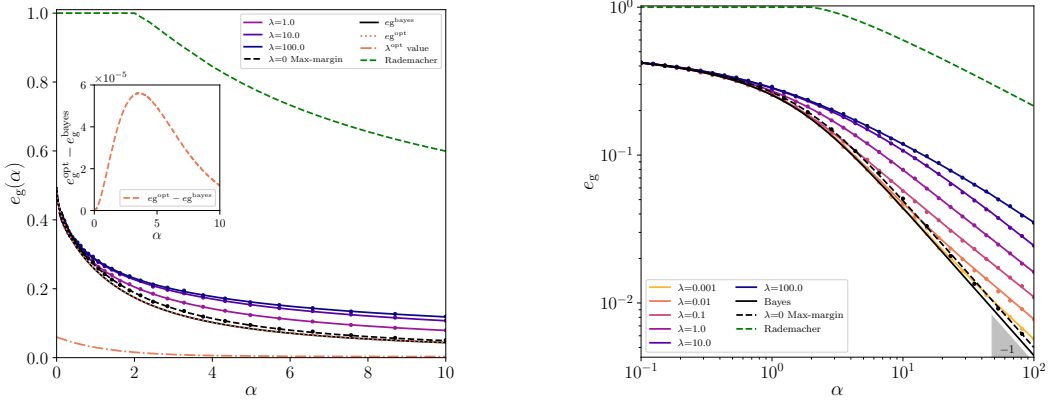

(c) Logistic regression: logistic loss with $\ell_2$ regularization - The value of $\lambda^{\mathrm{opt}}$ (dotted-dashed orange) is shown as well as its generalization error $e_{\mathrm{g}}^{\mathrm{opt}}$ (dotted orange). Visually indistinguishable from the Bayes-optimal line, their difference $e_{\mathrm{g}}^{\mathrm{opt}} - e_{\mathrm{g}}^{\mathrm{bayes}}$ is shown as an inset (dashed orange).

Figure 1: Asymptotic generalization error for $\ell_2$ regularization ($d \to \infty$) as a function of $\alpha$ for different regularizations strengths $\lambda$, compared to numerical simulation (points) of ridge regression for $d = 10^3$ and averaged over $n_s = 20$ samples. Numerics has been performed with the default methods *Ridge*, *LinearSVC*, *LogisticRegression* of `scikit-learn` package [34]. Bayes optimal performances are shown with a black line and goes as $\Theta\left(\alpha^{-1}\right)$, while the Rademacher complexity [38] (dashed green) decreases as $\Theta\left(\alpha^{-1/2}\right)$. Both hinge and logistic converge to max-margin estimator (limit $\lambda = 0$) which is shown in dashed black and deceases as $\Theta(\alpha^{-1})$, while Ridge decreases as $\Theta(\alpha^{-1/2})$.

that if the Rademacher complexity or the Vapnik-Chervonenkis dimension $d_{\mathrm{VC}}$ is finite, then for a large enough number of samples the generalization gap will vanish uniformly over all possible values of parameters. Informally, uniform convergence tells us that with high probability, for any value of the weights $\mathbf{w}$, the generalization gap satisfies $\mathcal{R}_{\mathrm{population}}(\mathbf{w}) - \mathcal{R}_{\mathrm{empirical}}^{n}(\mathbf{w}) = \Theta\left(\sqrt{d_{\mathrm{VC}}/n}\right)$ where $d_{\mathrm{VC}} = d - 1$ for our GLM hypothesis class. Therefore, given that the empirical risk can go to *zero* (since our data are separable), this provides a generalization error upper-bound $e_{\mathrm{g}} \leq \Theta(\alpha^{-1/2})$. This is much worse that what we observe in practice, where we reach the Bayes rate $e_{\mathrm{g}} = \Theta(\alpha^{-1})$. Tighter bounds can be obtained using the Rademacher complexity, and this was studied recently (using the aforementioned *replica method*) in [38] for the very same problem. To bring to light interesting conclusions, we reproduced their results and plotted the Rademacher complexity generalization bound in Fig.1 (dashed-green) that decreases as $\Theta\left(\alpha^{-1/2}\right)$ for the binary classification task eq. (2).

One may wonder if this could be somehow improved. Another statistical-physics heuristic computation, however, suggests that, unfortunately, uniform bound are plagued to a slow rate $\Theta\left(\alpha^{-1/2}\right)$. Indeed, the authors of [55] showed with a replica method-style computation that *there exists* some set of weights, in the binary classification task eq. (2), that leads to $\Theta\left(\alpha^{-1/2}\right)$ rates: the uniform bound is thus tight. The gap observed between the uniform bound and the almost Bayes-optimal results observed in practice in this case is therefore not a paradox, but an illustration that the price to pay for uniform convergence is the inability to describe the optimal rates one can sometimes get in practice. Therefore, we believe, that the fact this phenomena can be observed in a such simple problem sheds an interesting light on the current debate in understanding generalization in deep learning [7].

Remarking our synthetic dataset is linearly separable, we may try to take this fact into consideration to improve the generalization rate. In particular, it can be done using the max-margin based generalization error for separable data:

**Theorem 3.1** (Hard-margin generalization bound [36, 37, 54]). *Given $S = \{\boldsymbol{x}_1, \cdots, \boldsymbol{x}_n\}$ such that $\forall \mu \in [1:n], \|\boldsymbol{x}_\mu\| \leq r$. Let $\hat{\boldsymbol{w}}$ the hard-margin SVM estimator on $S$ drawn with distribution $D$. With probability $1 - \delta$, the generalization error is bounded by*

$$e_{\mathrm{g}}(\alpha) \underset{\alpha \to \infty}{\leq} \left(4r\|\hat{\boldsymbol{w}}\| + \sqrt{\log(4/\delta)\log_2\|\hat{\boldsymbol{w}}\|}\right)/\sqrt{n}. \tag{16}$$

In our case one has $r^2 \simeq \frac{1}{d}\mathbb{E}_{\mathbf{x}}\|\mathbf{x}\|_2^2 = \frac{1}{d}\sum_{i=1}^{d}\mathbb{E}x_i^2 = 1$. On the other hand, in the large size limit, the norm of the estimator $\|\hat{\boldsymbol{w}}\|_2/\sqrt{d} \simeq \sqrt{q}$, that yields $e_{\mathrm{g}}(\alpha) \leq 4\sqrt{\frac{q}{\alpha}}$. We now need to plug the values of the norm $q$ obtained by our max-margin solution to finally obtain the results. Unfortunately, this bound turns out to be even worse than the previous one. Indeed the norm of the hard margin estimator $q$ is found to grow with $\alpha$ in the solution of the fixed point equation, and therefore the margin decay rather fast, rendering the bound vacuous. For small values of $\alpha$, one finds that $q \sim \alpha$ that provides a vacuous constant generalisation bound $e_{\mathrm{g}} \leq \Theta(1)$, while for large $\alpha$, $q \sim \alpha^2$ that yields an even worse bound $e_{\mathrm{g}} \leq \Theta(\sqrt{\alpha})$. Clearly, max-margin based bounds do not perform well in this high-dimensional example.

## 4 Reaching Bayes optimality

Given the fact that logistic and hinge losses reach values extremely close to Bayes optimal generalization performances, one may wonder if by somehow slightly altering these losses one could actually reach the Bayesian values with a plug-in estimator obtained by ERM. This is what we achieve in this section, by constructing a (non-convex) optimization problem with a specially tuned loss and regularization from the knowledge of the teacher distributions $P_{\mathrm{out}^\star}$, $P_{\mathrm{w}^\star}$, whose solution yields Bayes-optimal generalization. Indeed, in the Bayes-optimal setting, we may directly use the Bayes-optimal AMP algorithm to achieve optimal performances as proven in [10]. Nevertheless, it seems to us interesting to point out that Bayes performances, which require in principle to compute an intractable high-dimensional posterior sampling, can be obtained instead by the easier, more common and practical ERM estimation. Recent insights have shown that indeed one can sometime re-interpret Bayesian estimation as an optimization program in inverse problems [39,40,56,57]. In particular, [41] showed explicitly, on the basis of the non-rigorous replica method of statistical mechanics, that some Bayes-optimal reconstruction problems could be turned into convex M-estimation.

Matching ERM and Bayes-optimal generalization errors eqs. (6)-(14) with overlaps respectively solutions of eq. (11)-(13) and assuming that $\mathcal{Z}_{\mathrm{w}^\star}(\gamma, \Lambda) \equiv \mathbb{E}_{w \sim P_{\mathrm{w}^\star}} \exp\left(-1/2\Lambda w^2 + \gamma w\right)$ and $\mathcal{Z}_{\mathrm{out}^\star}(y, \omega, V)$, defined in (12), are log-concave in $\gamma$ and $\omega$, we define the optimal loss and regularizer $l^{\mathrm{opt}}, r^{\mathrm{opt}}$:

$$
\begin{aligned}
l^{\mathrm{opt}}(y, z) &= -\min_\omega \left( \frac{(z - \omega)^2}{2(\rho_{\mathrm{w}^\star} - q_{\mathrm{b}})} + \log \mathcal{Z}_{\mathrm{out}^\star}(y, \omega, \rho_{\mathrm{w}^\star} - q_{\mathrm{b}}) \right), \\
r^{\mathrm{opt}}(w) &= -\min_\gamma \left( \frac{1}{2}\hat{q}_{\mathrm{b}} w^2 - \gamma w + \log \mathcal{Z}_{\mathrm{w}^\star}(\gamma, \hat{q}_{\mathrm{b}}) \right), \quad \text{with } (q_{\mathrm{b}}, \hat{q}_{\mathrm{b}}) \text{ solution of eq. (13)}.
\end{aligned}
\tag{17}
$$

See SM. VI for the derivation. Following these considerations, we provide the following theorem:

**Theorem 4.1.** *The result of empirical risk minimization eq. (3) with $l^{\mathrm{opt}}$ and $r^{\mathrm{opt}}$ in eq. (17), leads to Bayes optimal generalization error in the high-dimensional regime.*

*Proof.* We present only the sketch of the proof here. First we note that the so called Bayes-optimal Generalized Approximate Message Passing (GAMP) algorithm [31], recalled in SM. VI.1, with Bayes-optimal updates $f_{\mathrm{out}}^{\mathrm{bayes}}$ and $f_{\mathrm{w}}^{\mathrm{bayes}}$ in SM. I.3.1 is provably convergent and reaches Bayes-optimal performances (see [58]). Second, we remark that the GAMP algorithm is valid for ERM estimation with the corresponding updates $f_{\mathrm{out}}^{\mathrm{erm}}(l, r), f_{\mathrm{w}}^{\mathrm{erm}}(l, r)$, defined in SM. I.3.2, for a given loss $l$ and regularizer $r$. To achieve Bayes-optimal performances, we design optimal loss $l^{\mathrm{opt}}$ and regularizer $r^{\mathrm{opt}}$ eq. 17 such that at each time step the ERM denoisers match the Bayes-optimal ones: $f_{\mathrm{out}}^{\mathrm{erm}}(l^{\mathrm{opt}}, r^{\mathrm{opt}}) = f_{\mathrm{out}}^{\mathrm{bayes}}$ and $f_{\mathrm{w}}^{\mathrm{erm}}(l^{\mathrm{opt}}, r^{\mathrm{opt}}) = f_{\mathrm{w}}^{\mathrm{bayes}}$. In this context, AMP algorithm for ERM with loss and regularization given by (17) is exactly identical to the Bayes-optimal AMP. This shows that AMP applied to the ERM problem corresponding to (17) both converge to its fixed point and reach Bayes-optimal performances. The theorem finally follows by noting (see [33, 59]) that the AMP fixed point corresponds to the extremization conditions of the loss. ☐

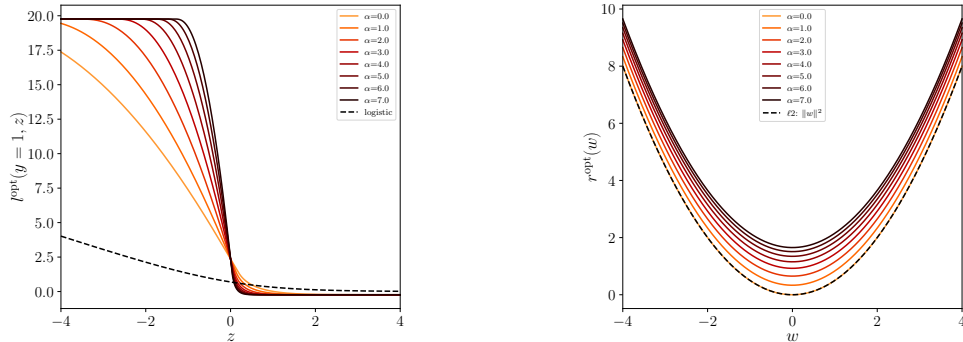

Figure 2: Optimal loss $l^{\mathrm{opt}}(y = 1, z)$ and regularizer $r^{\mathrm{opt}}(w)$ for model eq. (2) as a function of $\alpha$.

The optimal loss and regularizer $\lambda^{\mathrm{opt}}$ and $r^{\mathrm{opt}}$ for the model (2) are illustrated in Fig. (2). And numerical evidences of ERM with (17) compared to $\ell_2$ logistic regression and Bayes performances are presented in SM. VI.

## 5   Acknowledgments

This work is supported by the ERC under the European Unions Horizon 2020 Research and Innovation Program 714608-SMiLe, by the French Agence Nationale de la Recherche under grant ANR-17-CE23-0023-01 PAIL and ANR-19-P3IA-0001 PRAIRIE, and by the US National Science Foundation under grants CCF-1718698 and CCF-1910410. We would also like to thank the Kavli Institute for Theoretical Physics (KITP) for welcoming us during part of this research, with the support of the National Science Foundation under Grant No. NSF PHY-1748958. We also acknowledge support from the chaire CFM-ENS "Science des données". Part of this work was done when Yue M. Lu was visiting Ecole Normale as a CFM-ENS "Laplace" invited researcher.

## Broader Impact

Our work is theoretical in nature, and as such the potential societal consequence are difficult to foresee. We anticipate that deeper theoretical understanding of the functioning of machine learning systems will lead to their improvement in the long term.

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
