[Supplementary Material]

# —Supplementary material—
# Generalization error in high-dimensional perceptrons: Approaching Bayes error with convex optimization

**Benjamin Aubin**
Université Paris-Saclay
CNRS, CEA
Institut de physique théorique
91191, Gif-sur-Yvette, France.

**Florent Krzakala**
IdePHICS laboratory
École Polytechnique Fédérale de Lausanne
1015, Lausanne, Switzerland

**Yue M. Lu**
John A. Paulson
School of Engineering and Applied Sciences
Harvard University
Cambridge, MA 02138, USA

**Lenka Zdeborová**
SPOC laboratory
École Polytechnique Fédérale de Lausanne
1015, Lausanne, Switzerland

## Abstract

In this supplementary material (SM), we provide the proofs and computation details leading to the results presented in the main manuscript. In Sec. I, we first recall the definition of the statistical model used in Sec. 1 and we give proper definitions of the denoising distributions involved in the analysis of the Bayes-optimal and Empirical Risk Minimization (ERM) estimation. In particular, we provide the analytical expressions of the denoising functions used in Sec. 3 to analyze ridge, hinge and logistic regressions. In Sec. II, we detail the computation of the binary classification generalization error leading to the expressions in Proposition. 2.1 and Thm. 2.4 respectively for ERM and Bayes-optimal estimation. In Sec. III, we present the proofs of the central theorems stated in Sec. 2. In particular, we derive the Gordon-based proof of the Thm. 2.2 in the more general *regression* (real-valued) version and provide as well the proof of Corollary. 2.3 which establishes the equivalence between the set of fixed-point equations of the Gordon's proof in the binary classification case and the one resulting from the heuristic replica computation. The corresponding statistical physics framework used to analyze Bayes and ERM statistical estimations and the replica computation leading to expressions in Corollary. 2.3 are detailed In Sec. IV. The section V is devoted to provide additional technical details on the results with $\ell_2$ regularization addressed in Sec. 3. In particular, we present the large $\alpha$ expansions of the generalization error for the *Bayes-optimal*, *ridge*, *pseudo-inverse* and *max-margin* estimators, and we investigate the performances of logistic regression on non-linearly separable data. Finally in Sec. VI, we show the derivation of the fine-tuned loss and regularizer provably leading to Bayes-optimal performances, as explained and advocated in Sec. 4, and we show some numerical evidences that ERM achieves indeed Bayes-optimal error in Fig. 3.

# Contents

# I  Definitions and notations

## I.1  Statistical model

We recall the supervised machine learning task considered in the main manuscript eq. (1), whose dataset is generated by a single layer neural network, often named a *teacher*, that belongs to the Generalized Linear Model (GLM) class. Therefore we assume the $n$ samples are drawn according to

$$\mathbf{y} = \varphi_{\text{out}}^{\star} \left( \frac{1}{\sqrt{d}} \mathrm{X} \mathbf{w}^{\star} \right) \Leftrightarrow \mathbf{y} \sim P_{\text{out}}^{\star}(.) \, , \tag{I.1}$$

where $\mathbf{w}^{\star} \in \mathbb{R}^d$ denotes the ground truth vector drawn from a probability distribution $P_{\mathbf{w}^{\star}}$ with second moment $\rho_{\mathbf{w}^{\star}} \equiv \frac{1}{d} \mathbb{E} \left[ \|\mathbf{w}^{\star}\|_2^2 \right]$ and $\varphi_{\text{out}}^{\star}$ represents a deterministic or stochastic activation function equivalently associated to a distribution $P_{\text{out}}^{\star}$. The input data matrix $\mathrm{X} = (\mathbf{x}_\mu)_{\mu=1}^{n} \in \mathbb{R}^{n \times d}$ contains i.i.d Gaussian vectors, i.e $\forall \mu \in [1:n]$, $\mathbf{x}_\mu \sim \mathcal{N}(\mathbf{0}, \mathrm{I}_d)$.

## I.2  Bayes-optimal and ERM estimation

Inferring the above statistical model from observations $\{\mathbf{y}, \mathrm{X}\}$ can be tackled in several ways. In particular, Bayesian inference provides a generic framework for statistical estimation based on the high-dimensional, often intractable, posterior distribution

$$\mathbb{P}(\mathbf{w}|\mathbf{y}, \mathrm{X}) = \frac{\mathbb{P}(\mathbf{y}|\mathbf{w}, \mathrm{X}) \, \mathbb{P}(\mathbf{w})}{\mathbb{P}(\mathbf{y}, \mathrm{X})} \, . \tag{I.2}$$

Estimating the average of the above posterior distribution in the case we have access to the ground truth prior distributions $\mathbb{P}(\mathbf{y}|\mathbf{w}, \mathrm{X}) = P_{\text{out}^{\star}}(\mathbf{y}|\mathbf{z})$ with $\mathbf{z} \equiv \frac{1}{\sqrt{d}} \mathrm{X}\mathbf{w}$ and $\mathbb{P}(\mathbf{w}) = P_{\mathbf{w}^{\star}}(\mathbf{w})$, refers to Bayes-optimal estimation and leads to the corresponding Minimal Mean-Squared Error (MMSE) estimator $\hat{\mathbf{w}}_{\text{mmse}} = \mathbb{E}_{\mathbb{P}(\mathbf{w}|\mathbf{y}, \mathrm{X})}[\mathbf{w}]$. It has been rigorously analyzed in details in [1] for the whole GLM class eq. (I.1). Another celebrated approach and widely used in practice is the Empirical Risk Minimization (ERM) that minimizes instead a regularized loss: $\hat{\mathbf{w}}_{\text{erm}} = \operatorname{argmin}_{\mathbf{w}} [\mathcal{L}(\mathbf{w}; \mathbf{y}, \mathrm{X})]$ with

$$\mathcal{L}(\mathbf{w}; \mathbf{y}, \mathrm{X}) = \sum_{\mu=1}^{n} l(\mathbf{w}; y_\mu, \mathbf{x}_\mu) + r(\mathbf{w}) \, . \tag{I.3}$$

Interestingly analyzing the ERM estimation may be included in the above Bayesian framework. Indeed exponentiating eq. (I.3), we see that minimizing the loss $\mathcal{L}$ is equivalent to maximize the posterior distribution $\mathbb{P}(\mathbf{w}|\mathbf{y}, \mathrm{X}) = e^{-\mathcal{L}(\mathbf{w}; \mathbf{y}, \mathrm{X})}$ if we choose carefully the prior distributions as functions of the regularizer $r$ and the loss $l$:

$$-\log \mathbb{P}(\mathbf{y}|\mathbf{w}, \mathrm{X}) = l(\mathbf{w}; \mathbf{y}, \mathrm{X}) \, , \quad -\log \mathbb{P}(\mathbf{w}) = r(\mathbf{w}) \, . \tag{I.4}$$

Computing the maximum of the posterior $\mathbb{P}(\mathbf{y}|\mathbf{w}, \mathrm{X})$ refers instead to the so-called Maximum A Posteriori (MAP) estimator, and therefore analyzing the empirical minimization of (I.3) is equivalent to obtain the performance of the MAP estimator with prior distributions given by (I.4). Thus both the study of ERM (MAP) and Bayes-optimal (MMSE) estimations are simply reduced to the analysis of the posterior eq. (I.2).

## I.3  Denoising distributions and updates

Analyzing the posterior distribution eq. (I.2) in the high-dimensional regime [1] will boil down to introducing the scalar denoising distributions $Q_{\text{w}}, Q_{\text{out}}$ and their respective normalizations $\mathcal{Z}_{\text{w}}, \mathcal{Z}_{\text{out}}$

$$Q_{\text{w}}(w; \gamma, \Lambda) \equiv \frac{P_{\text{w}}(w)}{\mathcal{Z}_{\text{w}}(\gamma, \Lambda)} e^{-\frac{1}{2}\Lambda w^2 + \gamma w} \, , \quad Q_{\text{out}}(z; y, \omega, V) \equiv \frac{P_{\text{out}}(y|z)}{\mathcal{Z}_{\text{out}}(y, \omega, V)} \frac{e^{-\frac{1}{2}V^{-1}(z-\omega)^2}}{\sqrt{2\pi V}} \, ,$$
$$\mathcal{Z}_{\text{w}}(\gamma, \Lambda) \equiv \mathbb{E}_{w \sim P_{\text{w}}} \left[ e^{-\frac{1}{2}\Lambda w^2 + \gamma w} \right] \, , \quad \mathcal{Z}_{\text{out}}(y, \omega, V) \equiv \mathbb{E}_{z \sim \mathcal{N}(0,1)} \left[ P_{\text{out}}\left( y | \sqrt{V} z + \omega \right) \right] \, . \tag{I.5}$$

We define as well the denoising functions, that play a central role in Bayesian inference. Note in particular that they correspond to the *updates* of the Approximate Message Passing algorithm in [2] that we recalled in Sec. VI.1. They are defined as the derivatives of $\log \mathcal{Z}_{\mathrm{w}}$ and $\log \mathcal{Z}_{\mathrm{out}}$, namely

$$f_{\mathrm{w}}(\gamma, \Lambda) \equiv \partial_{\gamma} \log (\mathcal{Z}_{\mathrm{w}}) = \mathbb{E}_{Q_{\mathrm{w}}}[w] \quad \text{and} \quad \partial_{\gamma} f_{\mathrm{w}}(\gamma, \Lambda) \equiv \mathbb{E}_{Q_{\mathrm{w}}}\left[w^2\right] - f_{\mathrm{w}}^2$$

$$f_{\mathrm{out}}(y, \omega, V) \equiv \partial_{\omega} \log (\mathcal{Z}_{\mathrm{out}}) = V^{-1} \mathbb{E}_{Q_{\mathrm{out}}}[z - \omega] \quad \text{and} \quad \partial_{\omega} f_{\mathrm{out}}(y, \omega, V) \equiv \frac{\partial f_{\mathrm{out}}(y, \omega, V)}{\partial \omega} .$$

$$(\text{I.6})$$

### I.3.1 Bayes-optimal - MMSE denoising functions

In Bayes-optimal estimation, the ground truth prior and channel distributions $P_{\mathrm{w}^{\star}}(w)$ and $P_{\mathrm{out}^{\star}}(y|z)$ of the *teacher* eq. (1) are known. Hence, replacing $P_{\mathrm{w}}$ and $P_{\mathrm{out}}$ in (I.5), we obtain the Bayes-optimal scalar denoising distributions in terms of which the Bayes-optimal free entropy eq. (IV.18) is written

$$Q_{\mathrm{w}^{\star}}(w; \gamma, \Lambda) \equiv \frac{P_{\mathrm{w}^{\star}}(w)}{\mathcal{Z}_{\mathrm{w}^{\star}}(\gamma, \Lambda)} e^{-\frac{1}{2}\Lambda w^2 + \gamma w}, \quad Q_{\mathrm{out}^{\star}}(z; y, \omega, V) \equiv \frac{P_{\mathrm{out}^{\star}}(y|z)}{\mathcal{Z}_{\mathrm{out}^{\star}}(y, \omega, V)} \frac{e^{-\frac{1}{2}V^{-1}(z-\omega)^2}}{\sqrt{2\pi V}} ,$$

$$(\text{I.7})$$

and the denoising updates are therefore given by eq. (I.6) with the corresponding distributions

$$f_{\mathrm{w}^{\star}}(\gamma, \Lambda) \equiv \partial_{\gamma} \log \mathcal{Z}_{\mathrm{w}^{\star}}(\gamma, \Lambda), \qquad f_{\mathrm{out}^{\star}}(y, \omega, V) \equiv \partial_{\omega} \log \mathcal{Z}_{\mathrm{out}^{\star}}(y, \omega, V). \qquad (\text{I.8})$$

### I.3.2 ERM - MAP denoising functions

Before defining similar denoising functions to analyze the MAP for ERM estimation, we first recall the definition of the Moreau-Yosida regularization.

**Moreau-Yosida regularization and proximal** Let $\Sigma > 0$, $f(, z)$ a convex function in $z$. Defining the regularized functional

$$\mathcal{L}_{\Sigma}[f(, .)](z; x) = f(, z) + \frac{1}{2\Sigma}(z - x)^2 , \qquad (\text{I.9})$$

the Moreau-Yosida regularization $\mathcal{M}_{\Sigma}$ and the proximal map $\mathcal{P}_{\Sigma}$ are defined by

$$\mathcal{P}_{\Sigma}[f(, .)](x) = \operatorname{argmin}_z \mathcal{L}_{\Sigma}[f(, .)](z; x) = \operatorname{argmin}_z \left[ f(, z) + \frac{1}{2\Sigma}(z - x)^2 \right] , \qquad (\text{I.10})$$

$$\mathcal{M}_{\Sigma}[f(, .)](x) = \min_z \mathcal{L}_{\Sigma}[f(, .)](z; x) = \min_z \left[ f(, z) + \frac{1}{2\Sigma}(z - x)^2 \right] , \qquad (\text{I.11})$$

where $(, z)$ denotes all the arguments of the function $f$, where $z$ plays a central role. The MAP denoising functions for any convex loss $l(, .)$ and convex separable regularizer $r(.)$ can be written in terms of the Moreau-Yosida regularization or the proximal map as follows

$$f_{\mathrm{w}}^{\mathrm{map},r}(\gamma, \Lambda) \equiv \mathcal{P}_{\Lambda^{-1}}\left[r(.)\right](\Lambda^{-1}\gamma) = \Lambda^{-1}\gamma - \Lambda^{-1}\partial_{\Lambda^{-1}\gamma} \mathcal{M}_{\Lambda^{-1}}\left[r(.)\right](\Lambda^{-1}\gamma),$$

$$f_{\mathrm{out}}^{\mathrm{map},l}(y, \omega, V) \equiv -\partial_{\omega} \mathcal{M}_V[l(y, .)](\omega) = V^{-1}\left(\mathcal{P}_V[l(y, .)](\omega) - \omega\right) .$$

$$(\text{I.12})$$

The above updates can be considered as definitions, but it is instructive to derive them from the generic definition of the denoising distributions eq. (I.6) if we maximize the posterior distribution. This is done by taking, in a physics language, a *zero temperature* limit and we present it in details in the next paragraph.

**Derivation of the MAP updates** To have access to the maximum of the generic distributions eq. (I.5), we introduce a *fictive* noise/temperature $\Delta$ or inverse temperature $\beta$, $\Delta = \frac{1}{\beta}$. In particular for Bayes-optimal estimation this temperature is finite and fixed to $\Delta = \beta = 1$. Indeed with the mapping eq. (I.4), minimizing the loss function $\mathcal{L}$ (I.3) is equivalent to maximize the posterior distribution. Therefore it can be done by taking the *zero noise/temperature* limit $\Delta \to 0$ of the channel and prior denoising distributions $Q_{\mathrm{out}}$ and $Q_{\mathrm{w}}$. It is the purpose of the following paragraphs where we present the derivation leading to the result (I.12).

**Channel** Using the mapping eq. (I.4), we assume that the channel distribution can be expressed as $\mathbb{P}(y|z) \propto e^{-l(y,z)}$. Therefore we introduce the corresponding channel distribution $P_{\text{out}}$ at finite temperature $\Delta$ associated to the convex loss $l(y,z)$

$$P_{\text{out}}^{\text{map}}(y|z) = \frac{e^{-\frac{1}{\Delta}l(y,z)}}{\sqrt{2\pi\Delta}} .$$

Note that the case of the square loss $l(y,z) = \frac{1}{2}(y-z)^2$ is very specific. Its channel distribution simply reads $P_{\text{out}}(y|z) = \frac{e^{-\frac{1}{2\Delta}(y-z)^2}}{\sqrt{2\pi\Delta}}$ and is therefore equivalent to predict labels $y$ according to a noisy Gaussian linear model $y = z + \sqrt{\Delta}\xi$, where $\xi \sim \mathcal{N}(0,1)$ and $\Delta$ denotes therefore the *real* noise of the model.

In order to obtain a non trivial limit and a closed set of equations when $\Delta \to 0$, we must define rescaled variables as follows:

$$V_\dagger \equiv \lim_{\Delta \to 0} \frac{V}{\Delta} , \qquad\qquad f_{\text{out},\dagger}^{\text{map}}(y,\omega,V_\dagger) \equiv \lim_{\Delta \to 0} \Delta \times f_{\text{out}}^{\text{map}}(y,\omega,V) ,$$

where we denote the rescaled quantities after taking the limit $\Delta \to 0$ by $\dagger$. Similarly to eq. (I.9), we introduce therefore the rescaled functional

$$\mathcal{L}_{V_\dagger}[l(y,.)](z;\omega) = l(y,z) + \frac{1}{2V_\dagger}(z-\omega)^2 , \tag{I.13}$$

such that, injecting $P_{\text{out}}^{\text{map}}$, the channel denoising distribution $Q_{\text{out}}^{\text{map}}$ and the corresponding partition function $\mathcal{Z}_{\text{out}}^{\text{map}}$ eq. (I.5) simplify in the zero temperature limit as follows:

$$Q_{\text{out}}^{\text{map}}(z;y,\omega,V) \equiv \lim_{\Delta \to 0} \frac{e^{-\frac{1}{\Delta}l(y,z)+\frac{1}{2V}(z-\omega)^2}}{\sqrt{2\pi\Delta V_\dagger}\sqrt{2\pi\Delta}} = \lim_{\Delta \to 0} \frac{e^{-\frac{1}{\Delta}\mathcal{L}_{V_\dagger}[l(y,.)](z;\omega)}}{\sqrt{2\pi\Delta V_\dagger}\sqrt{2\pi\Delta}} , \tag{I.14}$$

$$\propto \delta\left(z - \mathcal{P}_{V_\dagger}[l(y,.)](\omega)\right)$$

$$\mathcal{Z}_{\text{out}}^{\text{map}}(y,\omega,V) = \lim_{\Delta \to 0} \int_{\mathbb{R}} \mathrm{d}z Q_{\text{out}}^{\text{map}}(z;y,\omega,V) = \lim_{\Delta \to 0} \frac{e^{-\frac{1}{\Delta}\mathcal{M}_{V_\dagger}[l(y,.)](\omega)}}{\sqrt{2\pi\Delta V_\dagger}\sqrt{2\pi\Delta}} , \tag{I.15}$$

that involve the proximal map and the Moreau-Yosida regularization defined in eq. (I.11). Finally taking the zero temperature limit, the MAP denoising function $f_{\text{out},\dagger}^{\text{map}}$ leads to the result (I.12):

$$\begin{aligned} f_{\text{out},\dagger}^{\text{map}}(y,\omega,V_\dagger) &\equiv \lim_{\Delta \to 0} \Delta \times f_{\text{out}}^{\text{map}}(y,\omega,V) \\ &\equiv \lim_{\Delta \to 0} \Delta \times \partial_\omega \log \mathcal{Z}_{\text{out}}^{\text{map}} \equiv \lim_{\Delta \to 0} \Delta V^{-1} \mathbb{E}_{Q_{\text{out}}^{\text{map}}}[z-\omega] \\ &= -\partial_\omega \mathcal{M}_{V_\dagger}[l(y,.)](\omega) = V_\dagger^{-1}\left(\mathcal{P}_{V_\dagger}[l(y,.)](\omega) - \omega\right) . \end{aligned} \tag{I.16}$$

**Prior** Similarly as above, using the mapping eq. (I.4), for a convex and separable regularizer $r$, the corresponding prior distribution at temperature $\Delta$ can be written

$$P_{\text{w}}^{\text{map}}(w) = e^{-\frac{1}{\Delta}r(w)} .$$

Note that at $\Delta = 1$ the classical $\ell_1$ regularization with strength $\lambda$, $r^{\ell_1}(w) = -\lambda|w|$, and the $\ell_2$ regularization $r^{\ell_2}(w) = -\lambda w^2/2$ are equivalent to choosing a Laplace prior $P_{\text{w}}(w) \propto e^{-\lambda|w|}$ or a Gaussian prior $P_{\text{w}}(w) \propto e^{-\frac{\lambda w^2}{2}}$. To obtain a meaningful limit as $\Delta \to 0$, we again introduce the following rescaled variables

$$\Lambda_\dagger \equiv \lim_{\Delta \to 0} \Delta \times \Lambda , \qquad\qquad \gamma_\dagger \equiv \lim_{\Delta \to 0} \Delta \times \gamma ,$$

and the functional

$$\mathcal{L}_{\Lambda_\dagger^{-1}}[r(.)](w;\Lambda_\dagger^{-1}\gamma_\dagger) = r(w) + \frac{1}{2}\Lambda_\dagger\left(w-\Lambda_\dagger^{-1}\gamma_\dagger\right)^2 = \left[r(w) + \frac{1}{2}\Lambda_\dagger w^2 - \gamma_\dagger w\right] + \frac{1}{2}\gamma_\dagger^2\Lambda_\dagger^{-1} , \tag{I.17}$$

such that in the zero temperature limit, the prior denoising distribution $Q_{\text{w}}^{\text{map}}$ and the partition function $\mathcal{Z}_{\text{w}}^{\text{map}}$ reduce to

$$Q_{\text{w}}^{\text{map}}(w;\gamma,\Lambda) \equiv \lim_{\Delta\to 0} P_{\text{w}}(w)e^{-\frac{1}{2}\Lambda w^2+\gamma w} = \lim_{\Delta\to 0} e^{-\frac{1}{\Delta}\mathcal{L}_{\Lambda_\dagger^{-1}}[r](w;\Lambda_\dagger^{-1}\gamma_\dagger)} e^{-\frac{1}{2\Delta}\gamma_\dagger^2\Lambda_\dagger^{-1}}$$

$$\propto \delta\left(w - \mathcal{P}_{\Lambda_\dagger^{-1}}[r]\left(\Lambda_\dagger^{-1}\gamma_\dagger\right)\right) \tag{I.18}$$

$$\mathcal{Z}_{\text{w}}^{\text{map}}(y,\omega,V) = \lim_{\Delta\to 0}\int_{\mathbb{R}} dw Q_{\text{w}}^{\text{map}}(w;\gamma,\Lambda) = \lim_{\Delta\to 0} e^{-\frac{1}{\Delta}\mathcal{M}_{\Lambda_\dagger^{-1}}[r](\Lambda_\dagger^{-1}\gamma_\dagger)} e^{-\frac{1}{2\Delta}\gamma_\dagger^2\Lambda_\dagger^{-1}}, \tag{I.19}$$

that involve again the proximal map $\mathcal{P}_{\Lambda_\dagger^{-1}}$ and the Moreau-Yosida regularization $\mathcal{M}_{\Lambda_\dagger^{-1}}$ defined in eq. (I.11). Finally the MAP denoising update $f_{\text{w},\dagger}^{\text{map}}$ is simply given by:

$$f_{\text{w},\dagger}^{\text{map}}(\gamma_\dagger,\Lambda_\dagger) \equiv \lim_{\Delta\to 0} f_{\text{w}}^{\text{map}}(\gamma,\Lambda) = \lim_{\Delta\to 0}\partial_\gamma \log\mathcal{Z}_{\text{w}}^{\text{map}} \equiv \lim_{\Delta\to 0}\mathbb{E}_{Q_{\text{w}}^{\text{map}}}[w]$$

$$= \lim_{\Delta\to 0}\partial_\gamma\left(-\frac{1}{\Delta}\mathcal{M}_{\Lambda_\dagger^{-1}}[r(.)]\left(\Lambda_\dagger^{-1}\gamma_\dagger\right) - \frac{1}{2\Delta}\gamma_\dagger^2\Lambda_\dagger^{-1}\right)$$

$$= \partial_{\gamma_\dagger}\left(-\mathcal{M}_{\Lambda_\dagger^{-1}}[r(.)]\left(\Lambda_\dagger^{-1}\gamma_\dagger\right) - \frac{1}{2}\gamma_\dagger^2\Lambda_\dagger^{-1}\right) \tag{I.20}$$

$$= \Lambda_\dagger^{-1}\gamma_\dagger - \Lambda_\dagger^{-1}\partial_{\Lambda_\dagger^{-1}\gamma_\dagger}\mathcal{M}_{\Lambda_\dagger^{-1}}[r(.)]\left(\Lambda_\dagger^{-1}\gamma_\dagger\right) = \mathcal{P}_{\Lambda_\dagger^{-1}}[r(.)]\left(\Lambda_\dagger^{-1}\gamma_\dagger\right)$$

$$= \text{argmin}_w\left[r(w) + \frac{1}{2}\Lambda_\dagger(w - \Lambda_\dagger^{-1}\gamma_\dagger)^2\right] = \text{argmin}_w\left[r(w) + \frac{1}{2}\Lambda_\dagger w^2 - \gamma_\dagger w\right],$$

and we recover the result (I.12).

### I.4 Applications

In this section we list the explicit expressions of the Bayes-optimal eq. (I.8) and ERM eq. (I.12) denoising functions largely used to produce the examples in Sec. 3.

#### I.4.1 Bayes-optimal updates

The Bayes-optimal denoising functions (I.8) are detailed in the case of a *linear*, *sign* and *rectangle door* channel with a Gaussian noise $\xi \sim \mathcal{N}(0,1)$ and variance $\Delta \geq 0$, and for *Gaussian* and *sparse-binary* weights.

**Channel**
- *Linear:* $y = \varphi_{\text{out}^\star}(z) = z + \sqrt{\Delta}\xi$

$$\mathcal{Z}_{\text{out}^\star}(y,\omega,V) = \mathcal{N}_\omega(y,\Delta^\star + V),$$
$$f_{\text{out}^\star}(y,\omega,V) = (\Delta^\star + V)^{-1}(y-\omega), \quad \partial_\omega f_{\text{out}^\star}(y,\omega,V) = -(\Delta^\star + V)^{-1}. \tag{I.21}$$

- *Sign:* $y = \varphi_{\text{out}^\star}(z) = sign(z) + \sqrt{\Delta^\star}\xi$

$$\mathcal{Z}_{\text{out}^\star}(y,\omega,V) = \mathcal{N}_y(1,\Delta^\star)\frac{1}{2}\left(1 + \text{erf}\left(\frac{\omega}{\sqrt{2V}}\right)\right) + \mathcal{N}_y(-1,\Delta^\star)\frac{1}{2}\left(1 - \text{erf}\left(\frac{\omega}{\sqrt{2V}}\right)\right),$$
$$f_{\text{out}^\star}(y,\omega,V) = \frac{\mathcal{N}_y(1,\Delta^\star) - \mathcal{N}_y(-1,\Delta^\star)}{\mathcal{Z}_{\text{out}^\star}(y,\omega,V)}\mathcal{N}_\omega(0,V). \tag{I.22}$$

- *Rectangle door:* $y = \varphi_{\text{out}^\star}(z) = \mathbb{1}(\kappa_m \leq z \leq \kappa_M) - \mathbb{1}(z \leq \kappa_m \text{ or } z \geq \kappa_M) + \sqrt{\Delta^\star}\xi$
  For $\kappa_m < \kappa_M$, we obtain

$$\mathcal{Z}_{\mathrm{out}^\star}(y,\omega,V) = \mathcal{N}_y(1,\Delta^\star)\frac{1}{2}\left(\mathrm{erf}\left(\frac{\kappa_M-\omega}{\sqrt{2V}}\right)-\mathrm{erf}\left(\frac{\kappa_m-\omega}{\sqrt{2V}}\right)\right)$$

$$+\mathcal{N}_y(-1,\Delta^\star)\frac{1}{2}\left(1-\frac{1}{2}\left(\mathrm{erf}\left(\frac{\kappa_M-\omega}{\sqrt{2V}}\right)-\mathrm{erf}\left(\frac{\kappa_m-\omega}{\sqrt{2V}}\right)\right)\right), \quad \text{(I.23)}$$

$$f_{\mathrm{out}^\star}(y,\omega,V) = \frac{1}{\mathcal{Z}_{\mathrm{out}}}\left(\mathcal{N}_y(1,\Delta^\star)\left(-\mathcal{N}_\omega(\kappa_M,V)+\mathcal{N}_\omega(\kappa_m,V)\right)\right.$$

$$\left.+\mathcal{N}_y(-1,\Delta^\star)\left(\mathcal{N}_\omega(\kappa_M,V)-\mathcal{N}_\omega(\kappa_m,V)\right)\right).$$

**Prior**

• *Gaussian weights:* $w \sim P_{\mathrm{w}}(w) = \mathcal{N}_w(\mu,\sigma)$

$$\mathcal{Z}_{\mathrm{w}^\star}(\gamma,\Lambda) = \frac{e^{\frac{\gamma^2\sigma+2\gamma\mu-\Lambda\mu^2}{2(\Lambda\sigma+1)}}}{\sqrt{\Lambda\sigma+1}}, \quad f_{\mathrm{w}^\star}(\gamma,\Lambda) = \frac{\gamma\sigma+\mu}{1+\Lambda\sigma}, \quad \partial_\gamma f_{\mathrm{w}^\star}(\gamma,\Lambda) = \frac{\sigma}{1+\Lambda\sigma}. \quad \text{(I.24)}$$

• *Sparse-binary weights:* $w \sim P_{\mathrm{w}}(w) = \rho\delta(w)+(\rho-1)\frac{1}{2}\left(\delta(w-1)+\delta(w+1)\right)$

$$\mathcal{Z}_{\mathrm{w}^\star}(\gamma,\Lambda) = \rho + e^{-\frac{\Lambda}{2}}(1-\rho)\cosh(\gamma),$$

$$f_{\mathrm{w}^\star}(\gamma,\Lambda) = \frac{e^{-\frac{\Lambda}{2}}(1-\rho)\sinh(\gamma)}{\rho + e^{-\frac{\Lambda}{2}}(1-\rho)\cosh(\gamma)}, \quad \partial_\gamma f_{\mathrm{w}^\star}(\gamma,\Lambda) = \frac{e^{-\frac{\Lambda}{2}}(1-\rho)\cosh(\gamma)}{\rho + e^{-\frac{\Lambda}{2}}(1-\rho)\cosh(\gamma)}. \quad \text{(I.25)}$$

### I.4.2 ERM updates

The ERM denoising functions (I.12) have, very often, no explicit expression except for the *square* and *hinge* losses, and for $\ell_1$, $\ell_2$ regularizations that are analytical. However, in the particular case of a two times differentiable convex loss the denoising functions can still be written as the solution of an implicit equation detailed below.

**Convex losses**

• *Square loss*

The proximal map for the square loss $l^{\mathrm{square}}(y,z) = \frac{1}{2}(y-z)^2$ is easily obtained and reads

$$\mathcal{P}_V\left[\frac{1}{2}(y,.)^2\right](\omega) = \mathrm{argmin}_z\left[\frac{1}{2}(y-z)^2 + \frac{1}{2V}(z-\omega)^2\right] = (1+V)^{-1}(\omega+yV).$$

Therefore (I.12) yields

$$f_{\mathrm{out}}^{\mathrm{square}}(y,\omega,V) = V^{-1}\left(\mathcal{P}_V\left[\frac{1}{2}(y,.)^2\right](\omega)-\omega\right) = (1+V)^{-1}(y-\omega),$$

$$\partial_\omega f_{\mathrm{out}}^{\mathrm{square}}(y,\omega,V) = -(1+V)^{-1}. \quad \text{(I.26)}$$

• *Hinge loss*

The proximal map of the hinge loss $l^{\mathrm{hinge}}(y,z) = \max(0,1-yz)$

$$\mathcal{P}_V\left[l^{\mathrm{hinge}}(y,.)\right](\omega) = \mathrm{argmin}_z\left[\underbrace{\max(0,1-yz) + \frac{1}{2V}(z-\omega)^2}_{\equiv\mathcal{L}_0}\right] \equiv z^\star(y,\omega,V).$$

can be expressed analytically by distinguishing all the possible cases:

- $1 - yz < 0$: $\mathcal{L}_0 = \frac{1}{2V}(z - \omega)^2 \Rightarrow z^\star = \omega$ if $yz^\star < 1 \Leftrightarrow z^\star = \omega$ if $\omega y < 1$.
- $1 - yz > 0$: $\mathcal{L}_0 = \frac{1}{2V}(z - \omega)^2 + 1 - yz \Rightarrow (z^\star - \omega) = yV \Leftrightarrow z^\star = \omega + Vy$ if $1 - yz^\star > 0 \Leftrightarrow z^\star = \omega + Vy$ if $\omega y < 1 - y^2 V = 1 - V$, as $y^2 = 1$.
- Hence we have one last region to study $1 - V < \omega y < 1$. It follows $y(1 - V) < \omega < y$:

$$\frac{1}{2V}(z - y)^2 \le \frac{1}{2V}(z - \omega)^2 \Rightarrow z^\star = y.$$

Finally we obtain a simple analytical expression for the proximal and its derivative

$$\mathcal{P}_V\left[l^{\mathrm{hinge}}(y, .)\right](\omega) = \begin{cases} \omega + Vy & \text{if } \omega y < 1 - V \\ y & \text{if } 1 - V < \omega y < 1 \\ \omega & \text{if } \omega y > 1 \end{cases}, \partial_\omega \mathcal{P}_V\left[l^{\mathrm{hinge}}(y, .)\right](\omega) = \begin{cases} 1 & \text{if } \omega y < 1 - V \\ 0 & \text{if } 1 - V < \omega y < 1 \\ 1 & \text{if } \omega y > 1 \end{cases}.$$

Hence with (I.12), the hinge denoising function and its derivative read

$$f_{\mathrm{out}}^{\mathrm{hinge}}(y, \omega, V) = \begin{cases} y & \text{if } \omega y < 1 - V \\ \frac{(y - \omega)}{V} & \text{if } 1 - V < \omega y < 1 \\ 0 & \text{otherwise} \end{cases}, \partial_\omega f_{\mathrm{out}}^{\mathrm{hinge}}(y, \omega, V) = \begin{cases} -\frac{1}{V} & \text{if } 1 - V < \omega y < 1 \\ 0 & \text{otherwise} \end{cases}.$$

$$\text{(I.27)}$$

- *Generic differentiable convex loss*

In general, finding the proximal map in (I.12) is intractable. In particular, it is the case for the logistic loss considered in Sec. V.5. However assuming the convex loss is a generic two times differentiable function $l \in \mathcal{D}^2$, taking the derivative of the proximal map

$$\mathcal{P}_V\left[l(y, .)\right](\omega) = \mathrm{argmin}_z\left[l(y, z) + \frac{1}{2V}(z - \omega)^2\right] \equiv z^\star(y, \omega, V),$$

verifies therefore the implicit equations:

$$z^\star(y, \omega, V) = \omega - V\partial_z l(y, z^\star(y, \omega, V)), \quad \partial_\omega z^\star(y, \omega, V) = \left(1 + V\partial_z^2 l(y, z^\star(y, \omega, V))\right)^{-1}.$$

$$\text{(I.28)}$$

Once those equations solved, the denoising function and its derivative are simply expressed as

$$f_{\mathrm{out}}^{\mathrm{diff}}(y, \omega, V) = V^{-1}(z^\star(y, \omega, V) - \omega), \quad \partial_\omega f_{\mathrm{out}}^{\mathrm{diff}}(y, \omega, V) = V^{-1}(\partial_\omega z^\star(y, \omega, V) - 1),$$

$$\text{(I.29)}$$

with $z^\star(y, \omega, V) = \mathcal{P}_V\left[l(y, .)\right](\omega)$ solution of (I.28).

## Regularizations

- *$\ell_2$ regularization*

Using the definition of the prior update in eq. (I.12) for the $\ell_2$ regularization $r(w) = \frac{\lambda w^2}{2}$, we obtain

$$f_{\mathrm{w}}^{\ell_2}(\gamma, \Lambda) = \mathrm{argmin}_w\left[\frac{\lambda w^2}{2} + \frac{1}{2}\Lambda w^2 - \gamma w\right] = \frac{\gamma}{\lambda + \Lambda},$$

$$\partial_\gamma f_{\mathrm{w}}^{\ell_2}(\gamma, \Lambda) = \frac{1}{\lambda + \Lambda} \quad \text{and} \quad \mathcal{Z}_{\mathrm{w}}^{\ell_2}(\gamma, \Lambda) = \exp\left(\frac{\gamma^2 \Lambda}{2(\lambda + \Lambda)^2}\right).$$

$$\text{(I.30)}$$

- *$\ell_1$ regularization*

Performing the same computation for the $\ell_1$ regularization $r(w) = \lambda|w|$, we obtain

$$f_{\mathrm{w}}^{\ell_1}(\gamma, \Lambda) = \mathrm{argmin}_w\left[\lambda\|w\| + \frac{1}{2}\Lambda w^2 - \gamma w\right] = \begin{cases} \frac{\gamma - \lambda}{\Lambda} & \gamma > \lambda \\ \frac{\gamma + \lambda}{\Lambda} & \gamma + \lambda < 0 \\ 0 & \text{otherwise} \end{cases},$$

$$\partial_\gamma f_{\mathrm{w}}^{\ell_1}(\gamma, \Lambda) = \begin{cases} \frac{1}{\Lambda} & \|\gamma\| > \lambda \\ 0 & \text{otherwise} \end{cases}.$$

$$\text{(I.31)}$$

## II Binary classification generalization errors

In this section, we present the computation of the asymptotic generalization error

$$e_{\mathrm{g}}(\alpha) \equiv \lim_{d\to\infty} \mathbb{E}_{y,\mathbf{x}} \mathbb{1}\left[y \neq \hat{y}\left(\hat{\mathbf{w}}(\alpha);\mathbf{x}\right)\right] , \tag{II.1}$$

leading to expressions in Proposition. 2.1 and Thm. 2.4. The computation at finite dimension is similar if we do not consider the limit $d \to \infty$.

### II.1 General case

The generalization error $e_{\mathrm{g}}$ is the prediction error of the estimator $\hat{\mathbf{w}}$ on new samples $\{\mathbf{y}, \mathrm{X}\}$, where X is an i.i.d Gaussian matrix and $\mathbf{y}$ are $\pm 1$ labels generated according to (I.1):

$$\mathbf{y} = \varphi_{\mathrm{out}^\star}(\mathbf{z}) \quad \text{with} \quad \mathbf{z} = \frac{1}{\sqrt{d}}\mathrm{X}\mathbf{w}^\star . \tag{II.2}$$

As the model fitted by ERM may not lead to binary outputs, we may add a non-linearity $\varphi : \mathbb{R} \mapsto \{\pm 1\}$ (for example a sign) on top of it to insure to obtain binary outputs $\hat{\mathbf{y}} = \pm 1$ according to

$$\hat{\mathbf{y}} = \varphi(\hat{\mathbf{z}}) \quad \text{with} \quad \hat{\mathbf{z}} = \frac{1}{\sqrt{d}}\mathrm{X}\hat{\mathbf{w}} . \tag{II.3}$$

The classification generalization error is given by the probability that the predicted labels $\hat{y}$ and the true labels $y$ do not match. To compute it, first note that the vectors $(\mathbf{z}, \hat{\mathbf{z}})$ averaged over all possible ground truth vectors $\mathbf{w}^\star$ (or equivalently labels $y$) and input matrix X follow in the large size limit a joint Gaussian distribution with zero mean and covariance matrix

$$\sigma = \lim_{d\to\infty} \mathbb{E}_{\mathbf{w}^\star,\mathrm{X}} \frac{1}{d} \begin{bmatrix} \mathbf{w}^{\star\mathsf{T}}\mathbf{w}^\star & \mathbf{w}^{\star\mathsf{T}}\hat{\mathbf{w}} \\ \mathbf{w}^{\star\mathsf{T}}\hat{\mathbf{w}} & \hat{\mathbf{w}}^\mathsf{T}\hat{\mathbf{w}} \end{bmatrix} \equiv \begin{bmatrix} \sigma_{\mathbf{w}^\star} & \sigma_{\mathbf{w}^\star\hat{\mathbf{w}}} \\ \sigma_{\mathbf{w}^\star\hat{\mathbf{w}}} & \sigma_{\hat{\mathbf{w}}} \end{bmatrix} . \tag{II.4}$$

The asymptotic generalization error depends only on the covariance matrix $\sigma$ and as the samples are i.i.d it reads

$$e_{\mathrm{g}}(\alpha) = \lim_{d\to\infty} \mathbb{E}_{y,\mathbf{x}} \mathbb{1}\left[y \neq \hat{y}\left(\hat{\mathbf{w}}(\alpha);\mathbf{x}\right)\right] = 1 - \mathbb{P}[y = \hat{y}\left(\hat{\mathbf{w}}(\alpha);\mathbf{x}\right)] = 1 - 2 \int_{(\mathbb{R}^+)^2} d\mathbf{x} \mathcal{N}_{\mathbf{x}}\left(\mathbf{0}, \sigma\right)$$

$$= 1 - \left(\frac{1}{2} + \frac{1}{\pi}\operatorname{atan}\left(\sqrt{\frac{\sigma_{\mathbf{w}^\star\hat{\mathbf{w}}}^2}{\sigma_{\mathbf{w}^\star}\sigma_{\hat{\mathbf{w}}} - \sigma_{\mathbf{w}^\star\hat{\mathbf{w}}}^2}}\right)\right) = \frac{1}{\pi}\operatorname{acos}\left(\frac{\sigma_{\mathbf{w}^\star\hat{\mathbf{w}}}}{\sqrt{\sigma_{\mathbf{w}^\star}\sigma_{\hat{\mathbf{w}}}}}\right) ,$$
$$\tag{II.5}$$

where we used the fact that $\operatorname{atan}(x) = \frac{\pi}{2} - \frac{1}{2}\operatorname{acos}\left(\frac{x^2-1}{1+x^2}\right)$ and $\frac{1}{2}\operatorname{acos}(2x^2 - 1) = \operatorname{acos}(x)$. Finally

$$e_{\mathrm{g}}(\alpha) \equiv \lim_{d\to\infty} \mathbb{E}_{y,\mathbf{x}} \mathbb{1}\left[y \neq \hat{y}\left(\hat{\mathbf{w}}(\alpha);\mathbf{x}\right)\right] = \frac{1}{\pi}\operatorname{acos}\left(\frac{\sigma_{\mathbf{w}^\star\hat{\mathbf{w}}}}{\sqrt{\rho_{\mathbf{w}^\star}\sigma_{\hat{\mathbf{w}}}}}\right) , \tag{II.6}$$

with

$$\sigma_{\mathbf{w}^\star\hat{\mathbf{w}}} \equiv \lim_{d\to\infty} \mathbb{E}_{\mathbf{w}^\star,\mathrm{X}} \frac{1}{d}\hat{\mathbf{w}}^\mathsf{T}\mathbf{w}^\star , \qquad \rho_{\mathbf{w}^\star} \equiv \lim_{d\to\infty} \mathbb{E}_{\mathbf{w}^\star} \frac{1}{d}\|\mathbf{w}^\star\|_2^2 , \qquad \sigma_{\hat{\mathbf{w}}} \equiv \lim_{d\to\infty} \mathbb{E}_{\mathbf{w}^\star,\mathrm{X}} \frac{1}{d}\|\hat{\mathbf{w}}\|_2^2 .$$

### II.2 Bayes-optimal generalization error

The Bayes-optimal generalization error for classification is equal to eq. (II.6) where the Bayes estimator $\hat{\mathbf{w}}$ is the average over the posterior distribution eq. (I.2) denoted $\langle.\rangle$, knowing the teacher prior $P_{\mathbf{w}^\star}$ and channel $P_{\mathrm{out}^\star}$ distributions: $\hat{\mathbf{w}} = \langle\mathbf{w}\rangle_{\mathbf{w}}$. Hence the parameters $\sigma_{\hat{\mathbf{w}}}$ and $\sigma_{\mathbf{w}^\star\hat{\mathbf{w}}}$ read in the Bayes-optimal case

$$\sigma_{\hat{\mathbf{w}}} \equiv \lim_{d\to\infty} \mathbb{E}_{\mathbf{w}^\star,\mathrm{X}} \frac{1}{d}\|\hat{\mathbf{w}}\|_2^2 = \lim_{d\to\infty} \mathbb{E}_{\mathbf{w}^\star,\mathrm{X}} \frac{1}{d}\|\langle\mathbf{w}\rangle_{\mathbf{w}}\|_2^2 \equiv q_{\mathrm{b}} ,$$

$$\sigma_{\mathbf{w}^\star\hat{\mathbf{w}}} \equiv \lim_{d\to\infty} \mathbb{E}_{\mathbf{w}^\star,\mathrm{X}} \frac{1}{d}\hat{\mathbf{w}}^\mathsf{T}\mathbf{w}^\star = \lim_{d\to\infty} \mathbb{E}_{\mathbf{w}^\star,\mathrm{X}} \frac{1}{d}\langle\mathbf{w}\rangle_{\mathbf{w}}^\mathsf{T}\mathbf{w}^\star \equiv m_{\mathrm{b}} .$$

Using Nishimori identity [3], we easily obtain $m_{\mathrm{b}} = q_{\mathrm{b}}$ which is solution of eq. (13). Therefore the generalization error simplifies

$$e_{\mathrm{g}}^{\mathrm{bayes}}(\alpha) = \frac{1}{\pi}\operatorname{acos}\left(\sqrt{\eta_{\mathrm{b}}}\right) , \quad \text{with} \quad \eta_{\mathrm{b}} = \frac{q_{\mathrm{b}}}{\rho_{\mathbf{w}^\star}} . \tag{II.7}$$

### II.3 ERM generalization error

The generalization error of the ERM estimator is given again by eq. (II.6) with parameters

$$\sigma_{\hat{\mathbf{w}}} \equiv \lim_{d \to \infty} \mathbb{E}_{\mathbf{w}^\star, \mathbf{X}} \frac{1}{d} \|\hat{\mathbf{w}}\|_2^2 = \lim_{d \to \infty} \mathbb{E}_{\mathbf{w}^\star, \mathbf{X}} \frac{1}{d} \|\hat{\mathbf{w}}^{\mathrm{erm}}\|_2^2 \equiv q \,,$$

$$\sigma_{\mathbf{w}^\star \hat{\mathbf{w}}} \equiv \lim_{d \to \infty} \mathbb{E}_{\mathbf{w}^\star, \mathbf{X}} \frac{1}{d} \hat{\mathbf{w}}^\mathsf{T} \mathbf{w}^\star = \lim_{d \to \infty} \mathbb{E}_{\mathbf{w}^\star, \mathbf{X}} \frac{1}{d} (\hat{\mathbf{w}}^{\mathrm{erm}})^\mathsf{T} \mathbf{w}^\star \equiv m \,.$$

where the parameters $m, q$ are the asymptotic ERM overlaps solutions of eq. (11) and that finally lead to the ERM generalization error for classification:

$$e_{\mathrm{g}}^{\mathrm{erm}}(\alpha) = \frac{1}{\pi} \mathrm{acos}\left(\sqrt{\eta}\right) \,, \qquad\qquad \text{with } \eta \equiv \frac{m^2}{\rho_{\mathbf{w}^\star} q} \,. \qquad\qquad \text{(II.8)}$$

# III Proofs of the ERM fixed points

## III.1 Gordon's result and proofs

We consider in this section that the data have been generated by a teacher (I.1) with Gaussian weights

$$\mathbf{w}^\star \sim P_{\mathbf{w}^\star}(\mathbf{w}^\star) = \mathcal{N}_{\mathbf{w}^\star}\left(\mathbf{0}, \rho_{\mathbf{w}^\star}\mathrm{I}_d\right) \quad \text{with} \quad \rho_{\mathbf{w}^\star} \equiv \mathbb{E}\left[(w^\star)^2\right] . \tag{III.1}$$

### III.1.1 For real outputs - Regression with $\ell_2$ regularization

In what follows, we prove a theorem that characterizes the asymptotic performance of empirical risk minimization

$$\hat{\mathbf{w}}_{\mathrm{erm}} = \mathrm{argmin}_{\mathbf{w}} \sum_{i=1}^{n} l\left(y_i, \tfrac{1}{\sqrt{d}}\mathbf{x}_i^\mathsf{T}\mathbf{w}\right) + \frac{\lambda\|\mathbf{w}\|^2}{2}, \tag{III.2}$$

where $\{y_i\}_{1 \leq i \leq n}$ are general real-valued outputs (that are not necessarily binary), $l(y, z)$ is a loss function that is convex with respect to $z$, and $\lambda > 0$ is the strength of the $\ell_2$ regularization. Note that this setting is more general than the one considered in Thm. 2.2 in the main text, which focuses on binary outputs and loss functions in the form of $l(y, z) = \ell(yz)$ for some convex function $\ell(\cdot)$.

**Theorem III.1** (Regression with $\ell_2$ regularization). *As $n, d \to \infty$ with $n/d = \alpha = \Theta(1)$, the overlap parameters $m, q$ concentrate to*

$$m \underset{d \to \infty}{\longrightarrow} \sqrt{\rho_{\mathbf{w}^\star}}\mu^*, \qquad\qquad q \underset{d \to \infty}{\longrightarrow} (\mu^*)^2 + (\delta^*)^2, \tag{III.3}$$

*where the parameters $\mu^*, \delta^*$ are the solutions of*

$$(\mu^*, \delta^*) = \arg\min_{\mu, \delta \geq 0} \sup_{\tau > 0}\left\{\frac{\lambda(\mu^2 + \delta^2)}{2} - \frac{\delta^2}{2\tau} + \alpha\mathbb{E}_{g,s}\mathcal{M}_\tau[l(\varphi_{\mathrm{out}^\star}(\sqrt{\rho_{\mathbf{w}^\star}}s), .)](\mu s + \delta g)\right\}. \tag{III.4}$$

*Here, $\mathcal{M}_\tau[l(, .)](x)$ is the Moreau-Yosida regularization defined in (I.11), and $g, s$ are two i.i.d standard normal random variables.*

*Proof.* Since the teacher weight vector $\mathbf{w}^\star$ is independent of the input data matrix X, we can assume without loss of generality that

$$\mathbf{w}^\star = \sqrt{d}\rho_d\mathbf{e}_1,$$

where $\mathbf{e}_1$ is the first natural basis vector of $\mathbb{R}^d$, and $\rho_d = \|\mathbf{w}^\star\|/\sqrt{d}$. As $d \to \infty$, $\rho_d \to \sqrt{\rho_{\mathbf{w}^\star}}$. Accordingly, it will be convenient to split the data matrix into two parts:

$$\mathrm{X} = \begin{bmatrix}\mathbf{s} & \mathrm{B}\end{bmatrix}, \tag{III.5}$$

where $\mathbf{s} \in \mathbb{R}^{n \times 1}$ and $\mathrm{B} \in \mathbb{R}^{n \times (d-1)}$ are two sub-matrices of i.i.d standard normal entries. The weight vector $\mathbf{w}$ in (III.2) can also be written as $\mathbf{w} = [\sqrt{d}\mu, \mathbf{v}^\mathsf{T}]^\mathsf{T}$, where $\mu \in \mathbb{R}$ denotes the projection of $\mathbf{w}$ onto the direction spanned by the teacher weight vector $\mathbf{w}^\star$, and $\mathbf{v} \in \mathbb{R}^{d-1}$ is the projection of $\mathbf{w}$ onto the complement subspace. These representations serve to simplify the notations in our subsequent derivations. For example, we can now write the output as

$$y_i = \varphi_{\mathrm{out}^\star}(\rho_d s_i), \tag{III.6}$$

where $s_i$ is the $i$th entry of the Gaussian vector $\mathbf{s}$ in (III.5).

Let $\Phi_d$ denote the cost of the ERM in (III.2), normalized by $d$. Using our new representations introduced above, we have

$$\Phi_d = \min_{\mu, \mathbf{v}} \frac{1}{d}\sum_{i=1}^{n} l\left(y_i, \mu s_i + \tfrac{1}{\sqrt{d}}\mathbf{b}_i^\mathsf{T}\mathbf{v}\right) + \frac{\lambda(d\mu^2 + \|\mathbf{v}\|^2)}{2d}, \tag{III.7}$$

where $\mathbf{b}_i^\mathsf{T}$ denotes the $i$th row of B. Since the loss function $l(y_i, z)$ is convex with respect to $z$, we can rewrite it as

$$l(y_i, z) = \sup_q\{qz - l^*(y_i, q)\}, \tag{III.8}$$

where $l^*(y_i, q) = \sup_z\{qz - l(y_i, z)\}$ is its convex conjugate. Substituting (III.8) into (III.7), we have

$$\Phi_d = \min_{\mu,\mathbf{v}} \sup_{\mathbf{q}} \left\{ \frac{\mu\mathbf{q}^\mathsf{T}\mathbf{s}}{d} + \frac{1}{d^{3/2}}\mathbf{q}^\mathsf{T}\mathbf{B}\mathbf{v} - \frac{1}{d}\sum_{i=1}^n l^*(y_i, q_i) + \frac{\lambda\left(d\mu^2 + \|\mathbf{v}\|^2\right)}{2d} \right\}. \qquad \text{(III.9)}$$

Now consider a new optimization problem

$$\widetilde{\Phi}_d = \min_{\mu,\mathbf{v}} \sup_{\mathbf{q}} \left\{ \frac{\mu\mathbf{q}^\mathsf{T}\mathbf{s}}{d} + \frac{\|\mathbf{q}\|}{\sqrt{d}}\frac{\mathbf{h}^\mathsf{T}\mathbf{v}}{d} + \frac{\|\mathbf{v}\|}{\sqrt{d}}\frac{\mathbf{g}^\mathsf{T}\mathbf{q}}{d} - \frac{1}{d}\sum_{i=1}^n l^*(y_i, q_i) + \frac{\lambda\left(d\mu^2 + \|\mathbf{v}\|^2\right)}{2d} \right\}, \quad \text{(III.10)}$$

where $h \sim \mathcal{N}\left(\mathbf{0}, \mathrm{I}_{d-1}\right)$ and $g \sim \mathcal{N}\left(\mathbf{0}, \mathrm{I}_n\right)$ are two independent standard normal vectors. It follows from Gordon's minimax comparison inequality (see, *e.g.*, [4]) that

$$\mathbb{P}(|\Phi_d - c| \geq \epsilon) \leq 2\mathbb{P}\left(\left|\widetilde{\Phi}_d - c\right| \geq \epsilon\right) \qquad \text{(III.11)}$$

for any constants $c$ and $\epsilon > 0$. This implies that $\widetilde{\Phi}_d$ serves as a surrogate of $\Phi_d$. Specifically, if $\widetilde{\Phi}_d$ concentrates around some deterministic limit $c$ as $d \to \infty$, so does $\Phi_d$. In what follows, we proceed to solve the surrogate problem in (III.10). First, let $\delta = \|\mathbf{v}\|/\sqrt{d}$. It is easy to see that (III.10) can be simplified as

$$\widetilde{\Phi}_d = \min_{\mu,\delta\geq 0} \sup_{\mathbf{q}} \left\{ \frac{\mathbf{q}^\mathsf{T}(\mu\mathbf{s} + \delta\mathbf{g})}{d} - \delta\frac{\|\mathbf{q}\|}{\sqrt{d}}\frac{\|\mathbf{h}\|}{\sqrt{d}} - \frac{1}{d}\sum_{i=1}^n l^*(y_i, q_i) + \frac{\lambda(\mu^2 + \delta^2)}{2} \right\}$$

$$\overset{(a)}{=} \min_{\mu,\delta\geq 0} \sup_{\tau>0} \sup_{\mathbf{q}} \left\{ -\frac{\tau\|\mathbf{q}\|^2}{2d} - \frac{\delta^2\|\mathbf{h}\|^2}{2\tau d} + \frac{\mathbf{q}^\mathsf{T}(\mu\mathbf{s} + \delta\mathbf{g})}{d} - \frac{1}{d}\sum_{i=1}^n l^*(y_i, q_i) + \frac{\lambda(\mu^2 + \delta^2)}{2} \right\}$$

$$= \min_{\mu,\delta\geq 0} \sup_{\tau>0} \left\{ \frac{\lambda(\mu^2 + \delta^2)}{2} - \frac{\delta^2\|\mathbf{h}\|^2}{2\tau d} - \frac{\alpha}{n}\inf_{\mathbf{q}}\left[\frac{\tau\|\mathbf{q}\|^2}{2} - \mathbf{q}^\mathsf{T}(\mu\mathbf{s} + \delta\mathbf{g}) + \sum_{i=1}^n l^*(y_i, q_i)\right] \right\}$$

$$\overset{(b)}{=} \min_{\mu,\delta\geq 0} \sup_{\tau>0} \left\{ \frac{\lambda(\mu^2 + \delta^2)}{2} - \frac{\delta^2\|\mathbf{h}\|^2}{2\tau d} - \frac{\alpha}{n}\sum_{i=1}^n \mathcal{M}_\tau[l(y_i, .)](\mu s_i + \delta g_i) \right\}.$$

In $(a)$, we have introduced an auxiliary variable $\tau$ to rewrite $-\delta\frac{\|\mathbf{q}\|}{\sqrt{d}}\frac{\|\mathbf{h}\|}{\sqrt{d}}$ as

$$-\delta\frac{\|\mathbf{q}\|}{\sqrt{d}}\frac{\|\mathbf{h}\|}{\sqrt{d}} = \sup_{\tau>0}\left\{ -\frac{\tau\|\mathbf{q}\|^2}{2d} - \frac{\delta^2\|\mathbf{h}\|^2}{2\tau d} \right\},$$

and to get $(b)$, we use the identity

$$\inf_q\left\{ \frac{\tau}{2}q^2 - qz + \ell^*(q) \right\} = -\inf_x\left\{ \frac{(z-x)^2}{2\tau} + \ell(x) \right\}$$

that holds for any $z$ and for any convex function $\ell(x)$ and its conjugate $\ell^*(q)$. As $d \to \infty$, standard concentration arguments give us $\frac{\|\mathbf{h}\|^2}{d} \to 1$ and $\frac{1}{n}\sum_{i=1}^n \mathcal{M}_\tau[l(y_i, .)](\mu s_i + \delta g_i) \to \mathbb{E}_{g,s}\mathcal{M}_\tau[l(y, .)](\mu s + \delta g)$ locally uniformly over $\tau, \mu$ and $\delta$. Using (III.11) and recalling (III.6), we can then conclude that the normalized cost of the ERM $\Phi_d$ converges to the optimal value of the deterministic optimization problem in (III.4). Finally, since $\lambda > 0$, one can show that the cost function of (III.4) has a unique global minima at $\mu^*$ and $\delta^*$. It follows that the empirical values of $(\mu, \delta)$ associated with the surrogate optimization problem (III.10) converge to their corresponding deterministic limits $(\mu^*, \delta^*)$. Finally, the convergence of $(\mu, \delta)$ associated with the original optimization problem (III.9) towards the same limits can be established by evoking standard arguments (see, *e.g.*, [5, Theorem 6.1, statement (iii)]). □

### III.1.2 For binary outputs - Classification with $\ell_2$ regularization

In what follows, we specialize the previous theorem to the case of binary classification, with a convex loss function in the form of $l(y, z) = \ell(yz)$ for some function $\ell(\cdot)$.

**Theorem III.2** (Thm. 2.2 in the main text. Gordon's min-max fixed point - Classification with $\ell_2$ regularization). *As $n, d \to \infty$ with $n/d = \alpha = \Theta(1)$, the overlap parameters $m, q$ concentrate to*

$$m \xrightarrow[d \to \infty]{} \sqrt{\rho_{w^\star}} \mu^\star , \qquad\qquad q \xrightarrow[d \to \infty]{} (\mu^\star)^2 + (\delta^\star)^2 , \qquad (\text{III.12})$$

*where parameters $\mu^\star, \delta^\star$ are solutions of*

$$(\mu^\star, \delta^\star) = \underset{\mu, \delta \geq 0}{\arg\min} \; \underset{\tau > 0}{\sup} \left\{ \frac{\lambda(\mu^2 + \delta^2)}{2} - \frac{\delta^2}{2\tau} + \alpha \mathbb{E}_{g,s} \mathcal{M}_\tau [\delta g + \mu s \varphi_{\text{out}^\star}(\sqrt{\rho_{w^\star}} s)] \right\}, \quad (\text{III.13})$$

*and $g, s$ are two i.i.d standard normal random variables. The solutions $(\mu^\star, \delta^\star, \tau^\star)$ of (III.13) can be reformulated as a set of fixed point equations*

$$\mu^\star = \frac{\alpha}{\lambda \tau^\star + \alpha} \mathbb{E}[s \cdot \varphi_{\text{out}^\star}(\sqrt{\rho_{w^\star}} s) \cdot \mathcal{P}_{\tau^\star}(\delta^\star g + \mu^\star s \varphi_{\text{out}^\star}(\sqrt{\rho_{w^\star}} s))] ,$$

$$\delta^\star = \frac{\alpha}{\lambda \tau^\star + \alpha - 1} \mathbb{E}[g \cdot \mathcal{P}_{\tau^\star}(\delta^\star g + \mu^\star s \varphi_{\text{out}^\star}(\sqrt{\rho_{w^\star}} s))] , \qquad (\text{III.14})$$

$$(\delta^\star)^2 = \alpha \mathbb{E}[(\delta^\star g + \mu^\star s \varphi_{\text{out}^\star}(\sqrt{\rho_{w^\star}} s) - \mathcal{P}_{\tau^\star}(\delta^\star g + \mu^\star s \varphi_{\text{out}^\star}(\sqrt{\rho_{w^\star}} s)))^2] ,$$

*where $\mathcal{M}_\tau$ and $\mathcal{P}_\tau$ denote the Moreau-Yosida regularization and the proximal map of a convex loss function $(y, z) \mapsto \ell(yz)$:*

$$\mathcal{M}_\tau(z) = \min_x \left\{ \ell(x) + \frac{(x - z)^2}{2\tau} \right\} , \qquad \mathcal{P}_\tau(z) = \arg\min_x \left\{ \ell(x) + \frac{(x - z)^2}{2\tau} \right\} .$$

*Proof.* We start by deriving (III.13) as a special case of (III.4). To that end, we note that

$$\mathcal{M}_\tau[l(y, .)](z) = \min_x \left\{ l(y; x) + \frac{(x - z)^2}{2\tau} \right\}$$

$$= \min_x \left\{ \ell(yx) + \frac{(x - z)^2}{2\tau} \right\}$$

$$= \min_x \left\{ \ell(x) + \frac{(x - yz)^2}{2\tau} \right\} = \mathcal{M}_\tau(yz),$$

where to reach the last equality we have used the fact that $y \in \{\pm 1\}$. Substituting this special form into (III.4) and recalling (III.6), we reach (III.13).

Finally, to obtain the fixed point equations (III.14), we simply take the partial derivatives of the cost function in (III.13) with respect to $\mu, \delta, \tau$, and use the following well-known calculus rules for the Moreau-Yosida regularization [6]:

$$\frac{\partial \mathcal{M}_\tau(z)}{\partial z} = \frac{z - \mathcal{P}_\tau(z)}{\tau} ,$$

$$\frac{\partial \mathcal{M}_\tau(z)}{\partial \tau} = -\frac{(z - \mathcal{P}_\tau(z))^2}{2\tau^2} .$$

$\square$

### III.2 Replica's formulation

The replica computation presented in Sec. IV boils down to the characterization of the overlaps $m, q$ in the high-dimensional limit $n, d \to \infty$ with $\alpha = \frac{n}{d} = \Theta(1)$, given by the solution of a set of, in the most general case, six fixed point equations over $m, q, Q, \hat{m}, \hat{q}, \hat{Q}$. Introducing the natural variables $\Sigma \equiv Q - q$, $\hat{\Sigma} \equiv \hat{Q} + \hat{q}$, $\eta \equiv \frac{m^2}{\rho_{w^\star} q}$ and $\hat{\eta} \equiv \frac{\hat{m}^2}{\hat{q}}$, the set of fixed point equations for arbitrary

$P_{\mathrm{w}^\star}, P_{\mathrm{out}^\star}$, convex loss $l(y,z)$ and regularizer $r(w)$, is finally given by

$$m = \mathbb{E}_\xi \left[ \mathcal{Z}_{\mathrm{w}^\star}\left(\sqrt{\hat{\eta}}\xi, \hat{\eta}\right) f_{\mathrm{w}^\star}\left(\sqrt{\hat{\eta}}\xi, \hat{\eta}\right) f_{\mathrm{w}}\left(\hat{q}^{1/2}\xi, \hat{\Sigma}\right) \right],$$

$$q = \mathbb{E}_\xi \left[ \mathcal{Z}_{\mathrm{w}^\star}\left(\sqrt{\hat{\eta}}\xi, \hat{\eta}\right) f_{\mathrm{w}}\left(\hat{q}^{1/2}\xi, \hat{\Sigma}\right)^2 \right],$$

$$\Sigma = \mathbb{E}_\xi \left[ \mathcal{Z}_{\mathrm{w}^\star}\left(\sqrt{\hat{\eta}}\xi, \hat{\eta}\right) \partial_\gamma f_{\mathrm{w}}\left(\hat{q}^{1/2}\xi, \hat{\Sigma}\right) \right],$$

$$\hat{m} = \alpha\mathbb{E}_{y,\xi} \left[ \mathcal{Z}_{\mathrm{out}^\star}(.) \cdot f_{\mathrm{out}^\star}\left(y, \sqrt{\rho_{\mathrm{w}^\star}\eta}\xi, \rho_{\mathrm{w}^\star}(1-\eta)\right) f_{\mathrm{out}}\left(y, q^{1/2}\xi, \Sigma\right) \right],$$

$$\hat{q} = \alpha\mathbb{E}_{y,\xi} \left[ \mathcal{Z}_{\mathrm{out}^\star}\left(y, \sqrt{\rho_{\mathrm{w}^\star}\eta}\xi, \rho_{\mathrm{w}^\star}(1-\eta)\right) f_{\mathrm{out}}\left(y, q^{1/2}\xi, \Sigma\right)^2 \right],$$

$$\hat{\Sigma} = -\alpha\mathbb{E}_{y,\xi} \left[ \mathcal{Z}_{\mathrm{out}^\star}\left(y, \sqrt{\rho_{\mathrm{w}^\star}\eta}\xi, \rho_{\mathrm{w}^\star}(1-\eta)\right) \partial_\omega f_{\mathrm{out}}\left(y, q^{1/2}\xi, \Sigma\right) \right].$$

(III.15)

The above equations depend on the Bayes-optimal partition functions $\mathcal{Z}_{\mathrm{w}^\star}, \mathcal{Z}_{\mathrm{out}^\star}$ defined in eq. (I.7), the updates $f_{\mathrm{w}^\star}, f_{\mathrm{out}^\star}$ in eq. (I.8) and the ERM updates $f_{\mathrm{w}}, f_{\mathrm{out}}$ eq. (I.12).

### III.3 Equivalence Gordon-Replica's formulation - $\ell_2$ regularization and Gaussian weights

#### III.3.1 Replica's formulation for $\ell_2$ regularization

The proximal for the $\ell_2$ penalty with strength $\lambda$ can be computed explicitly in eq. (I.30) and the corresponding denoising function is simply given by $f_{\mathrm{w}}^{\ell_2,\lambda}(\gamma, \Lambda) = \frac{\gamma}{\lambda+\Lambda}$. Therefore, for a Gaussian teacher (III.1) already considered in Thm. (III.14) with second moment $\rho_{\mathrm{w}^\star}$, using the denoising function (I.24), the fixed point equations over $m, q, \Sigma$ can be computed analytically and lead to

$$m = \frac{\rho_{\mathrm{w}^\star}\hat{m}}{\lambda + \hat{\Sigma}}, \qquad q = \frac{\rho_{\mathrm{w}^\star}\hat{m}^2 + \hat{q}}{(\lambda + \hat{\Sigma})^2}, \qquad \Sigma = \frac{1}{\lambda + \hat{\Sigma}}.$$

(III.16)

Hence, removing the *hat* variables in eqs. (III.15), the set of fixed point equations can be rewritten in a more compact way leading to the Corollary. 2.3 that we recall here:

**Corollary III.3** (Corollary. 2.3 in the main text. Equivalence Gordon-Replicas). *The set of fixed point equations* (III.14) *in Thm. III.2 that govern the asymptotic behaviour of the overlaps $m$ and $q$ is equivalent to the following set of equations, obtained from the heuristic replica computation:*

$$m = \alpha\Sigma\rho_{\mathrm{w}^\star} \cdot \mathbb{E}_{y,\xi} \left[ \mathcal{Z}_{\mathrm{out}^\star}(.) \cdot f_{\mathrm{out}^\star}\left(y, \sqrt{\rho_{\mathrm{w}^\star}\eta}\xi, \rho_{\mathrm{w}^\star}(1-\eta)\right) \cdot f_{\mathrm{out}}\left(y, q^{1/2}\xi, \Sigma\right) \right]$$

$$q = m^2/\rho_{\mathrm{w}^\star} + \alpha\Sigma^2 \cdot \mathbb{E}_{y,\xi} \left[ \mathcal{Z}_{\mathrm{out}^\star}\left(y, \sqrt{\rho_{\mathrm{w}^\star}\eta}\xi, \rho_{\mathrm{w}^\star}(1-\eta)\right) \cdot f_{\mathrm{out}}\left(y, q^{1/2}\xi, \Sigma\right)^2 \right]$$

(III.17)

$$\Sigma = \left(\lambda - \alpha \cdot \mathbb{E}_{y,\xi} \left[ \mathcal{Z}_{\mathrm{out}^\star}\left(y, \sqrt{\rho_{\mathrm{w}^\star}\eta}\xi, \rho_{\mathrm{w}^\star}(1-\eta)\right) \cdot \partial_\omega f_{\mathrm{out}}\left(y, q^{1/2}\xi, \Sigma\right) \right]\right)^{-1}$$

*with $\eta \equiv \frac{m^2}{\rho_{\mathrm{w}^\star}q}$, $\xi \sim \mathcal{N}(0,1)$ and $\mathbb{E}_y$ the continuous or discrete sum over all possible values $y$ according to $P_{\mathrm{out}^\star}$.*

*Proof of Corollary. III.3(Corollary. 2.3).* For the sake of clarity, we use the abusive notation $\mathcal{P}_V(y,\omega) = \mathcal{P}_V[l(y,.)](\omega)$, and we remove the $*$.

**Dictionary** We first map the Gordon's parameters $(\mu, \delta, \tau)$ in eq. (III.14) to $(m, q, \Sigma)$ in eq. (III.17):

$$\sqrt{\rho_{\mathrm{w}^\star}}\mu \leftrightarrow m, \qquad\qquad \mu^2 + \delta^2 \leftrightarrow q, \qquad\qquad \tau \leftrightarrow \Sigma.$$

so that

$$\eta = \frac{m^2}{\rho_{\mathrm{w}^\star}q} = \frac{\mu^2}{\mu^2 + \delta^2}, \qquad\qquad 1 - \eta = \frac{\delta^2}{\mu^2 + \delta^2}.$$

From eq. (I.7), we can rewrite the channel partition function $\mathcal{Z}_{\mathrm{out}^\star}$ and its derivative

$$\mathcal{Z}_{\mathrm{out}^\star}(y, \omega, V) = \mathbb{E}_z \left[ P_{\mathrm{out}^\star}\left(y | \sqrt{V}z + \omega\right) \right],$$

$$\partial_\omega \mathcal{Z}_{\mathrm{out}^\star}(y, \omega, V) = \frac{1}{\sqrt{V}}\mathbb{E}_z \left[ z P_{\mathrm{out}^\star}\left(y | \sqrt{V}z + \omega\right) \right],$$

(III.18)

where $z$ denotes a standard normal random variable.

**Equation over $m$**  Let us start with the equation over $m$ in eq. (III.17):

$$m = \Sigma \alpha \rho_{\mathrm{w}^\star} \mathbb{E}_{y,\xi} \left[ \mathcal{Z}_{\mathrm{out}^\star} \left( y, \sqrt{\rho_{\mathrm{w}^\star}} \eta \xi, \rho_{\mathrm{w}^\star} (1-\eta) \right) f_{\mathrm{out}^\star} \left( y, \sqrt{\rho_{\mathrm{w}^\star}} \eta \xi, \rho_{\mathrm{w}^\star} (1-\eta) \right) \right.$$

$$\left. \times f_{\mathrm{out}} \left( y, q^{1/2}\xi, \Sigma \right) \right]$$

$$= \Sigma \alpha \frac{\sqrt{\rho_{\mathrm{w}^\star}}}{\sqrt{1-\eta}} \mathbb{E}_{y,\xi,z} \left[ z P_{\mathrm{out}^\star} \left( y | \sqrt{\rho_{\mathrm{w}^\star}} \left( \sqrt{1-\eta} z + \sqrt{\eta} \xi \right) \right) \Sigma^{-1} (\mathcal{P}_{\Sigma} (y, \sqrt{q}\xi) - \sqrt{q}\xi) \right]$$
$$\text{(Using eq. (III.18))}$$

$$\Leftrightarrow \mu = \frac{\sqrt{\mu^2 + \delta^2}}{\delta} \alpha \mathbb{E}_{y,\xi,z} \left[ z P_{\mathrm{out}^\star} \left[ y | \sqrt{\rho_{\mathrm{w}^\star}} \frac{\delta z + \mu \xi}{\sqrt{\mu^2 + \delta^2}} \right] \left( \mathcal{P}_{\tau} \left( y, \sqrt{\mu^2 + \delta^2}\xi \right) - \sqrt{\mu^2 + \delta^2}\xi \right) \right]$$
$$\text{(Dictionary)}$$

$$= \frac{\sqrt{\mu^2 + \delta^2}}{\delta} \alpha \mathbb{E}_{\xi,z} \left[ z \left( \mathcal{P}_{\tau} \left( \varphi_{\mathrm{out}^\star} \left( \sqrt{\rho_{\mathrm{w}^\star}} \frac{\delta z + \mu \xi}{\sqrt{\mu^2 + \delta^2}} \right), \sqrt{\mu^2 + \delta^2}\xi \right) - \sqrt{\mu^2 + \delta^2}\xi \right) \right]$$
$$\text{(Integration over } y\text{)}$$

$$= \alpha \mathbb{E}_{s,g} \left[ \left( s - \frac{\mu}{\delta} g \right) \left( \mathcal{P}_{\tau} \left( \varphi_{\mathrm{out}^\star} \left( \sqrt{\rho_{\mathrm{w}^\star}} s \right), \delta g + \mu s \right) - (\delta g + \mu s) \right) \right]$$
$$\text{(Change of variables } (\xi, z) \to (g, s)\text{)}$$

$$= \alpha \mathbb{E}_{s,g} \left[ \left( s - \frac{\mu}{\delta} g \right) \left( \mathcal{P}_{\tau} \left( \varphi_{\mathrm{out}^\star} \left( \sqrt{\rho_{\mathrm{w}^\star}} s \right), \delta g + \mu s \right) \right) \right] \qquad \text{(Gaussian integrations)}$$

$$\Leftrightarrow \mu = \frac{\alpha \mathbb{E}_{s,g} \left[ s \cdot \mathcal{P}_{\tau} \left( \varphi_{\mathrm{out}^\star} \left( \sqrt{\rho_{\mathrm{w}^\star}} s \right), \delta g + \mu s \right) \right]}{1 + \frac{\alpha}{\delta} \mathbb{E}_{s,g} \left[ g \cdot \mathcal{P}_{\tau} \left( \varphi_{\mathrm{out}^\star} \left( \sqrt{\rho_{\mathrm{w}^\star}} s \right), \delta g + \mu s \right) \right]}$$

$$= \frac{\alpha}{\lambda \tau + \alpha} \mathbb{E}_{s,g} \left[ s \cdot \varphi_{\mathrm{out}^\star} \left( \sqrt{\rho_{\mathrm{w}^\star}} s \right) \left( \mathcal{P}_{\tau} (\delta g + \mu s) \varphi_{\mathrm{out}^\star} \left( \sqrt{\rho_{\mathrm{w}^\star}} s \right) \right) \right],$$
$$\text{(Second fixed point equation)}$$

where we used the fact that $P_{\mathrm{out}^\star}(y|z) = \delta(y - \varphi_{\mathrm{out}^\star}(z))$, the change of variables

$$\begin{cases} s = \frac{\mu \xi + \delta z}{\sqrt{\mu^2 + \delta^2}} \\ g = \frac{\delta \xi - \mu z}{\sqrt{\mu^2 + \delta^2}} \end{cases} \quad \Leftrightarrow \quad \begin{cases} \xi = \frac{\delta g + \mu s}{\sqrt{\mu^2 + \delta^2}} \\ z = \frac{\delta s - \mu g}{\sqrt{\mu^2 + \delta^2}} \end{cases}, \tag{III.19}$$

and finally in the last equality the definition of the second fixed point equation in eqs. (III.14):

$$\delta = \alpha \frac{\mathbb{E}_{s,g} \left[ g \cdot \mathcal{P}_{\tau} \left( \varphi_{\mathrm{out}^\star} \left( \sqrt{\rho_{\mathrm{w}^\star}} s \right), \delta g + \mu s \right) \right]}{\lambda \tau + \alpha - 1}. \tag{III.20}$$

**Equation over $q$**  Let us now compute the equation over $q$ in eq. (III.17):

$$q - m^2/\rho_{\mathrm{w}^\star} = \Sigma^2 \alpha \mathbb{E}_{y,\xi} \left[ \mathcal{Z}_{\mathrm{out}^\star} \left( y, \sqrt{\rho_{\mathrm{w}^\star}} \eta \xi, \rho_{\mathrm{w}^\star} (1-\eta) \right) f_{\mathrm{out}} \left( y, q^{1/2}\xi, \Sigma \right)^2 \right]$$

$$= \Sigma^2 \alpha \mathbb{E}_{y,\xi,z} \left[ P_{\mathrm{out}^\star} \left( y | \sqrt{\rho_{\mathrm{w}^\star}} \left( \sqrt{1-\eta} z + \sqrt{\eta} \xi \right) \right) \frac{1}{\Sigma^2} (p_{\Sigma} (y, \sqrt{q}\xi) - \sqrt{q}\xi)^2 \right]$$
$$\text{(Using eq. (III.18))}$$

$$\Leftrightarrow \delta^2 = \alpha \mathbb{E}_{y,\xi,z} \left[ P_{\mathrm{out}^\star} \left( y | \sqrt{\rho_{\mathrm{w}^\star}} \frac{\delta z + \mu \xi}{\sqrt{\mu^2 + \delta^2}} \right) \left( p_{\tau} \left( y, \sqrt{\mu^2 + \delta^2}\xi \right) - \sqrt{\mu^2 + \delta^2}\xi \right)^2 \right]$$
$$\text{(Dictionary)}$$

$$= \alpha \mathbb{E}_{\xi,z} \left[ \left( p_{\tau} \left( \varphi_{\mathrm{out}^\star} \left( \sqrt{\rho_{\mathrm{w}^\star}} \frac{\delta z + \mu \xi}{\sqrt{\mu^2 + \delta^2}} \right), \sqrt{\mu^2 + \delta^2}\xi \right) - \sqrt{\mu^2 + \delta^2}\xi \right)^2 \right]$$
$$\text{(Integration over } y\text{)}$$

$$= \alpha \mathbb{E}_{g,s} \left[ \left( p_{\tau} \left( \varphi_{\mathrm{out}^\star} \left( \sqrt{\rho_{\mathrm{w}^\star}} s \right), \delta g + \mu s \right) - (\delta g + \mu s) \right)^2 \right]$$
$$\text{(Change of variables } (\xi, z) \to (g, s)\text{)}$$

**Equation over $\Sigma$**   Let us conclude with the equation over $\Sigma$ in eq. (III.17) that we encountered in eq. (III.20). Let us first compute

$$\alpha \mathbb{E}_{y,\xi} \left[ \mathcal{Z}_{\text{out}^\star} \left( y, \sqrt{\rho_{\text{w}^\star}} \eta \xi, \rho_{\text{w}^\star} (1 - \eta) \right) \partial_\omega f_{\text{out}} \left( y, q^{1/2} \xi, \Sigma \right) \right]$$

$$= \alpha \mathbb{E}_{y,\xi,z} \left[ P_{\text{out}^\star} \left( y | \sqrt{\rho_{\text{w}^\star}} \left( \sqrt{1 - \eta} z + \sqrt{\eta} \xi \right) \right) \frac{1}{\Sigma} \left( \partial_\omega p_\Sigma \left( y, \sqrt{q} \xi \right) - 1 \right) \right] \quad \text{(Using eq. (III.18))}$$

$$= \frac{\alpha}{\tau} \mathbb{E}_{y,\xi,z} \left[ P_{\text{out}^\star} \left( y | \sqrt{\rho_{\text{w}^\star}} \frac{\delta z + \mu \xi}{\sqrt{\mu^2 + \delta^2}} \right) \left( \partial_\omega \mathcal{P}_\tau \left( y, \sqrt{\mu^2 + \delta^2} \xi \right) - 1 \right) \right] \quad \text{(Dictionary)}$$

$$= \frac{\alpha}{\tau} \mathbb{E}_{\xi,z} \left[ \partial_\omega \mathcal{P}_\tau \left( \varphi_{\text{out}^\star} \left( \sqrt{\rho_{\text{w}^\star}} \frac{\delta z + \mu \xi}{\sqrt{\mu^2 + \delta^2}} \right), \sqrt{\mu^2 + \delta^2} \xi \right) \right] - \frac{\alpha}{\tau} \quad \text{(Integration over } y)$$

$$= \frac{1}{\tau} \alpha \left( \mathbb{E}_{g,s} \left[ \partial_\omega \mathcal{P}_\tau \left( \varphi_{\text{out}^\star} \left( \sqrt{\rho_{\text{w}^\star}} s \right), \delta g + \mu s \right) \right] - 1 \right) \quad \text{(Change of variables } (\xi, z) \to (g, s))$$

therefore, the last equation over $\Sigma$ in eq. (III.17) reads

$$\Sigma = \left( \lambda - \alpha \mathbb{E}_{y,\xi} \left[ \mathcal{Z}_{\text{out}^\star} \left( y, \sqrt{\rho_{\text{w}^\star}} \eta \xi, \rho_{\text{w}^\star} (1 - \eta) \right) \partial_\omega f_{\text{out}} \left( y, q^{1/2} \xi, \Sigma \right) \right] \right)^{-1}$$

$$\Leftrightarrow$$

$$\tau = \left( \lambda - \frac{1}{\tau} \alpha \left( \mathbb{E}_{g,s} \left[ \partial_\omega \mathcal{P}_\tau \left( \varphi_{\text{out}^\star} \left( \sqrt{\rho_{\text{w}^\star}} s \right), \delta g + \mu s \right) \right] - 1 \right) \right)^{-1}$$

$$\Leftrightarrow$$

$$\alpha \mathbb{E}_{g,s} \left[ \partial_\omega \mathcal{P}_\tau \left( \varphi_{\text{out}^\star} \left( \sqrt{\rho_{\text{w}^\star}} s \right), \delta g + \mu s \right) \right] = \tau \lambda + \alpha - 1 \,.$$

Noting that

$$\mathbb{E}_{g,s} \left[ \partial_\omega \mathcal{P}_\tau \left( \varphi_{\text{out}^\star} \left( \sqrt{\rho_{\text{w}^\star}} s \right), \delta g + \mu s \right) \right] = \frac{1}{\delta} \mathbb{E}_{g,s} \left[ d \partial_\omega \mathcal{P}_\tau \left( \varphi_{\text{out}^\star} \left( \sqrt{\rho_{\text{w}^\star}} s \right), \delta g + \mu s \right) \right]$$

$$= \frac{1}{\delta} \mathbb{E}_{g,s} \left[ \partial_g \mathcal{P}_\tau \left( \varphi_{\text{out}^\star} \left( \sqrt{\rho_{\text{w}^\star}} s \right), \delta g + \mu s \right) \right] = \frac{1}{\delta} \mathbb{E}_{g,s} \left[ g \mathcal{P}_\tau \left( \delta g + \mu s \varphi_{\text{out}^\star} \left( \sqrt{\rho_{\text{w}^\star}} s \right) \right) \right]$$

$$\text{(Stein's lemma)}$$

where we used the Stein's lemma in the last equality, we finally obtain

$$\alpha \mathbb{E}_{g,s} \left[ \partial_\omega \mathcal{P}_\tau \left( \varphi_{\text{out}^\star} \left( \sqrt{\rho_{\text{w}^\star}} s \right), \delta g + \mu s \right) \right] = \tau \lambda + \alpha - 1$$

$$\Leftrightarrow \delta = \frac{\alpha}{\tau \lambda + \alpha - 1} \mathbb{E}_{g,s} \left[ g \cdot \mathcal{P}_\tau \left( \varphi_{\text{out}^\star} \left( \sqrt{\rho_{\text{w}^\star}} s \right), \delta g + \mu s \right) \right] \,.$$

**Gauge transformation**   We still remain to prove that

$$\mathbb{E}_{s,g} \left[ g \cdot \mathcal{P}_\tau \left( \varphi_{\text{out}^\star} \left( \sqrt{\rho_{\text{w}^\star}} s \right), \delta g + \mu s \right) \right] = \mathbb{E}_{s,g} \left[ g \cdot \mathcal{P}_\tau \left( \delta g + \mu s \varphi_{\text{out}^\star} \left( \sqrt{\rho_{\text{w}^\star}} s \right) \right) \right]$$

$$\mathbb{E}_{s,g} \left[ s \cdot \mathcal{P}_\tau \left( \varphi_{\text{out}^\star} \left( \sqrt{\rho_{\text{w}^\star}} s \right), \delta g + \mu s \right) \right] = \mathbb{E}_{s,g} \left[ s \cdot \mathcal{P}_\tau \left( \delta g + \mu s \varphi_{\text{out}^\star} \left( \sqrt{\rho_{\text{w}^\star}} s \right) \right) \right]$$

$$\mathbb{E}_{g,s} \left[ \left( p_\tau \left( \varphi_{\text{out}^\star} \left( \sqrt{\rho_{\text{w}^\star}} s \right), \delta g + \mu s \right) - (\delta g + \mu s) \right)^2 \right] = \mathbb{E}_{g,s} \Big[$$

$$\left( (p_\tau - \mathbb{1}) \left( \delta g + \mu s \varphi_{\text{out}^\star} \left( \sqrt{\rho_{\text{w}^\star}} s \right) \right) \right)^2 \Big]$$

$$\text{(III.21)}$$

As $\varphi_{\text{out}^\star} \left( \sqrt{\rho_{\text{w}^\star}} s \right) = \pm 1$, we can transform $s \to s \varphi_{\text{out}^\star} \left( \sqrt{\rho_{\text{w}^\star}} s \right) = \tilde{s}$. It does not change the distribution of the random variable $\tilde{s}$ that is still a normal random variable. Finally denoting $\mathcal{P}_\tau \left( 1, \delta g + \mu s \varphi_{\text{out}^\star} \left( \sqrt{\rho_{\text{w}^\star}} s \right) \right) = \mathcal{P}_\tau \left( \delta g + \mu s \varphi_{\text{out}^\star} \left( \sqrt{\rho_{\text{w}^\star}} s \right) \right)$, we obtain the equivalence with eq. (III.14), which concludes the proof.   $\square$

# IV   Replica computation for Bayes-optimal and ERM estimations

In this section, we present the statistical physics framework and the replica computation leading to the general set of fixed point equations (11) and to the Bayes-optimal fixed point equations (13).

## IV.1   Statistical inference and free entropy

As stressed in Sec. I, both ERM and Bayes-optimal estimations can be analyzed in a unified framework that consists in studying the joint distribution $\mathbb{P}(\mathbf{y}, \mathrm{X})$ in the following posterior distribution

$$\mathbb{P}(\mathbf{w}|\mathbf{y}, \mathrm{X}) = \frac{\mathbb{P}(\mathbf{y}|\mathbf{w}, \mathrm{X})\,\mathbb{P}(\mathbf{w})}{\mathbb{P}(\mathbf{y}, \mathrm{X})}\,, \tag{IV.1}$$

known as the so-called *partition function* in the physics literature. It is the generating function of many useful statistical quantities and is defined by

$$\begin{aligned}
\mathcal{Z}(\mathbf{y}, \mathrm{X}) \equiv P(\mathbf{y}, \mathrm{X}) &= \int_{\mathbb{R}^d} \mathrm{d}\mathbf{w}\, P_{\mathrm{out}}(\mathbf{y}|\mathbf{w}, \mathrm{X})\, P_{\mathrm{w}}(\mathbf{w}) \\
&= \int_{\mathbb{R}^n} \mathrm{d}\mathbf{z}\, P_{\mathrm{out}}(\mathbf{y}|\mathbf{z}) \int_{\mathbb{R}^d} \mathrm{d}\mathbf{w}\, P_{\mathrm{w}}(\mathbf{w})\, \delta\left(\mathbf{z} - \frac{1}{\sqrt{d}}\mathrm{X}\mathbf{w}\right),
\end{aligned} \tag{IV.2}$$

where we introduced the variable $\mathbf{z} = \frac{1}{\sqrt{d}}\mathrm{X}\mathbf{w}$. However in the considered high-dimensional regime $(d \to \infty, n \to \infty, \alpha = \Theta(1))$, we are interested instead in the *averaged* (over instances of input data X and teacher weights $\mathbf{w}^\star$ or equivalently over the output labels $\mathbf{y}$) *free entropy* $\Phi$ defined as

$$\Phi(\alpha) \equiv \mathbb{E}_{\mathbf{y}, \mathrm{X}}\left[\lim_{d \to \infty} \frac{1}{d} \log \mathcal{Z}(\mathbf{y}, \mathrm{X})\right]. \tag{IV.3}$$

The replica method is an heuristic method of statistical mechanics that allows to compute the above average over the random dataset $\{\mathbf{y}, \mathrm{X}\}$. We show in the next section the classical computation for the Generalized Linear Model hypothesis class and i.i.d data X.

## IV.2   Replica computation

### IV.2.1   Derivation

We present here the replica computation of the averaged free entropy $\Phi(\alpha)$ in eq. (IV.3) for general prior distributions $P_{\mathrm{w}}, P_{\mathrm{w}^\star}$ and channel distributions $P_{\mathrm{out}}, P_{\mathrm{out}^\star}$, so that the computation remain valid for both Bayes-optimal and ERM estimation (with any convex loss $l$ and regularizer $r$).

**Replica trick**   The average in eq. (IV.3) is intractable in general, and the computation relies on the so called *replica trick* that consists in applying the identity

$$\mathbb{E}_{\mathbf{y}, \mathrm{X}}\left[\lim_{d \to \infty} \frac{1}{d} \log \mathcal{Z}(\mathbf{y}, \mathrm{X})\right] = \lim_{r \to 0}\left[\lim_{d \to \infty} \frac{1}{d} \frac{\partial \log \mathbb{E}_{\mathbf{y}, \mathrm{X}}\left[\mathcal{Z}(\mathbf{y}, \mathrm{X})^r\right]}{\partial r}\right]. \tag{IV.4}$$

This is interesting in the sense that it reduces the intractable average to the computation of the moments of the averaged partition function, which are easiest quantities to compute. Note that for $r \in \mathbb{N}$, $\mathcal{Z}(\mathbf{y}, \mathrm{X})^r$ represents the partition function of $r \in \mathbb{N}$ identical non-interacting copies of the initial system, called *replicas*. Taking the average will then correlate the replicas, before taking the number of replicas $r \to 0$. Therefore, we assume there exists an analytical continuation so that $r \in \mathbb{R}$ and the limit is well defined. Finally, note we exchanged the order of the limits $r \to 0$ and $d \to \infty$. These technicalities are crucial points but are not rigorously justified and we will ignore them in the rest of the computation.

Thus the replicated partition function in eq. (IV.4) can be written as

$$
\mathbb{E}_{\mathbf{y},\mathrm{X}}\left[\mathcal{Z}\left(\mathbf{y},\mathrm{X}\right)^r\right] = \mathbb{E}_{\mathbf{w}^\star,\mathrm{X}}\left[\prod_{a=1}^r \int_{\mathbb{R}^n} \mathrm{d}\mathbf{z}^a P_{\mathrm{out}^a}\left(\mathbf{y}|\mathbf{z}^a\right) \int_{\mathbb{R}^d} \mathrm{d}\mathbf{w}^a P_{\mathrm{w}^a}\left(\mathbf{w}^a\right) \delta\left(\mathbf{z}^a - \frac{1}{\sqrt{d}}\mathrm{X}\mathbf{w}^a\right)\right]
$$

$$
= \mathbb{E}_{\mathrm{X}} \int_{\mathbb{R}^n} \mathrm{d}\mathbf{y} \int_{\mathbb{R}^n} \mathrm{d}\mathbf{z}^\star P_{\mathrm{out}^\star}\left(\mathbf{y}|\mathbf{z}^\star\right) \int_{\mathbb{R}^d} \mathrm{d}\mathbf{w}^\star P_{\mathrm{w}^\star}\left(\mathbf{w}^\star\right) \delta\left(\mathbf{z}^\star - \frac{1}{\sqrt{d}}\mathrm{X}\mathbf{w}^\star\right)
$$

$$
\times \left[\prod_{a=1}^r \int_{\mathbb{R}^n} \mathrm{d}\mathbf{z}^a P_{\mathrm{out}^a}\left(\mathbf{y}|\mathbf{z}^a\right) \int_{\mathbb{R}^d} \mathrm{d}\mathbf{w}^a P_{\mathrm{w}^a}\left(\mathbf{w}^a\right) \delta\left(\mathbf{z}^a - \frac{1}{\sqrt{d}}\mathrm{X}\mathbf{w}^a\right)\right]
$$

$$
= \mathbb{E}_{\mathrm{X}} \int_{\mathbb{R}^n} \mathrm{d}\mathbf{y} \prod_{a=0}^r \int_{\mathbb{R}^n} \mathrm{d}\mathbf{z}^a P_{\mathrm{out}^a}\left(\mathbf{y}|\mathbf{z}^a\right) \int_{\mathbb{R}^d} \mathrm{d}\mathbf{w}^a P_{\mathrm{w}^a}\left(\mathbf{w}^a\right) \delta\left(\mathbf{z}^a - \frac{1}{\sqrt{d}}\mathrm{X}\mathbf{w}^a\right)
$$

$$(\text{IV.5})$$

with the decoupled channel $P_{\mathrm{out}}\left(\mathbf{y}|\mathbf{z}\right) = \prod_{\mu=1}^n P_{\mathrm{out}}\left(y_\mu|z_\mu\right)$. Note that the average over $\mathbf{y}$ is equivalent to the one over the ground truth vector $\mathbf{w}^\star$, which can be considered as a new replica $\mathbf{w}^0$ with index $a = 0$ leading to a total of $r + 1$ replicas.

We suppose that inputs are drawn from an i.i.d distribution, for example a Gaussian $\mathcal{N}\left(0, 1\right)$. More precisely, for $i, j \in [1 : d]$, $\mu, \nu \in [1 : n]$, $\mathbb{E}_{\mathrm{X}}\left[x_i^{(\mu)} x_j^{(\nu)}\right] = \delta_{\mu\nu}\delta_{ij}$. Hence $z_\mu^a = \frac{1}{\sqrt{d}}\sum_{i=1}^d x_i^{(\mu)} w_i^a$ is the sum of i.i.d random variables. The central limit theorem insures that $z_\mu^a \sim \mathcal{N}\left(\mathbb{E}_{\mathrm{X}}[z_\mu^a], \mathbb{E}_{\mathrm{X}}[z_\mu^a z_\mu^b]\right)$, with the two first moments given by:

$$
\begin{cases}
\mathbb{E}_{\mathrm{X}}[z_\mu^a] = \frac{1}{\sqrt{d}}\sum_{i=1}^d \mathbb{E}_{\mathrm{X}}\left[x_i^{(\mu)}\right] w_i^a = 0 \\
\mathbb{E}_{\mathrm{X}}[z_\mu^a z_\mu^b] = \frac{1}{d}\sum_{ij} \mathbb{E}_{\mathrm{X}}\left[x_i^{(\mu)} x_j^{(\mu)}\right] w_i^a w_j^b = \frac{1}{d}\sum_{ij} \delta_{ij} w_i^a w_j^b = \frac{1}{d}\mathbf{w}^a \cdot \mathbf{w}^b\,.
\end{cases}
$$

$$(\text{IV.6})$$

In the following we introduce the symmetric *overlap* matrix $Q(\{\mathbf{w}^a\}) \equiv \left(\frac{1}{d}\mathbf{w}^a \cdot \mathbf{w}^b\right)_{a,b=0..r}$. Let us define $\tilde{\mathbf{z}}_\mu \equiv (z_\mu^a)_{a=0..r}$ and $\tilde{\mathbf{w}}_i \equiv (w_i^a)_{a=0..r}$. The vector $\tilde{\mathbf{z}}_\mu$ follows a multivariate Gaussian distribution $\tilde{\mathbf{z}}_\mu \sim P_{\tilde{\mathbf{z}}}(\tilde{\mathbf{z}}; Q) = \mathcal{N}_{\tilde{\mathbf{z}}}(\mathbf{0}_{r+1}, Q)$ and as $P_{\tilde{\mathbf{w}}}(\tilde{\mathbf{w}}) = \prod_{a=0}^r P_{\mathrm{w}}(\tilde{w}^a)$ it follows

$$
\mathbb{E}_{\mathbf{y},\mathrm{X}}\left[\mathcal{Z}\left(\mathbf{y},\mathrm{X}\right)^r\right] = \mathbb{E}_{\mathrm{X}} \int_{\mathbb{R}^n} \mathrm{d}\mathbf{y} \prod_{a=0}^r \int_{\mathbb{R}^n} \mathrm{d}\mathbf{z}^a P_{\mathrm{out}^a}\left(\mathbf{y}|\mathbf{z}^a\right) \int_{\mathbb{R}^d} \mathrm{d}\mathbf{w}^a P_{\mathrm{w}^a}\left(\mathbf{w}^a\right) \delta\left(\mathbf{z}^a - \frac{1}{\sqrt{d}}\mathrm{X}\mathbf{w}^a\right)
$$

$$
= \left[\int_{\mathbb{R}} \mathrm{d}y \int_{\mathbb{R}^{r+1}} \mathrm{d}\tilde{\mathbf{z}} P_{\mathrm{out}}\left(y|\tilde{\mathbf{z}}\right) P_{\tilde{\mathbf{z}}}(\tilde{\mathbf{z}}; Q(\tilde{\mathbf{w}}))\right]^n \left[\int_{\mathbb{R}^{r+1}} \mathrm{d}\tilde{\mathbf{w}} P_{\tilde{\mathbf{w}}}\left(\tilde{\mathbf{w}}\right)\right]^d,
$$

because the channel and the prior distributions factorize. Introducing the change of variable and the Fourier representation of the $\delta$-Dirac function, which involves a new ad-hoc parameter $\hat{Q}$:

$$
1 = \int_{\mathbb{R}^{r+1 \times r+1}} \mathrm{d}Q \prod_{a \leq b} \delta\left(dQ_{ab} - \sum_{i=1}^d w_i^a w_i^b\right)
$$

$$
\propto \int_{\mathbb{R}^{r+1 \times r+1}} \mathrm{d}Q \int_{\mathbb{R}^{r+1 \times r+1}} \mathrm{d}\hat{Q} \exp\left(-d\mathrm{Tr}\left[Q\hat{Q}\right]\right) \exp\left(\frac{1}{2}\sum_{i=1}^d \tilde{\mathbf{w}}_i^\mathsf{T} \hat{Q} \tilde{\mathbf{w}}_i\right),
$$

the replicated partition function becomes an integral over the symmetric matrices $Q \in \mathbb{R}^{r+1 \times r+1}$ and $\hat{Q} \in \mathbb{R}^{r+1 \times r+1}$, that can be evaluated using a Laplace method in the $d \to \infty$ limit,

$$
\mathbb{E}_{\mathbf{y},\mathrm{X}}\left[\mathcal{Z}\left(\mathbf{y},\mathrm{X}\right)^r\right] = \int_{\mathbb{R}^{r+1 \times r+1}} \mathrm{d}Q \int_{\mathbb{R}^{r+1 \times r+1}} \mathrm{d}\hat{Q} e^{d\Phi^{(r)}(Q,\hat{Q})} \tag{IV.7}
$$

$$
\underset{d \to \infty}{\simeq} \exp\left(d \cdot \mathbf{extr}_{Q,\hat{Q}}\left\{\Phi^{(r)}(Q, \hat{Q})\right\}\right), \tag{IV.8}
$$

where we defined

$$
\begin{cases}
\Phi^{(r)}(Q,\hat{Q}) = -\mathrm{Tr}\left[Q\hat{Q}\right] + \log \Psi_{\mathrm{w}}^{(r)}(\hat{Q}) + \alpha \log \Psi_{\mathrm{out}}^{(r)}(Q) \\[2mm]
\Psi_{\mathrm{w}}^{(r)}(\hat{Q}) = \displaystyle\int_{\mathbb{R}^{r+1}} \mathrm{d}\tilde{\mathbf{w}} P_{\tilde{\mathrm{w}}}(\tilde{\mathbf{w}}) e^{\frac{1}{2}\tilde{\mathbf{w}}^{\mathsf{T}}\hat{Q}\tilde{\mathbf{w}}} \\[2mm]
\Psi_{\mathrm{out}}^{(r)}(Q) = \displaystyle\int \mathrm{d}y \int_{\mathbb{R}^{r+1}} \mathrm{d}\tilde{\mathbf{z}} P_{\tilde{z}}(\tilde{\mathbf{z}};Q) P_{\mathrm{out}}(y|\tilde{\mathbf{z}}),
\end{cases}
\tag{IV.9}
$$

and $P_{\tilde{z}}(\tilde{\mathbf{z}};Q) = \dfrac{e^{-\frac{1}{2}\tilde{\mathbf{z}}^{\mathsf{T}}Q^{-1}\tilde{\mathbf{z}}}}{\det(2\pi Q)^{1/2}}$.

Finally switching the two limits $r \to 0$ and $d \to \infty$, the quenched free entropy $\Phi$ simplifies as a saddle point equation

$$
\Phi(\alpha) = \mathbf{extr}_{Q,\hat{Q}} \left\{ \lim_{r\to 0} \frac{\partial \Phi^{(r)}(Q,\hat{Q})}{\partial r} \right\},
\tag{IV.10}
$$

over symmetric matrices $Q \in \mathbb{R}^{r+1\times r+1}$ and $\hat{Q} \in \mathbb{R}^{r+1\times r+1}$. In the following we will assume a simple ansatz for these matrices in order to first obtain an analytic expression in $r$ before taking the derivative with respect to $r$.

**RS free entropy**  Let's compute the functional $\Phi^{(r)}(Q,\hat{Q})$ appearing in the free entropy eq. (IV.10) in the simplest ansatz: the Replica Symmetric ansatz. This later assumes that all replica remain equivalent with a common overlap $q = \frac{1}{d}\mathbf{w}^a \cdot \mathbf{w}^b$ for $a \neq b$, a norm $Q = \frac{1}{d}\|\mathbf{w}^a\|_2^2$, and an overlap with the ground truth $m = \frac{1}{d}\mathbf{w}^a \cdot \mathbf{w}^\star$, leading to the following expressions of the replica symmetric matrices $Q_{\mathrm{rs}} \in \mathbb{R}^{r+1\times r+1}$ and $\hat{Q}_{\mathrm{rs}} \in \mathbb{R}^{r+1\times r+1}$:

$$
Q_{\mathrm{rs}} = \begin{pmatrix} Q^0 & m & \dots & m \\ m & Q & \dots & \dots \\ \dots & \dots & \dots & q \\ m & \dots & q & Q \end{pmatrix}
\quad \text{and} \quad
\hat{Q}_{\mathrm{rs}} = \begin{pmatrix} \hat{Q}^0 & \hat{m} & \dots & \hat{m} \\ \hat{m} & -\frac{1}{2}\hat{Q} & \dots & \dots \\ \dots & \dots & \dots & \hat{q} \\ \hat{m} & \dots & \hat{q} & -\frac{1}{2}\hat{Q} \end{pmatrix},
\tag{IV.11}
$$

with $Q^0 = \rho_{\mathrm{w}^\star} = \frac{1}{d}\|\mathbf{w}^\star\|_2^2$. Let's compute separately the terms involved in the functional $\Phi^{(r)}(Q,\hat{Q})$ eq. (IV.9) with this ansatz: the first is a trace term, the second a term $\Psi_{\mathrm{w}}^{(r)}$ depending on the prior distributions $P_{\mathrm{w}}$, $P_{\mathrm{w}^\star}$ and finally the third a term $\Psi_{\mathrm{out}}^{(r)}$ that depends on the channel distributions $P_{\mathrm{out}^\star}, P_{\mathrm{out}}$.

**Trace term**  The trace term can be easily computed and takes the following form:

$$
\mathrm{Tr}\left(Q\hat{Q}\right)\Big|_{\mathrm{rs}} = Q^0\hat{Q}^0 + rm\hat{m} - \frac{1}{2}rQ\hat{Q} + \frac{r(r-1)}{2}q\hat{q}.
\tag{IV.12}
$$

**Prior integral**  Evaluated at the RS fixed point, and using a Gaussian identity also known as a Hubbard-Stratonovich transformation $\mathbb{E}_\xi \exp(\sqrt{a}\xi) = e^{\frac{a}{2}}$, the prior integral can be further simplified

$$
\begin{aligned}
\Psi_{\mathrm{w}}^{(r)}(\hat{Q})\Big|_{\mathrm{rs}} &= \int_{\mathbb{R}^{r+1}} d\tilde{\mathbf{w}} P_{\tilde{\mathrm{w}}}(\tilde{\mathbf{w}}) e^{\frac{1}{2}\tilde{\mathbf{w}}^{\mathsf{T}}\hat{Q}_{\mathrm{rs}}\tilde{\mathbf{w}}} \\
&= \mathbb{E}_{w^\star} e^{\frac{1}{2}\hat{Q}^0(w^\star)^2} \int_{\mathbb{R}^r} d\tilde{\mathbf{w}} P_{\tilde{\mathrm{w}}}(\tilde{\mathbf{w}}) e^{w^\star\hat{m}\sum_{a=1}^r \tilde{w}^a - \frac{1}{2}(\hat{Q}+\hat{q})\sum_{a=1}^r (\tilde{w}^a)^2 + \frac{1}{2}\hat{q}(\sum_{a=1}^r \tilde{w}^a)^2} \\
&= \mathbb{E}_{\xi,w^\star} e^{\frac{1}{2}\hat{Q}^0(w^\star)^2} \left[ \mathbb{E}_w \exp\left( \left[\hat{m}w^\star w - \frac{1}{2}(\hat{Q}+\hat{q})w^2 + \hat{q}^{1/2}\xi w\right] \right) \right]^r.
\end{aligned}
\tag{IV.13}
$$

**Channel integral**  Let's focus on the inverse matrix

$$
Q_{\mathrm{rs}}^{-1} = \begin{bmatrix} Q_{00}^{-1} & Q_{01}^{-1} & Q_{01}^{-1} & Q_{01}^{-1} \\ Q_{01}^{-1} & Q_{11}^{-1} & Q_{12}^{-1} & Q_{12}^{-1} \\ Q_{01}^{-1} & Q_{12}^{-1} & Q_{11}^{-1} & Q_{12}^{-1} \\ Q_{01}^{-1} & Q_{12}^{-1} & Q_{12}^{-1} & Q_{11}^{-1} \end{bmatrix}
\tag{IV.14}
$$

with

$$
\begin{cases}
Q_{00}^{-1} &= \left(Q^0 - rm(Q + (r-1)q)^{-1}m\right)^{-1} \\[4pt]
Q_{01}^{-1} &= -\left(Q^0 - rm(Q + (r-1)q)^{-1}m\right)^{-1} m(q + (r-1)q)^{-1} \\[4pt]
Q_{11}^{-1} &= (Q-q)^{-1} - (Q + (r-1)q)^{-1}q(Q-q)^{-1} \\
&\quad +(Q + (r-1)q)^{-1}m \left(Q^0 - rm(Q + (r-1)q)^{-1}m\right)^{-1} m(Q + (r-1)q)^{-1} \\[4pt]
Q_{12}^{-1} &= -(Q + (r-1)q)^{-1}q(Q-q)^{-1} \\
&\quad +(Q + (r-1)q)^{-1}m \left(Q - rm(Q + (r-1)q)^{-1}m\right)^{-1} m(Q + (r-1)q)^{-1}
\end{cases}
$$

and its determinant:

$$
\det Q_{\mathrm{rs}} = (Q-q)^{r-1}\left(Q + (r-1)q\right)\left(Q^0 - rm(Q + (r-1)q)^{-1}m\right)
$$

Using the same kind of Gaussian transformation, we obtain

$$
\Psi_{\mathrm{out}}^{(r)}(Q)\Big|_{\mathrm{rs}} = \int \mathrm{d}y \int_{\mathbb{R}^{r+1}} d\tilde{\mathbf{z}}\, e^{-\frac{1}{2}\tilde{\mathbf{z}}^{\mathsf{T}} Q_{\mathrm{rs}}^{-1}\tilde{\mathbf{z}} - \frac{1}{2}\log(\det(2\pi Q_{\mathrm{rs}}))} P_{\mathrm{out}}(y|\tilde{\mathbf{z}})
$$

$$
= \mathbb{E}_{y,\xi}\, e^{-\frac{1}{2}\log(\det(2\pi Q_{\mathrm{rs}}))}
$$

$$
\times \int \mathrm{d}z^\star P_{\mathrm{out}^\star}(y|z^\star)\, e^{-\frac{1}{2}Q_{00}^{-1}(z^\star)^2} \left[\int dz P_{\mathrm{out}}(y|z)\, e^{-Q_{01}^{-1}z^\star z - \frac{1}{2}\left(Q_{11}^{-1} - Q_{12}^{-1}\right)z^2 - Q_{12}^{-1/2}\xi z}\right]^r
$$

## IV.3   ERM and Bayes-optimal free entropy

Taking carefully the derivative and the $r \to 0$ limit imposes $\hat{Q}^0 = 0$ and we finally obtain the replica symmetric free entropy $\Phi_{\mathrm{rs}}$:

$$
\Phi_{\mathrm{rs}}(\alpha) \equiv \mathbb{E}_{\mathbf{y},\mathbf{X}}\left[\lim_{d\to\infty}\frac{1}{d}\log\left(\mathcal{Z}(\mathbf{y},\mathbf{X})\right)\right] \tag{IV.15}
$$

$$
= \mathbf{extr}_{Q,\hat{Q},q,\hat{q},m,\hat{m}}\left\{-m\hat{m} + \frac{1}{2}Q\hat{Q} + \frac{1}{2}q\hat{q} + \Psi_{\mathrm{w}}\left(\hat{Q},\hat{m},\hat{q}\right) + \alpha\Psi_{\mathrm{out}}\left(Q,m,q;\rho_{\mathrm{w}^\star}\right)\right\},
$$

where $\rho_{\mathrm{w}^\star} = \lim_{d\to\infty} \mathbb{E}_{\mathbf{w}^\star}\frac{1}{d}\|\mathbf{w}^\star\|_2^2$ and the channel and prior integrals are defined by

$$
\Psi_{\mathrm{w}}\left(\hat{Q},\hat{m},\hat{q}\right) \equiv \mathbb{E}_\xi\left[\mathcal{Z}_{\mathrm{w}^\star}\left(\hat{m}\hat{q}^{-1/2}\xi, \hat{m}\hat{q}^{-1}\hat{m}\right)\log \mathcal{Z}_{\mathrm{w}}\left(\hat{q}^{1/2}\xi, \hat{Q} + \hat{q}\right)\right],
$$

$$
\Psi_{\mathrm{out}}\left(Q,m,q;\rho_{\mathrm{w}^\star}\right) \equiv \mathbb{E}_{y,\xi}\left[\mathcal{Z}_{\mathrm{out}^\star}\left(y, mq^{-1/2}\xi, \rho_{\mathrm{w}^\star} - mq^{-1}m\right)\log \mathcal{Z}_{\mathrm{out}}\left(y, q^{1/2}\xi, Q - q\right)\right],
\tag{IV.16}
$$

where again $\mathcal{Z}_{\mathrm{out}^\star}$ and $\mathcal{Z}_{\mathrm{w}^\star}$ are defined in eq. (I.7) and depend on the *teacher*, while the denoising functions $\mathcal{Z}_{\mathrm{out}}$ and $\mathcal{Z}_{\mathrm{w}}$ depend on the inference model. In particular, we explicit in the next sections the above free entropy in the case of ERM and Bayes-optimal estimation.

### IV.3.1   ERM estimation

As described in eq. (I.4), the free entropy for ERM estimation is therefore given by eq. (IV.15) if we take $-\log \mathbb{P}\left(\mathbf{y}|\mathbf{z}\right) = l(\mathbf{y},\mathbf{z})$ and $-\log \mathbb{P}\left(\mathbf{w}\right) = r(\mathbf{w})$. As described in Sec. I.3.2 they lead to the following partition functions:

$$
\mathcal{Z}_{\mathrm{w}}^\lambda\left(\gamma,\Lambda\right) = \lim_{\Delta\to 0} e^{-\frac{1}{\Delta}\mathcal{M}_{\Lambda^{-1}}[r(\lambda,.)](\Lambda^{-1}\gamma)} e^{-\frac{1}{2\Delta}\gamma^2\Lambda^{-1}},
$$

$$
\mathcal{Z}_{\mathrm{out}}\left(y,\omega,V\right) = \lim_{\Delta\to 0} \frac{e^{-\frac{1}{\Delta}\mathcal{M}_{\frac{V}{\Delta}}[l(y,.)](\omega)}}{\sqrt{2\pi V}\sqrt{2\pi\Delta}},
\tag{IV.17}
$$

with the Moreau-Yosida regularization (I.11).

### IV.3.2 Bayes-optimal estimation

In the Bayes-optimal case, we have access to the ground truth distributions $\mathbb{P}(\mathbf{y}|\mathbf{z}) = P_{\text{out}^\star}(\mathbf{y}|\mathbf{z})$ and $\mathbb{P}(\mathbf{w}) = P_{\text{w}^\star}(\mathbf{w})$, and therefore $\mathcal{Z}_{\text{out}} = \mathcal{Z}_{\text{out}^\star}$, $\mathcal{Z}_{\text{w}} = \mathcal{Z}_{\text{w}^\star}$. Nishimori conditions in the Bayes-optimal case [3] imply that $Q = \rho_{\text{w}^\star}$, $m = q = q_{\text{b}}$, $\hat{Q} = 0$, $\hat{m} = \hat{q} = \hat{q}_{\text{b}}$. Therefore the free entropy eq. (IV.15) simplifies as an optimization problem over two scalar *overlaps* $q_{\text{b}}, \hat{q}_{\text{b}}$:

$$\Phi^{\text{b}}(\alpha) = \mathbf{extr}_{q_{\text{b}}, \hat{q}_{\text{b}}} \left\{ -\frac{1}{2} q_{\text{b}} \hat{q}_{\text{b}} + \Psi^{\text{b}}_{\text{w}}(\hat{q}_{\text{b}}) + \alpha \Psi^{\text{b}}_{\text{out}}(q_{\text{b}}; \rho_{\text{w}^\star}) \right\}, \tag{IV.18}$$

with free entropy terms $\Psi^{\text{b}}_{\text{w}}$ and $\Psi^{\text{b}}_{\text{out}}$ given by

$$\Psi^{\text{b}}_{\text{w}}(\hat{q}) = \mathbb{E}_\xi \left[ \mathcal{Z}_{\text{w}^\star} \left( \hat{q}^{1/2} \xi, \hat{q} \right) \log \mathcal{Z}_{\text{w}^\star} \left( \hat{q}^{1/2} \xi, \hat{q} \right) \right],$$

$$\Psi^{\text{b}}_{\text{out}}(q; \rho_{\text{w}^\star}) = \mathbb{E}_{y,\xi} \left[ \mathcal{Z}_{\text{out}^\star} \left( y, q^{1/2} \xi, \rho_{\text{w}^\star} - q \right) \log \mathcal{Z}_{\text{out}^\star} \left( y, q^{1/2} \xi, \rho_{\text{w}^\star} - q \right) \right].$$

and again $\mathcal{Z}_{\text{out}^\star}$ and $\mathcal{Z}_{\text{w}^\star}$ are defined in eq. (I.7). The above replica symmetric free entropy in the Bayes-optimal case has been rigorously proven in [1].

## IV.4 Sets of fixed point equations

As highlighted in Sec. II, the asymptotic overlaps $m, q$ measure the performances of the ERM or Bayes-optimal statistical estimators, whose behaviours are respectively characterized by extremizing the free entropy (IV.15) and (IV.18). This section is devoted to derive the corresponding sets of fixed point equations.

### IV.4.1 ERM estimation

Extremizing the free entropy eq. (IV.15), we easily obtain the set of six fixed point equations

$$\begin{aligned} \hat{Q} &= -2\alpha \partial_Q \Psi_{\text{out}}, & Q &= -2\partial_{\hat{Q}} \Psi_{\text{w}} \\ \hat{q} &= -2\alpha \partial_q \Psi_{\text{out}}, & q &= -2\partial_{\hat{q}} \Psi_{\text{w}}, \\ \hat{m} &= \alpha \partial_m \Psi_{\text{out}}, & m &= \partial_{\hat{m}} \Psi_{\text{w}}. \end{aligned} \tag{IV.19}$$

These equations can be formulated as functions of the partition functions $\mathcal{Z}_{\text{out}^\star}$, $\mathcal{Z}_{\text{w}^\star}$ and the denoising functions $f_{\text{out}^\star}, f_{\text{w}^\star}, f_{\text{out}}, f_{\text{w}}$ defined in eq. (I.8) and eq. (I.12). The derivation is shown in Appendix. IV.5.3 and defining the natural variables $\Sigma = Q - q$, $\hat{\Sigma} = \hat{Q} + \hat{q}$, $\eta \equiv \frac{m^2}{\rho_{\text{w}^\star} q}$ and $\hat{\eta} \equiv \frac{\hat{m}^2}{\hat{q}}$, it can be written as

$$\begin{aligned} m &= \mathbb{E}_\xi \left[ \mathcal{Z}_{\text{w}^\star} \left( \sqrt{\hat{\eta}} \xi, \hat{\eta} \right) f_{\text{w}^\star} \left( \sqrt{\hat{\eta}} \xi, \hat{\eta} \right) f_{\text{w}} \left( \hat{q}^{1/2} \xi, \hat{\Sigma} \right) \right], \\ q &= \mathbb{E}_\xi \left[ \mathcal{Z}_{\text{w}^\star} \left( \sqrt{\hat{\eta}} \xi, \hat{\eta} \right) f_{\text{w}} \left( \hat{q}^{1/2} \xi, \hat{\Sigma} \right)^2 \right], \\ \Sigma &= \mathbb{E}_\xi \left[ \mathcal{Z}_{\text{w}^\star} \left( \sqrt{\hat{\eta}} \xi, \hat{\eta} \right) \partial_\gamma f_{\text{w}} \left( \hat{q}^{1/2} \xi, \hat{\Sigma} \right) \right], \\ \hat{m} &= \alpha \mathbb{E}_{y,\xi} \left[ \mathcal{Z}_{\text{out}^\star}(.) \cdot f_{\text{out}^\star} \left( y, \sqrt{\rho_{\text{w}^\star} \eta} \xi, \rho_{\text{w}^\star}(1-\eta) \right) f_{\text{out}} \left( y, q^{1/2} \xi, \Sigma \right) \right], \\ \hat{q} &= \alpha \mathbb{E}_{y,\xi} \left[ \mathcal{Z}_{\text{out}^\star} \left( y, \sqrt{\rho_{\text{w}^\star} \eta} \xi, \rho_{\text{w}^\star}(1-\eta) \right) f_{\text{out}} \left( y, q^{1/2} \xi, \Sigma \right)^2 \right], \\ \hat{\Sigma} &= -\alpha \mathbb{E}_{y,\xi} \left[ \mathcal{Z}_{\text{out}^\star} \left( y, \sqrt{\rho_{\text{w}^\star} \eta} \xi, \rho_{\text{w}^\star}(1-\eta) \right) \partial_\omega f_{\text{out}} \left( y, q^{1/2} \xi, \Sigma \right) \right], \end{aligned} \tag{IV.20}$$

and we finally obtain the set of equations eqs. (III.15).

### IV.4.2 Bayes-optimal estimation

Extremizing the Bayes-optimal free entropy eq. (IV.18), we easily obtain the set of 2 fixed point equations over the scalar parameters $q_{\text{b}}, \hat{q}_{\text{b}}$. In fact, it can also be deduced from eq. (IV.20) using the

Nishimori conditions $f_{\mathrm{w}} = f_{\mathrm{w}^\star}$, $f_{\mathrm{out}} = f_{\mathrm{out}^\star}$, $m = q = q_{\mathrm{b}}$, $\Sigma = \rho_{\mathrm{w}^\star} - q$, $\hat{m} = \hat{q} = \hat{q}_{\mathrm{b}}$ and $\hat{Q} = 0$ that lead to the result (13) in Thm. 2.4, from [1]

$$
\begin{aligned}
\hat{q}_{\mathrm{b}} &= \alpha \mathbb{E}_{y,\xi} \left[ \mathcal{Z}_{\mathrm{out}^\star} \left( y, q_{\mathrm{b}}^{1/2}\xi, \rho_{\mathrm{w}^\star} - q_{\mathrm{b}} \right) f_{\mathrm{out}^\star} \left( y, q_{\mathrm{b}}^{1/2}\xi, \rho_{\mathrm{w}^\star} - q_{\mathrm{b}} \right)^2 \right] , \\
q_{\mathrm{b}} &= \mathbb{E}_{\xi} \left[ \mathcal{Z}_{\mathrm{w}^\star} \left( \hat{q}_{\mathrm{b}}^{1/2}\xi, \hat{q}_{\mathrm{b}} \right) f_{\mathrm{w}^\star} \left( \hat{q}_{\mathrm{b}}^{1/2}\xi, \hat{q}_{\mathrm{b}} \right)^2 \right] .
\end{aligned}
\tag{IV.21}
$$

## IV.5 Useful derivations

In this section, we give useful computation steps that we used to transform the sets of fixed point equations (IV.19).

### IV.5.1 Prior free entropy term

In specific simple cases, the prior free entropy term

$$
\Psi_{\mathrm{w}} \left( \hat{Q}, \hat{m}, \hat{q} \right) \equiv \mathbb{E}_{\xi} \left[ \mathcal{Z}_{\mathrm{w}^\star} \left( \hat{m}\hat{q}^{-1/2}\xi, \hat{m}\hat{q}^{-1}\hat{m} \right) \log \mathcal{Z}_{\mathrm{w}} \left( \hat{q}^{1/2}\xi, \hat{Q} + \hat{q} \right) \right]
$$

in (IV.16) can be computed explicitly. This is the case of Gaussian and binary priors $P_{\mathrm{w}^\star}$ with $\ell_2$ regularization. In particular, they lead surprisingly to the same expression meaning that choosing a binary or Gaussian teacher distribution does not affect the ERM performances with $\ell_2$ regularization.

**Gaussian prior** Let us compute the corresponding free entropy term with partition functions $\mathcal{Z}_{\mathrm{w}^\star}$ for a Gaussian prior $P_{\mathrm{w}^\star}(w^\star) = \mathcal{N}_{w^\star}(0, \rho_{\mathrm{w}^\star})$ and $\mathcal{Z}_{\mathrm{w}}^{\ell_2,\lambda}$ for a $\ell_2$ regularization respectively given by eq. (I.24) and eq. (I.30):

$$
\mathcal{Z}_{\mathrm{w}^\star}(\gamma, \Lambda) = \frac{e^{\frac{\gamma^2 \rho_{\mathrm{w}^\star}}{2(\Lambda \rho_{\mathrm{w}^\star} + 1)}}}{\sqrt{\Lambda \rho_{\mathrm{w}^\star} + 1}} , \quad \mathcal{Z}_{\mathrm{w}}^{\ell_2,\lambda}(\gamma, \Lambda) = \frac{e^{\frac{\gamma^2}{2(\Lambda + \lambda)}}}{\sqrt{\Lambda + \lambda}} .
$$

The prior free entropy term reads

$$
\begin{aligned}
\Psi_{\mathrm{w}} \left( \hat{Q}, \hat{m}, \hat{q} \right) &= \mathbb{E}_{\xi} \left[ \mathcal{Z}_{\mathrm{w}^\star} \left( \hat{m}\hat{q}^{-1/2}\xi, \hat{m}\hat{q}^{-1}\hat{m} \right) \log \mathcal{Z}_{\mathrm{w}}^{\ell_2,\lambda} \left( \hat{q}^{1/2}\xi, \hat{q} + \hat{Q} \right) \right] \\
&= \mathbb{E}_{\xi} \left[ \mathcal{Z}_{\mathrm{w}^\star} \left( \hat{m}\hat{q}^{-1/2}\xi, \hat{m}\hat{q}^{-1}\hat{m} \right) \left( \frac{\hat{q}\xi^2}{2\left(\lambda + \hat{Q} + \hat{q}\right)} - \frac{1}{2}\log\left(\lambda + \hat{Q} + \hat{q}\right) \right) \right] \\
&= \int \mathrm{d}\xi \mathcal{N}_{\xi}\left(0, 1 + \rho_{\mathrm{w}^\star}\hat{m}^2\hat{q}^{-1}\right) \left( \frac{\hat{q}\xi^2}{2\left(\lambda + \hat{Q} + \hat{q}\right)} - \frac{1}{2}\log\left(\lambda + \hat{Q} + \hat{q}\right) \right) \\
&= \frac{1}{2} \left( \frac{\hat{q} + \rho_{\mathrm{w}^\star}\hat{m}^2}{\lambda + \hat{Q} + \hat{q}} - \log\left(\lambda + \hat{Q} + \hat{q}\right) \right)
\end{aligned}
\tag{IV.22}
$$

In the Bayes-optimal case for $\rho_{\mathrm{w}^\star} = 1$, the computation is similar and is given by the above expression with $\lambda = 1$, $\hat{Q} = 0$, $\hat{m} = \hat{q}$:

$$
\Psi_{\mathrm{w}}^{\mathrm{bayes}}(\hat{q}) = \quad = \frac{1}{2}\left(\hat{q} - \log(1 + \hat{q})\right)
\tag{IV.23}
$$

**Binary prior** Let us compute the corresponding free entropy term with partition functions $\mathcal{Z}_{\mathrm{w}^\star}$ for a binary prior $P_{\mathrm{w}^\star}(w^\star) = \frac{1}{2}\left(\delta(w^\star - 1) + \delta(w^\star + 1)\right)$ and $\mathcal{Z}_{\mathrm{w}}^{\ell_2,\lambda}$ for a $\ell_2$ regularization respectively given by eq. (I.25) and eq. (I.30):

$$
\mathcal{Z}_{\mathrm{w}^\star}(\gamma, \Lambda) = e^{-\frac{\Lambda}{2}}\cosh(\gamma), \quad \mathcal{Z}_{\mathrm{w}}^{\ell_2,\lambda}(\gamma, \Lambda) = \frac{e^{\frac{\gamma^2}{2(\Lambda + \lambda)}}}{\sqrt{\Lambda + \lambda}} .
$$

The entropy term $\Psi_{\mathrm{w}}$ reads

$$
\begin{aligned}
\Psi_{\mathrm{w}}\left(\hat{Q}, \hat{m}, \hat{q}\right) &= \mathbb{E}_{\xi}\left[\mathcal{Z}_{\mathrm{w}^{\star}}\left(\hat{m}\hat{q}^{-1/2}\xi, \hat{m}\hat{q}^{-1}\hat{m}\right)\log \mathcal{Z}_{\mathrm{w}}^{\ell_2,\lambda}\left(\hat{q}^{1/2}\xi, \hat{q}+\hat{Q}\right)\right] \\
&= \mathbb{E}_{\xi}\left[\mathcal{Z}_{\mathrm{w}^{\star}}\left(\hat{m}\hat{q}^{-1/2}\xi, \hat{m}\hat{q}^{-1}\hat{m}\right)\left(\frac{\hat{q}\xi^2}{2\left(\lambda+\hat{Q}+\hat{q}\right)}-\frac{1}{2}\log\left(\lambda+\hat{Q}+\hat{q}\right)\right)\right] \\
&= \int d\xi \frac{e^{-\frac{\xi^2}{2}}}{\sqrt{2\pi}} e^{-\frac{\hat{m}\hat{q}^{-1}\hat{m}}{2}}\cosh\left(\hat{m}\hat{q}^{-1/2}\xi\right)\left(\frac{\hat{q}\xi^2}{2\left(\lambda+\hat{Q}+\hat{q}\right)}-\frac{1}{2}\log\left(\lambda+\hat{Q}+\hat{q}\right)\right) \\
&= \frac{1}{2}\left(\frac{\hat{q}+\hat{m}^2}{\lambda+\hat{Q}+\hat{q}}-\log\left(\lambda+\hat{Q}+\hat{q}\right)\right)
\end{aligned}
\tag{IV.24}
$$

We recover exactly the same free entropy term than for Gaussian prior teacher eq. (IV.22) for $\rho_{\mathrm{w}^{\star}}=1$.

### IV.5.2 Updates derivatives

Let's compute, in full generality, the derivative of the partition functions defined in Sec. I.5 and that will be useful to simplify the set (IV.19).

$$
\begin{aligned}
\partial_{\gamma}\mathcal{Z}_{\mathrm{w}}\left(\gamma, \Lambda\right) &= \mathcal{Z}_{\mathrm{w}}\left(\gamma, \Lambda\right)\times \mathbb{E}_{Q_{\mathrm{w}}}\left[w\right] = \mathcal{Z}_{\mathrm{w}}\left(\gamma, \Lambda\right)f_{\mathrm{w}}\left(\gamma, \Lambda\right) \\
\partial_{\Lambda}\mathcal{Z}_{\mathrm{w}}\left(\gamma, \Lambda\right) &= -\frac{1}{2}\mathcal{Z}_{\mathrm{w}}\left(\gamma, \Lambda\right)\times \mathbb{E}_{Q_{\mathrm{w}}}\left[w^2\right] = -\frac{1}{2}\left(\partial_{\gamma}f_{\mathrm{w}}(\gamma, \Lambda)+f_{\mathrm{w}}^2(\gamma, \Lambda)\right) \\
\partial_{\omega}\mathcal{Z}_{\mathrm{out}}\left(y, \omega, V\right) &= \mathcal{Z}_{\mathrm{out}}\left(y, \omega, V\right)\times V^{-1}\mathbb{E}_{Q_{\mathrm{out}}}\left[z-\omega\right] \\
&= \mathcal{Z}_{\mathrm{out}}\left(y, \omega, V\right)f_{\mathrm{out}}\left(y, \omega, V\right) \\
\partial_{V}\mathcal{Z}_{\mathrm{out}}\left(y, \omega, V\right) &= \frac{1}{2}\mathcal{Z}_{\mathrm{out}}\left(y, \omega, V\right)\times\left(\mathbb{E}_{Q_{\mathrm{out}}}\left[V^{-2}(z-\omega)^2\right]-V^{-1}\right) \\
&= \frac{1}{2}\mathcal{Z}_{\mathrm{out}}\left(y, \omega, V\right)\left(\partial_{\omega}f_{\mathrm{out}}\left(y, \omega, V\right)+f_{\mathrm{out}}^2\left(y, \omega, V\right)\right)
\end{aligned}
\tag{IV.25}
$$

### IV.5.3 Simplifications of the fixed point equations

We recall the set of fixed point equations eq. (IV.19)

$$
\begin{aligned}
\hat{Q} &= -2\alpha\partial_{Q}\Psi_{\mathrm{out}}\,, & Q &= -2\partial_{\hat{Q}}\Psi_{\mathrm{w}} \\
\hat{q} &= -2\alpha\partial_{q}\Psi_{\mathrm{out}}\,, & q &= -2\partial_{\hat{q}}\Psi_{\mathrm{w}}\,, \\
\hat{m} &= \alpha\partial_{m}\Psi_{\mathrm{out}}\,, & m &= \partial_{\hat{m}}\Psi_{\mathrm{w}}\,,
\end{aligned}
\tag{IV.26}
$$

that can be simplified and formulated as functions of $\mathcal{Z}_{\mathrm{out}^{\star}}$, $\mathcal{Z}_{\mathrm{w}^{\star}}$, $f_{\mathrm{out}^{\star}}$, $f_{\mathrm{w}^{\star}}$, $f_{\mathrm{out}}$, and $f_{\mathrm{w}}$ defined in eq. (I.8) and eq. (I.12), using the derivatives in (IV.25).

**Equation over $\hat{q}$**

$$\partial_q \Psi_{\text{out}} = \partial_q \mathbb{E}_{y,\xi} \left[ \mathcal{Z}_{\text{out}^\star} \left( y, mq^{-1/2}\xi, \rho_{\text{w}^\star} - mq^{-1}m \right) \log \mathcal{Z}_{\text{out}} \left( y, q^{1/2}\xi, Q - q \right) \right]$$

$$= \mathbb{E}_{y,\xi} \left[ \partial_q \omega^\star \partial_\omega \mathcal{Z}_{\text{out}^\star} \log \mathcal{Z}_{\text{out}} + \partial_q V^\star \partial_V \mathcal{Z}_{\text{out}^\star} \log \mathcal{Z}_{\text{out}} \right.$$
$$\left. + \frac{\mathcal{Z}_{\text{out}^\star}}{\mathcal{Z}_{\text{out}}} \left( \partial_q \omega \partial_\omega \mathcal{Z}_{\text{out}} + \partial_q V \partial_V \mathcal{Z}_{\text{out}} \right) \right]$$

$$= \mathbb{E}_{y,\xi} \left[ -\frac{m}{2} q^{-3/2} \xi f_{\text{out}^\star} \mathcal{Z}_{\text{out}^\star} \log \mathcal{Z}_{\text{out}} + \frac{m^2 q^{-2}}{2} \left( \partial_\omega f_{\text{out}^\star} + f_{\text{out}^\star}^2 \right) \mathcal{Z}_{\text{out}^\star} \log \mathcal{Z}_{\text{out}} \right.$$
$$\left. + \frac{\mathcal{Z}_{\text{out}^\star}}{\mathcal{Z}_{\text{out}}} \left( \frac{1}{2} q^{-1/2} \xi f_{\text{out}} \mathcal{Z}_{\text{out}} - \frac{1}{2} \left( \partial_\omega f_{\text{out}} + f_{\text{out}}^2 \right) \mathcal{Z}_{\text{out}} \right) \right]$$

$$= \frac{1}{2} \mathbb{E}_{y,\xi} \left[ -m^2 q^{-2} \partial_\xi \left( f_{\text{out}^\star} \mathcal{Z}_{\text{out}^\star} \log \mathcal{Z}_{\text{out}} \right) + m^2 q^{-2} \left( \partial_\omega f_{\text{out}^\star} + f_{\text{out}^\star}^2 \right) \mathcal{Z}_{\text{out}^\star} \log \mathcal{Z}_{\text{out}} \right.$$
$$\left. + \left( \partial_\xi \left( f_{\text{out}} \mathcal{Z}_{\text{out}^\star} \right) - \left( \partial_\omega f_{\text{out}} + f_{\text{out}}^2 \right) \mathcal{Z}_{\text{out}^\star} \right) \right] \qquad \text{(Stein lemma)}$$

$$= \frac{1}{2} \mathbb{E}_{y,\xi} \left[ -m^2 q^{-2} \left( \partial_\omega f_{\text{out}^\star} \log \mathcal{Z}_{\text{out}} + \mathcal{Z}_{\text{out}^\star} f_{\text{out}^\star}^2 \log \mathcal{Z}_{\text{out}} \right. \right.$$
$$\left. - \left( \partial_\omega f_{\text{out}^\star} + f_{\text{out}^\star}^2 \right) \mathcal{Z}_{\text{out}^\star} \log \mathcal{Z}_{\text{out}} \right) \right] + \frac{1}{2} \mathbb{E}_{y,\xi} \left[ -mq^{-1} \mathcal{Z}_{\text{out}^\star} f_{\text{out}^\star} f_{\text{out}} \right]$$
$$+ \frac{1}{2} \mathbb{E}_{y,\xi} \left[ \partial_\omega f_{\text{out}} \mathcal{Z}_{\text{out}} + mq^{-1} \mathcal{Z}_{\text{out}^\star} f_{\text{out}^\star} f_{\text{out}} - \left( \partial_\omega f_{\text{out}} + f_{\text{out}}^2 \right) \mathcal{Z}_{\text{out}^\star} \right]$$

$$= -\frac{1}{2} \mathbb{E}_{y,\xi} \left[ \mathcal{Z}_{\text{out}^\star} \left( y, mq^{-1/2}\xi, \rho_{\text{w}^\star} - mq^{-1}m \right) f_{\text{out}}^2 \left( y, q^{1/2}\xi, Q - q \right) \right] ,$$
$$\text{(Simplifications with (IV.25))}$$

that leads to

$$\hat{q} = -2\alpha \partial_q \Psi_{\text{out}} = \alpha \mathbb{E}_{y,\xi} \left[ \mathcal{Z}_{\text{out}^\star} \left( y, mq^{-1/2}\xi, \rho_{\text{w}^\star} - mq^{-1}m \right) f_{\text{out}} \left( y, q^{1/2}\xi, Q - q \right)^2 \right] .$$
$$\text{(IV.27)}$$

**Equation over $\hat{m}$**

$$\partial_m \Psi_{\text{out}} = \mathbb{E}_{y,\xi} \left[ \partial_m \mathcal{Z}_{\text{out}^\star} \left( y, mq^{-1/2}\xi, \rho_{\text{w}^\star} - mq^{-1}m \right) \log \mathcal{Z}_{\text{out}} \left( y, q^{1/2}\xi, Q - q \right) \right]$$

$$= \mathbb{E}_{y,\xi} \left[ \left( \partial_m \omega^\star \partial_\omega \mathcal{Z}_{\text{out}^\star} + \partial_m V^\star \partial_V \mathcal{Z}_{\text{out}^\star} \right) \log \mathcal{Z}_{\text{out}} \right]$$

$$= \mathbb{E}_{y,\xi} \left[ \left( q^{-1/2} \xi f_{\text{out}^\star} \mathcal{Z}_{\text{out}^\star} - mq^{-1} \left( \partial_\omega f_{\text{out}^\star} + f_{\text{out}^\star}^2 \right) \mathcal{Z}_{\text{out}^\star} \right) \log \mathcal{Z}_{\text{out}} \right]$$

$$= \mathbb{E}_{y,\xi} \left[ \partial_\xi \left( f_{\text{out}^\star} \mathcal{Z}_{\text{out}^\star} \log \mathcal{Z}_{\text{out}} \right) - \left( \partial_\omega f_{\text{out}^\star} + f_{\text{out}^\star}^2 \right) \mathcal{Z}_{\text{out}^\star} \log \mathcal{Z}_{\text{out}} \right] \quad \text{(Stein Lemma)}$$

$$= \mathbb{E}_{y,\xi} \left[ mq^{-1} \left( \partial_\omega f_{\text{out}^\star} \mathcal{Z}_{\text{out}^\star} \log \mathcal{Z}_{\text{out}} + f_{\text{out}^\star} \partial_\omega \mathcal{Z}_{\text{out}^\star} \log \mathcal{Z}_{\text{out}} \right. \right.$$
$$\left. - \left( \partial_\omega f_{\text{out}^\star} + f_{\text{out}^\star}^2 \right) \mathcal{Z}_{\text{out}^\star} \right) \log \mathcal{Z}_{\text{out}} \right] + \mathbb{E}_{y,\xi} \left[ \mathcal{Z}_{\text{out}^\star} f_{\text{out}^\star} f_{\text{out}} \right]$$

$$= \mathbb{E}_{y,\xi} \left[ \mathcal{Z}_{\text{out}^\star} (.,.,.) f_{\text{out}^\star} \left( y, mq^{-1/2}\xi, \rho_{\text{w}^\star} - mq^{-1}m \right) f_{\text{out}} \left( y, q^{1/2}\xi, Q - q \right) \right]$$
$$\text{(Simplifications with (IV.25))}$$

that leads to

$$\hat{m} = \alpha \partial_m \Psi_{\text{out}}$$
$$= \alpha \mathbb{E}_{y,\xi} \left[ \mathcal{Z}_{\text{out}^\star} (.,.,.) f_{\text{out}^\star} \left( y, mq^{-1/2}\xi, \rho_{\text{w}^\star} - mq^{-1}m \right) f_{\text{out}} \left( y, q^{1/2}\xi, Q - q \right) \right] . \qquad \text{(IV.28)}$$

**Equation over $\hat{Q}$**

$$\partial_Q \Psi_{\text{out}} = \mathbb{E}_{y,\xi} \left[ \mathcal{Z}_{\text{out}^\star} \left( y, mq^{-1/2}\xi, \rho_{\text{w}^\star} - mq^{-1}m \right) \partial_Q \log \mathcal{Z}_{\text{out}} \left( y, q^{1/2}\xi, Q - q \right) \right]$$

$$= \mathbb{E}_{y,\xi} \left[ \mathcal{Z}_{\text{out}^\star} \left( y, mq^{-1/2}\xi, \rho_{\text{w}^\star} - mq^{-1}m \right) \partial_Q V \partial_V \log \mathcal{Z}_{\text{out}} \left( y, q^{1/2}\xi, Q - q \right) \right]$$

$$= \frac{1}{2} \mathbb{E}_{y,\xi} \left[ \mathcal{Z}_{\text{out}^\star} \left( y, mq^{-1/2}\xi, \rho_{\text{w}^\star} - mq^{-1}m \right) \left( \partial_\omega f_{\text{out}} + f_{\text{out}}^2 \right) \left( y, q^{1/2}\xi, Q - q \right) \right]$$

leading to

$$\hat{Q} = -2\alpha\partial_Q\Psi_{\text{out}}$$
$$= -\alpha\mathbb{E}_{y,\xi}\left[\mathcal{Z}_{\text{out}^\star}\left(y, mq^{-1/2}\xi, \rho_{\text{w}^\star} - mq^{-1}m\right)\partial_\omega f_{\text{out}}\left(y, q^{1/2}\xi, Q-q\right)\right] - \hat{q}. \tag{IV.29}$$

**Equation over $q$**

$$\partial_{\hat{q}}\Psi_w = \partial_{\hat{q}}\mathbb{E}_\xi\left[\mathcal{Z}_{\text{w}^\star}\left(\hat{m}\hat{q}^{-1/2}\xi, \hat{m}\hat{q}^{-1}\hat{m}\right)\log\mathcal{Z}_{\text{w}}\left(\hat{q}^{1/2}\xi, \hat{Q}+\hat{q}\right)\right]$$

$$= \mathbb{E}_\xi\left[\partial_{\hat{q}}\omega^\star\partial_\omega\mathcal{Z}_{\text{w}^\star}\log\mathcal{Z}_{\text{w}} + \partial_{\hat{q}}V^\star\partial_V\mathcal{Z}_{\text{w}^\star}\log\mathcal{Z}_{\text{w}} + \frac{\mathcal{Z}_{\text{w}^\star}}{\mathcal{Z}_{\text{w}}}\left(\partial_{\hat{q}}\omega\partial_\omega\mathcal{Z}_{\text{w}} + \partial_{\hat{q}}V\partial_V\mathcal{Z}_{\text{w}}\right)\right]$$

$$= \mathbb{E}_\xi\left[-\frac{\hat{m}}{2}\hat{q}^{-3/2}\xi f_{\text{w}^\star}\mathcal{Z}_{\text{w}^\star}\log\mathcal{Z}_{\text{w}} + \frac{\hat{m}^2\hat{q}^{-2}}{2}\left(\partial_\omega f_{\text{w}^\star} + f_{\text{w}^\star}^2\right)\mathcal{Z}_{\text{w}^\star}\log\mathcal{Z}_{\text{w}}\right.$$

$$\left. + \frac{\mathcal{Z}_{\text{w}^\star}}{\mathcal{Z}_{\text{w}}}\left(\frac{1}{2}\hat{q}^{-1/2}\xi f_{\text{w}}\mathcal{Z}_{\text{w}} - \frac{1}{2}\left(\partial_\omega f_{\text{w}} + f_{\text{w}}^2\right)\mathcal{Z}_{\text{w}}\right)\right]$$

$$= \mathbb{E}_\xi\left[-\frac{\hat{m}}{2}\hat{q}^{-3/2}\partial_\xi\left(f_{\text{w}^\star}\mathcal{Z}_{\text{w}^\star}\log\mathcal{Z}_{\text{w}}\right) + \frac{\hat{m}^2\hat{q}^{-2}}{2}\left(\partial_\omega f_{\text{w}^\star} + f_{\text{w}^\star}^2\right)\mathcal{Z}_{\text{w}^\star}\log\mathcal{Z}_{\text{w}}\right.$$

$$\left. + \left(\frac{1}{2}\hat{q}^{-1/2}\partial_\xi\left(f_{\text{w}}\mathcal{Z}_{\text{w}^\star}\right) - \frac{1}{2}\left(\partial_\omega f_{\text{w}} + f_{\text{w}}^2\right)\mathcal{Z}_{\text{w}^\star}\right)\right] \qquad \text{(Stein lemma)}$$

$$= \frac{1}{2}\mathbb{E}_\xi\left[-\hat{m}^2\hat{q}^{-2}\left(\partial_\omega f_{\text{w}^\star}\mathcal{Z}_{\text{w}^\star}\log\mathcal{Z}_{\text{w}} + \mathcal{Z}_{\text{w}^\star}f_{\text{w}^\star}^2\log\mathcal{Z}_{\text{w}} - \left(\partial_\omega f_{\text{w}^\star} + f_{\text{w}^\star}^2\right)\mathcal{Z}_{\text{w}^\star}\log\mathcal{Z}_{\text{w}}\right)\right.$$

$$\left. -\hat{m}\hat{q}^{-1}\mathcal{Z}_{\text{w}^\star}f_{\text{w}^\star}f_{\text{w}} + \left(\hat{m}\hat{q}^{-1}\mathcal{Z}_{\text{w}^\star}f_{\text{w}}f_{\text{w}^\star} + \mathcal{Z}_{\text{w}^\star}\partial_\omega f_{\text{w}} - \left(\partial_\omega f_{\text{w}} + f_{\text{w}}^2\right)\mathcal{Z}_{\text{w}^\star}\right)\right]$$

$$= -\frac{1}{2}\mathbb{E}_\xi\left[\mathcal{Z}_{\text{w}^\star}\left(\hat{m}\hat{q}^{-1/2}\xi, \hat{m}\hat{q}^{-1}\hat{m}\right)f_{\text{w}}\left(\hat{q}^{1/2}\xi, \hat{Q}+\hat{q}\right)^2\right] \quad \text{(Simplifications with (IV.25))}$$

leading to

$$q = -2\partial_{\hat{q}}\Psi_{\text{w}} = \mathbb{E}_\xi\left[\mathcal{Z}_{\text{w}^\star}\left(\hat{m}\hat{q}^{-1/2}\xi, \hat{m}\hat{q}^{-1}\hat{m}\right)f_{\text{w}}\left(\hat{q}^{1/2}\xi, \hat{q}+\hat{Q}\right)^2\right] \tag{IV.30}$$

**Equation over $m$**

$$\partial_{\hat{m}}\Psi_w = \partial_m\mathbb{E}_\xi\left[\mathcal{Z}_{\text{w}^\star}\left(\hat{m}\hat{q}^{-1/2}\xi, \hat{m}\hat{q}^{-1}\hat{m}\right)\log\mathcal{Z}_{\text{w}}\left(\hat{q}^{1/2}\xi, \hat{Q}+\hat{q}\right)\right]$$

$$= \mathbb{E}_\xi\left[\left(\partial_{\hat{m}}\omega^\star\partial_\omega\mathcal{Z}_{\text{w}^\star} + \partial_{\hat{m}}V^\star\partial_V\mathcal{Z}_{\text{w}^\star}\right)\log\mathcal{Z}_{\text{w}}\right]$$

$$= \mathbb{E}_\xi\left[\left(\hat{q}^{-1/2}\xi f_{\text{w}^\star}\mathcal{Z}_{\text{w}^\star} - \hat{m}\hat{q}^{-1}\left(\partial_\omega f_{\text{w}^\star} + f_{\text{w}^\star}^2\right)\mathcal{Z}_{\text{w}^\star}\right)\log\mathcal{Z}_{\text{w}}\right]$$

$$= \mathbb{E}_\xi\left[\hat{m}\hat{q}^{-1}\partial_\xi\left(f_{\text{w}^\star}\mathcal{Z}_{\text{w}^\star}\log\mathcal{Z}_{\text{w}}\right) - \left(\partial_\omega f_{\text{w}^\star} + f_{\text{w}^\star}^2\right)\mathcal{Z}_{\text{w}^\star}\log\mathcal{Z}_{\text{w}}\right] \qquad \text{(Stein Lemma)}$$

$$= \mathbb{E}_\xi\left[\hat{m}\hat{q}^{-1}\left(\partial_\omega f_{\text{w}^\star}\mathcal{Z}_{\text{w}^\star}\log\mathcal{Z}_{\text{w}} + \mathcal{Z}_{\text{w}^\star}f_{\text{w}^\star}^2\log\mathcal{Z}_{\text{w}} - \left(\partial_\omega f_{\text{w}^\star} + f_{\text{w}^\star}^2\right)\mathcal{Z}_{\text{w}^\star}\log\mathcal{Z}_{\text{w}}\right)\right.$$

$$\left. + \mathcal{Z}_{\text{w}^\star}f_{\text{w}^\star}f_{\text{w}}\right]$$

$$= \mathbb{E}_\xi\left[\mathcal{Z}_{\text{w}^\star}\left(\hat{m}\hat{q}^{-1/2}\xi, \hat{m}\hat{q}^{-1}\hat{m}\right)f_{\text{w}^\star}\left(\hat{m}\hat{q}^{-1/2}\xi, \hat{m}\hat{q}^{-1}\hat{m}\right)f_{\text{w}}\left(\hat{q}^{1/2}\xi, \hat{Q}+\hat{q}\right)\right]$$
$$\text{(Simplifications with (IV.25))}$$

leading to

$$m = 2\partial_{\hat{m}}\Psi_{\text{w}} = \mathbb{E}_\xi\left[\mathcal{Z}_{\text{w}^\star}\left(\hat{m}\hat{q}^{-1/2}\xi, \hat{m}\hat{q}^{-1}\hat{m}\right)f_{\text{w}^\star}\left(\hat{m}\hat{q}^{-1/2}\xi, \hat{m}\hat{q}^{-1}\hat{m}\right)f_{\text{w}}\left(\hat{q}^{1/2}\xi, \hat{q}+\hat{Q}\right)\right]$$
$$\tag{IV.31}$$

**Equation over $Q$**

$$\partial_{\hat{Q}}\Psi_w\left(\hat{Q}, \hat{m}, \hat{q}\right) = \partial_{\hat{Q}}\mathbb{E}_\xi\left[\mathcal{Z}_{\text{w}^\star}\left(\hat{m}\hat{q}^{-1/2}\xi, \hat{m}\hat{q}^{-1}\hat{m}\right)\log\mathcal{Z}_{\text{w}}\left(\hat{q}^{1/2}\xi, \hat{Q}+\hat{q}\right)\right]$$

$$= \mathbb{E}_\xi\left[\mathcal{Z}_{\text{w}^\star}\left(\hat{m}\hat{q}^{-1/2}\xi, \hat{m}\hat{q}^{-1}\hat{m}\right)\frac{1}{\mathcal{Z}_w}\partial_{\hat{Q}}\Lambda\partial_\Lambda\mathcal{Z}_{\text{w}}\left(\hat{q}^{1/2}\xi, \hat{Q}+\hat{q}\right)\right]$$

$$= -\frac{1}{2}\mathbb{E}_\xi\left[\mathcal{Z}_{\text{w}^\star}\left(\hat{m}\hat{q}^{-1/2}\xi, \hat{m}\hat{q}^{-1}\hat{m}\right)\left(\partial_\gamma f_w + f_w^2\right)\right] \qquad \text{(with (IV.25))}$$

hence

$$Q = -2\partial_{\hat{Q}}\Psi_{\mathrm{w}} = \mathbb{E}_{\xi}\left[\mathcal{Z}_{\mathrm{w}^{\star}}\left(\hat{m}\hat{q}^{-1/2}\xi, \hat{m}\hat{q}^{-1}\hat{m}\right)\partial_{\gamma}f_{\mathrm{w}}\left(\hat{q}^{1/2}\xi, \hat{q}+\hat{Q}\right)\right] + q\,. \qquad \text{(IV.32)}$$

# V  Applications

In this section, we provide details of the results presented in Sec. 3. In particular as an illustration, we consider a Gaussian teacher ($\rho_{w^\star} = 1$) with a noiseless sign activation:

$$P_{\text{out}^\star}(y|z) = \delta\left(y - \text{sign}(z)\right) , \qquad\qquad P_{w^\star}(w^\star) = \mathcal{N}_{w^\star}\left(0, \rho_{w^\star}\right) , \qquad (\text{V.1})$$

whose corresponding denoising functions are derived in eq. (I.22) and eq. (I.24).

**Remark V.1.** *Note that performances of ERM with $\ell_2$ regularization for a teacher with Gaussian weights $P_{w^\star}(w) = \mathcal{N}_w(0, 1)$ or binary weights $P_{w^\star}(w) = \frac{1}{2}\left(\delta(w - 1) + \delta(w + 1)\right)$, will be similar. Indeed free entropy terms $\Psi_w$ eq. (IV.16) for a Gaussian prior (IV.22) and for binary weights (IV.24) are equal in this setting, so do the set of fixed point equations.*

## V.1  Bayes-optimal estimation

Using expressions eq. (I.22) and eq. (I.24), corresponding to the *teacher* model eq. (V.1), the prior equation eq. (IV.21) can be simplified while the channel one has no analytical expression. Hence the set of fixed point equations eqs. (IV.23) for the model eq. (V.1) read

$$q_{\text{b}} = \frac{\hat{q}_{\text{b}}}{1 + \hat{q}_{\text{b}}} , \quad \hat{q}_{\text{b}} = \alpha\mathbb{E}_{y,\xi}\left[ \mathcal{Z}_{\text{out}^\star}\left(y, q_{\text{b}}^{1/2}\xi, \rho_{w^\star} - q_{\text{b}}\right) f_{\text{out}^\star}\left(y, q_{\text{b}}^{1/2}\xi, \rho_{w^\star} - q_{\text{b}}\right)^2 \right] . \quad (\text{V.2})$$

**Large $\alpha$ behaviour**   Let us derive the large $\alpha$ behaviour of the Bayes-optimal generalization error eq. (II.7) that depends only on the overlap $q_{\text{b}}$ solution of eq. (V.2). $q_{\text{b}}$ measures the correlation with the ground truth, so we expect that in the limit $\alpha \to \infty$, $q_{\text{b}} \to 1$. Therefore, we need to extract the behaviour of $\hat{q}_{\text{b}}$ in eq. (V.2). Injecting expressions $\mathcal{Z}_{\text{out}^\star}$ and $f_{\text{out}^\star}$ from eq. (I.22), we obtain

$$\hat{q}_{\text{b}} = \alpha\mathbb{E}_{y,\xi}\left[ \mathcal{Z}_{\text{out}^\star}\left(y, q_{\text{b}}^{1/2}\xi, 1 - q_{\text{b}}\right) f_{\text{out}^\star}\left(y, q_{\text{b}}^{1/2}\xi, 1 - q_{\text{b}}\right)^2 \right]$$

$$= 2\alpha\int D\xi y^2 \frac{\mathcal{N}_{\sqrt{q}\xi}(0, 1 - q_{\text{b}})^2}{\frac{1}{2}\left(1 + \text{erf}\left(\frac{\sqrt{q_{\text{b}}}\xi}{\sqrt{2(1-q_{\text{b}})}}\right)\right)} = \frac{2}{\pi}\frac{\alpha}{1 - q_{\text{b}}}\int D\xi \frac{e^{-\frac{q_{\text{b}}\xi^2}{1-q_{\text{b}}}}}{\left(1 + \text{erf}\left(\frac{\sqrt{q_{\text{b}}}\xi}{\sqrt{2(1-q_{\text{b}})}}\right)\right)} ,$$

where the last integral can be computed in the limit $q_{\text{b}} \to 1$:

$$\int D\xi \frac{e^{-\frac{q_{\text{b}}\xi^2}{1-q_{\text{b}}}}}{\left(1 + \text{erf}\left(\frac{\sqrt{q_{\text{b}}}\xi}{\sqrt{2(1-q_{\text{b}})}}\right)\right)} = \int d\xi \frac{\frac{-e^{\frac{\xi^2(q_{\text{b}}+1)}{2(1-q_{\text{b}})}}}{\sqrt{2\pi}}}{\left(1 + \text{erf}\left(\frac{\sqrt{q_{\text{b}}}\xi}{\sqrt{2(1-q_{\text{b}})}}\right)\right)}$$

$$\simeq \int d\xi \frac{\frac{-e^{\frac{\xi^2}{1-q_{\text{b}}}}}{\sqrt{2\pi}}}{\left(1 + \text{erf}\left(\frac{\xi}{\sqrt{2(1-q_{\text{b}})}}\right)\right)} = \frac{\sqrt{1 - q_{\text{b}}}}{\sqrt{2\pi}}\int d\eta \frac{e^{-\eta^2}}{1 + \text{erf}\left(\frac{\eta}{\sqrt{2}}\right)} = \frac{c_0}{\sqrt{2\pi}}\sqrt{1 - q_{\text{b}}} ,$$

with $c_0 \equiv \int d\eta \frac{e^{-\eta^2}}{1+\text{erf}\left(\frac{\eta}{\sqrt{2}}\right)} \simeq 2.83748$. Finally, we obtain in the large $\alpha$ limit:

$$\hat{q}_{\text{b}} = k\frac{\alpha}{\sqrt{1 - q_{\text{b}}}} , \qquad\qquad q_{\text{b}} = \frac{\hat{q}_{\text{b}}}{1 + \hat{q}_{\text{b}}} ,$$

with $k \equiv \frac{2c_0}{\pi\sqrt{2\pi}} \simeq 0.720647$. The above equations can be solved analytically and lead to:

$$q_{\text{b}} = \frac{1}{2}\left(\alpha k\sqrt{\alpha^2 k^2 + 4} - \alpha^2 k^2\right) \underset{\alpha\to\infty}{\simeq} 1 - \frac{1}{\alpha^2 k^2} , \qquad\qquad \hat{q}_{\text{b}} = k^2\alpha^2 ,$$

and therefore the Bayes-optimal asymptotic generalization error is given by

$$e_{\text{g}}^{\text{bayes}}(\alpha) = \frac{1}{\pi}\text{acos}\left(\sqrt{q_{\text{b}}}\right) \underset{\alpha\to\infty}{\simeq} \frac{1}{k\pi}\frac{1}{\alpha} \simeq \frac{0.4417}{\alpha} . \qquad (\text{V.3})$$

## V.2 Generalities on ERM with $\ell_2$ regularization

Combining the teacher update for Gaussian weights eq. (I.24) with the update associated to the $\ell_2$ regularization eq. (I.24), the free entropy term can be explicitly derived in (IV.22). Taking the corresponding derivatives, the fixed point equations for $m, q, \Sigma$ eq. (IV.19) are thus explicit and simply read

$$\Sigma = \frac{1}{\lambda + \hat{\Sigma}}, \qquad q = \frac{\rho_{\mathrm{w}^\star} \hat{m}^2 + \hat{q}}{(\lambda + \hat{\Sigma})^2}, \qquad m = \frac{\rho_{\mathrm{w}^\star} \hat{m}}{\lambda + \hat{\Sigma}}. \tag{V.4}$$

All the following examples have been performed with a $\ell_2$ regularization, so that the above equations (V.4) remain valid for the different losses considered in Sec. 3. In the next subsections, we provide some details on the asymptotic performances of ERM with various losses with $\ell_2$ regularization and $\rho_{\mathrm{w}^\star} = 1$.

In general for a generic loss, the proximal eq. (I.12) has no analytical expression, just as the fixed point equations (IV.20). The square loss is particular in the sense eqs. (IV.20) have a closed form solution. Also the Hinge loss has an analytical proximal. Apart from that, eqs. (IV.20) must be solved numerically. However it is useful to notice that the proximal can be easily found for a two times differentiable loss using eq. (I.29). This is for example the case of the logistic loss.

## V.3 Ridge regression - Square loss with $\ell_2$ regularization

The prior equations over $m, q, \Sigma$ are already derived in eq. (V.4) and remain valid. Combining eq. (I.22) for the considered sign channel with a potential additional Gaussian noise $\Delta^\star$ in (V.1) and the square loss eq. (I.26), the channel fixed point equations for $\hat{q}, \hat{m}, \hat{\Sigma}$ eqs. (IV.20) lead to

$$\Sigma = \frac{1}{\lambda + \hat{\Sigma}}, \qquad\qquad \hat{\Sigma} = \frac{\alpha}{\Sigma + 1},$$

$$q = \frac{\hat{m}^2 + \hat{q}}{(\lambda + \hat{\Sigma})^2}, \qquad\qquad \hat{q} = \alpha \frac{(1 + q + \Delta^\star) - 2\sqrt{\frac{2m^2}{\pi}}}{(\Sigma + 1)^2}, \tag{V.5}$$

$$m = \frac{\hat{m}}{\lambda + \hat{\Sigma}}, \qquad\qquad \hat{m} = \frac{\alpha\sqrt{\frac{2}{\pi}}}{\Sigma + 1}.$$

### V.3.1 Pseudo-inverse estimator

We analyze the fixed point equations eqs. (V.5) for the *pseudo-inverse* estimator, that is in the limit $\lambda \to 0$.

**Solving $\Sigma$** Combining the two first equations over $\Sigma$ and $\hat{\Sigma}$ in (V.5), we obtain

$$\Sigma = \frac{\sqrt{(\alpha + \lambda - 1)^2 + 4\lambda} - \alpha - \lambda + 1}{2\lambda} \underset{\lambda \to 0}{\simeq} \frac{1 - \alpha + |\alpha - 1|}{2\lambda} + \frac{1}{2}\left(\frac{\alpha + 1}{|\alpha - 1|} - 1\right), \tag{V.6}$$

that exhibits two different behaviour depending if $\alpha < 1$ or $\alpha > 1$.

**Regime $\alpha < 1$** In this regime $\alpha < 1$, eq. (V.6) becomes

$$\Sigma = \frac{1 - \alpha}{\lambda} + \frac{\alpha}{1 - \alpha},$$

that leads to the closed set of equations in the limit $\lambda \to 0$

$$\Sigma = \frac{(1 - \alpha)^2 + \alpha\lambda}{\lambda(1 - \alpha)} \underset{\lambda \to 0}{\simeq} \frac{1 - \alpha}{\lambda}, \qquad\qquad \hat{\Sigma} = \frac{(1 - \alpha)\alpha\lambda}{(\alpha - 1)^2 + \lambda} \underset{\lambda \to 0}{\simeq} \frac{\lambda\alpha}{1 - \alpha},$$

$$m = \frac{\alpha(1 - \alpha)}{\lambda + (1 - \alpha)}\sqrt{\frac{2}{\pi}} \underset{\lambda \to 0}{\simeq} \alpha\sqrt{\frac{2}{\pi}}, \qquad\qquad \hat{m} = \frac{\lambda\alpha\sqrt{\frac{2}{\pi}}}{\lambda + (1 - \alpha)} \underset{\lambda \to 0}{\simeq} \frac{\lambda\alpha\sqrt{\frac{2}{\pi}}}{1 - \alpha}, \tag{V.7}$$

$$q \underset{\lambda \to 0}{\simeq} \frac{\alpha(\pi(1 + \Delta^\star) - 2\alpha)}{\pi(1 - \alpha)}, \qquad\qquad \hat{q} \underset{\lambda \to 0}{\simeq} \frac{\alpha\lambda^2(2(\alpha - 2)\alpha + \pi(\Delta^\star + 1))}{\pi(1 - \alpha)(1 - \alpha + \lambda)^2}.$$

Hence we obtain for $\alpha < 1$:

$$m^{\text{pseudo}} = \alpha\sqrt{\frac{2}{\pi}} \qquad\qquad q^{\text{pseudo}} = \frac{\alpha(\pi(1+\Delta^\star) - 2\alpha)}{\pi(1-\alpha)} \qquad\qquad \text{(V.8)}$$

and the corresponding generalization error

$$e_{\text{g}}^{\text{pseudo}}(\alpha) = \frac{1}{\pi}\arccos\left(\sqrt{\frac{2\alpha(1-\alpha)}{\pi(1+\Delta^\star) - 2\alpha}}\right) \text{ if } \alpha < 1. \qquad\qquad \text{(V.9)}$$

Note in particular that $e_{\text{g}}^{\text{pseudo}}(\alpha) \underset{\alpha\to 1}{\longrightarrow} 0.5$, meaning that the interpolation peak at $\alpha = 1$ reaches the maximum generalization error.

**Regime $\alpha > 1$**    Eq. (V.6) becomes

$$\Sigma = \frac{1}{2}\left(\frac{\alpha+1}{\alpha-1} - 1\right) = \frac{1}{2}\left(\frac{\alpha+1}{\alpha-1} - 1\right) = \frac{1}{\alpha-1}.$$

In the limit $\lambda \to 0$, the fixed point equations eqs. (V.5) reduce to

$$\Sigma + 1 = \frac{\alpha}{\alpha-1}, \qquad\qquad \hat{\Sigma} = \alpha - 1,$$

$$q = \frac{(\alpha-1)^2\frac{2}{\pi} + \hat{q}}{(\alpha-1)^2}, \qquad\qquad \hat{q} = \frac{(\alpha-1)^2}{\alpha}\left((1+q+\Delta^\star) - \frac{4}{\pi}\right), \qquad \text{(V.10)}$$

$$m = \sqrt{\frac{2}{\pi}}, \qquad\qquad \hat{m} = (\alpha-1)\sqrt{\frac{2}{\pi}}.$$

In particular we obtain for $\alpha > 1$:

$$m^{\text{pseudo}} = \sqrt{\frac{2}{\pi}}, \qquad\qquad q^{\text{pseudo}} = \frac{1}{\alpha-1}\left(1 + \Delta^\star + \frac{2}{\pi}(\alpha-2)\right), \qquad \text{(V.11)}$$

and the corresponding generalization error

$$e_{\text{g}}^{\text{pseudo}}(\alpha) = \frac{1}{\pi}\arccos\left(\sqrt{\frac{\alpha-1}{\frac{\pi}{2}(1+\Delta^\star) + (\alpha-2)}}\right) \text{ if } \alpha > 1. \qquad\qquad \text{(V.12)}$$

**Large $\alpha$ behaviour**    From this expression we easily obtain the large $\alpha$ behaviour of the pseudo-inverse estimator:

$$e_{\text{g}}^{\text{pseudo}}(\alpha) = \frac{1}{\pi}\arccos\left(\sqrt{\frac{\alpha-1}{\frac{\pi}{2}(1+\Delta^\star) + (\alpha-2)}}\right) = \frac{1}{\pi}\arccos\left(\left(1 + \frac{C}{\alpha-1}\right)^{1/2}\right) \underset{\alpha\to\infty}{\simeq} \frac{c}{\sqrt{\alpha}}$$

where $C = \frac{\pi}{2}(1+\Delta^\star) - 1$ and $c = \frac{\sqrt{C}}{\pi}$. In particular for a noiseless teacher $\Delta^\star = 0$, $c = \sqrt{\frac{\pi-2}{2\pi^2}} \simeq 0.240487$, leading to

$$e_{\text{g}}^{\text{pseudo}}(\alpha) \underset{\alpha\to\infty}{\simeq} \frac{0.2405}{\sqrt{\alpha}}. \qquad\qquad \text{(V.13)}$$

### V.3.2    Ridge at finite $\lambda$

Let us now consider the set of fixed point equation eq. (V.5) for finite $\lambda \neq 0$. Defining

$$t_0 \equiv \sqrt{(\alpha+\lambda-1)^2 + 4\lambda}$$
$$t_1 \equiv (t_0 + \alpha + \lambda + 1)^{-1}$$
$$t_2 \equiv \sqrt{2(\alpha+1)\lambda + (\alpha-1)^2 + \lambda^2}$$
$$t_3 \equiv (t_2 + \alpha + \lambda + 1)^{-1}$$
$$t_4 \equiv \sqrt{\alpha^2 + 2\alpha(\lambda-1) + (\lambda+1)^2},$$

the equations can be in fact fully solved analytically and read

$$\Sigma = \frac{1}{2} \frac{t_0 - \alpha - \lambda + 1}{\lambda}$$

$$\hat{\Sigma} = \frac{1}{2}\left(t_0 + \alpha - \lambda - 1\right)$$

$$q = \frac{2\alpha\left(-8\alpha^2 t_1 + 2\alpha + \pi\Delta^\star + \pi\right)}{\pi\left(\alpha^2 + \alpha\left(t_2 + 2\lambda - 2\right) + \left(\lambda + 1\right)\left(t_2 + \lambda + 1\right)\right)}\,,$$

$$\hat{q} = \left(4\alpha\lambda^2\left(\pi(\Delta^\star + 1)\left(t_4 + (\alpha + \lambda)\left(t_2 + \alpha + \lambda\right) + 2\lambda + 1\right)\right.\right.$$
$$\left.\left. - 8\alpha t_3\left(t_4 + (\alpha + \lambda)\left(\sqrt{2(\alpha + 1)\lambda + (\alpha - 1)^2 + \lambda^2} + \alpha + \lambda\right) + 2\lambda\right) - 8\alpha t_3 + 4\alpha^2\right)\right),$$

$$m = \frac{2\sqrt{\frac{2}{\pi}}\alpha}{t_2 + \alpha + \lambda + 1}\,,$$

$$\hat{m} = \frac{2\sqrt{\frac{2}{\pi}}\alpha\lambda}{t_0 - \alpha + \lambda + 1}\,.$$

**Generalization error behaviour at large $\alpha$**    Expanding the ratio $\frac{m}{\sqrt{q}}$ in the large $\alpha$ limit, we obtain

$$\frac{m}{\sqrt{q}} \simeq 1 - \frac{C}{2\alpha} \text{ with } C = \frac{\pi}{2}\left(1 + \Delta^\star\right) - 1$$

leading to

$$e_{\mathrm{g}}^{\mathrm{ridge},\lambda}\left(\alpha\right) = \frac{1}{\pi}\arccos\left(\frac{m}{\sqrt{q}}\right) \underset{\alpha \to \infty}{\simeq} \frac{c}{\sqrt{\alpha}} \text{ with } c = \frac{\sqrt{C}}{\pi}\,. \tag{V.14}$$

Thus, the asymptotic generalization error for ridge regression with any regularization strength $\lambda \geq 0$ decrease as $\frac{0.2405}{\sqrt{\alpha}}$, similarly to the pseudo-inverse result.

**Optimal regularization**    The optimal value $\lambda^{\mathrm{opt}}(\alpha)$, introduced in Sec. 3, which minimizes the generalization error at a given $\alpha$ can be found taking the derivative of $\frac{m}{\sqrt{q}}$ and is written as the root of the following functional

$$F[\alpha, \lambda, \Delta^\star] = \partial_\lambda\left(\frac{m}{\sqrt{q}}\right) = \frac{a_1 a_2}{a_3 a_4^2}\,,$$
$$\text{with}$$
$$a_1 = -4\alpha\sqrt{\frac{a_4}{\alpha^2 + \alpha\left(t_2 + 2\lambda - 2\right) + \left(\lambda + 1\right)\left(t_2 + \lambda + 1\right)}}\,,$$
$$a_2 = 2\left(\alpha^2 t_3 + \alpha\left(2\lambda t_3 + \left(t_2 + 2\right)t_3 - 1\right) + \left(\lambda + 1\right)\left(t_2 + \lambda + 1\right)t_3\right) - \pi(1 + \Delta^\star)\,,$$
$$a_3 = \frac{t_0}{t_1}\,,$$
$$a_4 = \alpha\left(2 - 8t_1\right) + \pi\left(1 + \Delta^\star\right)\,.$$

Unfortunately, this functional cannot be analyzed analytically. Instead we plot its value for a wide range of $\alpha$ as a function of $\lambda$ (for $\Delta^\star = 0$) and we observe in particular that there exists a unique value $\lambda^{\mathrm{opt}} \simeq 0.570796$ as illustrated in Fig. 1 (**left**) that is independent of $\alpha$. As an illustration, we show the generalization error of ridge regression with the optimal regularization $\lambda^{\mathrm{opt}} = 0.5708$ compared to the Bayes-optimal performances in Fig. 1 (**right**).

Figure 1: (**Left**) Absolute value of the derivative of $m/\sqrt{q}$ with respect to $\lambda$ plotted in a logarithmic scale. $\lambda^{\mathrm{opt}}$ is reached at the root of the functional $F[\alpha, \lambda]$ that corresponds to the divergence in the logarithmic scale. Plotted for a wide range of $\alpha$, the optimal value is clearly constant and independent of $\alpha$. Its value is approximately $\lambda^{\mathrm{opt}} \simeq 0.570796$. (**Right**) Bayes-optimal (black) vs ridge regression (dashed red) generalization errors with optimal $\ell_2$ regularization $\lambda^{\mathrm{opt}} \simeq 0.570796$.

## V.4 Hinge regression / SVM - Hinge loss with $\ell_2$ regularization

The hinge loss $l^{\mathrm{hinge}}(y, z) = \max(0, 1 - yz)$ is linear by part and is therefore another simple example of analytical loss to analyze. In particular its proximal map can computed in eq. (I.27) and the corresponding denoising functions read:

$$
f_{\mathrm{out}}\left(y, q^{1/2}\xi, \Sigma\right) = \begin{cases} y \text{ if } \xi y < \frac{1-\Sigma}{\sqrt{q}} \\ \\ \frac{y-\sqrt{q}\xi}{\Sigma} \text{ if } \frac{1-\Sigma}{\sqrt{q}} < \xi y < \frac{1}{\sqrt{q}} \\ \\ 0 \text{ otherwise} \end{cases},
$$

$$
\partial_\omega f_{\mathrm{out}}\left(y, q^{1/2}\xi, \Sigma\right) = \begin{cases} -\frac{1}{\Sigma} \text{ if } \frac{1-\Sigma}{\sqrt{q}} < \xi y < \frac{1}{\sqrt{q}} \\ \\ 0 \text{ otherwise} \end{cases}.
$$

(V.15)

The fixed point equations eq. (IV.20) have unfortunately no closed form and need to be solved numerically.

### V.4.1 Max-margin estimator

As proven in [7] both the hinge and logistic estimators converge to the *max-margin* solution in the limit $\lambda \to 0$ as soon as the data are linearly separable. We will start with the fixed point equations for hinge, whose denoising functions (V.15) are analytical. Taking the $\lambda \to 0$ limit is non-trivial and we need therefore to introduce some rescaled variables to obtain a closed set of equations. Numerical evidences at finite $\alpha$ show that we shall use the following rescaled variables:

$$\hat{m} = \Theta\left(\lambda\right), \quad \hat{q} = \Theta\left(\lambda^2\right), \quad \hat{\Sigma} = \Theta\left(\lambda\right), \quad m = \Theta(1), \quad q = \Theta(1), \quad \Sigma = \Theta\left(\lambda^{-1}\right).$$

The fixed point equations eq. (IV.20) simplify and become

$$m = \frac{\hat{m}}{1 + \hat{\Sigma}}, \qquad q = \frac{\hat{m}^2 + \hat{q}}{(1 + \hat{\Sigma})^2}, \qquad \Sigma = \frac{1}{1 + \hat{\Sigma}},$$

$$\hat{m} = \frac{2\alpha}{\Sigma}\mathcal{I}_{\hat{m}}(q, \eta), \quad \hat{q} = \frac{2\alpha}{\Sigma^2}\mathcal{I}_{\hat{q}}(q, \eta), \quad \hat{\Sigma} = \frac{2\alpha}{\Sigma}\mathcal{I}_{\hat{\Sigma}}(q, \eta),$$

(V.16)

with

$$\mathcal{I}_{\hat{m}}(q, \eta) \equiv \int_{-\infty}^{\frac{1}{\sqrt{q}}} \mathrm{d}\xi \mathcal{N}_\xi(0, 1)\mathcal{N}_\xi\left(0, \frac{1 - \eta}{\sqrt{\eta}}\right)(1 - \sqrt{q}\xi),$$

$$= \frac{\sqrt{2\pi}\left(\mathrm{erf}\left(\frac{1}{\sqrt{2}\sqrt{q(1-\eta)}}\right) + 1\right) + 2e^{-\frac{1}{2q(1-\eta)}}\sqrt{q(1-\eta)}}{4\pi}$$

(V.17)

$$\mathcal{I}_{\hat{q}}(q, \eta) \equiv \int_{-\infty}^{\frac{1}{\sqrt{q}}} \mathrm{d}\xi \mathcal{N}_\xi(0, 1)\frac{1}{2}\left(1 + \mathrm{erf}\left(\frac{\sqrt{\eta}\xi}{\sqrt{2(1-\eta)}}\right)\right)(1 - \sqrt{q}\xi)^2,$$

$$\mathcal{I}_{\hat{\Sigma}}(q, \eta) \equiv \int_{-\infty}^{\frac{1}{\sqrt{q}}} \mathrm{d}\xi \mathcal{N}_\xi(0, 1)\frac{1}{2}\left(1 + \mathrm{erf}\left(\frac{\sqrt{\eta}\xi}{\sqrt{2(1-\eta)}}\right)\right).$$

**Large $\alpha$ expansion**   Numerically at large $\alpha$ (and $\lambda \to 0$), we obtain the following scalings

$$q = \Theta(\alpha^2), \quad m = \Theta(\alpha), \quad \Sigma = \Theta(1), \quad \hat{q} = \Theta(1), \quad \hat{m} = \Theta(\alpha), \quad \hat{\Sigma} = \Theta(1).$$

(V.18)

Therefore, in order to close the equations, we introduce new variables $(c_q, c_\eta)$ such that

$$q \underset{\alpha \to \infty}{=} c_q \alpha^2, \qquad\qquad \eta = 1 - \frac{c_\eta}{\alpha^2}.$$

(V.19)

In this limit, we can extract the large $\alpha$ behaviours of integrals $\mathcal{I}_{\hat{m}}, \mathcal{I}_{\hat{q}}, \mathcal{I}_{\hat{\Sigma}}$:

$$\mathcal{I}_{\hat{m}}(q, \eta) = \mathcal{I}_{\hat{m}}^\infty(c_q, c_\eta), \quad \mathcal{I}_{\hat{q}}(q, \eta) = \frac{\mathcal{I}_{\hat{q}}^\infty(c_q, c_\eta)}{\alpha}, \quad \mathcal{I}_{\hat{\Sigma}}(q, \eta) = \frac{\mathcal{I}_{\hat{\Sigma}}^\infty(c_q, c_\eta)}{\alpha},$$

(V.20)

where $\mathcal{I}_{\hat{m}}^\infty, \mathcal{I}_{\hat{q}}^\infty, \mathcal{I}_{\hat{\Sigma}}^\infty$ are $\Theta(1)$ and read

$$\mathcal{I}_{\hat{m}}^\infty(c_q, c_\eta) \equiv \frac{\sqrt{2\pi}\left(\mathrm{erf}\left(\frac{1}{\sqrt{2}\sqrt{c_\eta c_q}}\right) + 1\right) + 2e^{-\frac{1}{2c_\eta c_q}}\sqrt{c_\eta c_q}}{4\pi},$$

$$\mathcal{I}_{\hat{q}}^\infty(c_q, c_\eta) \equiv \frac{e^{-\frac{1}{2c_\eta c_q}}\left(\sqrt{2\pi}(3c_\eta c_q + 1)e^{\frac{1}{2c_\eta c_q}}\left(\mathrm{erf}\left(\frac{1}{\sqrt{2}\sqrt{c_\eta c_q}}\right) + 1\right) + 4(c_\eta c_q)^{3/2} + 2\sqrt{c_\eta c_q}\right)}{12\pi\sqrt{c_q}},$$

$$\mathcal{I}_{\hat{\Sigma}}^\infty(c_q, c_\eta) \equiv \frac{\sqrt{2\pi}\left(\mathrm{erf}\left(\frac{1}{\sqrt{2}\sqrt{c_\eta c_q}}\right) + 1\right) + 2e^{-\frac{1}{2c_\eta c_q}}\sqrt{c_\eta c_q}}{4\pi\sqrt{c_q}}.$$

(V.21)

Hence the set of fixed-point equations eq. (V.16) simplifies to:

$$
\hat{\Sigma} = \frac{2\mathcal{I}_{\hat{\Sigma}}^{\infty}(c_q, c_\eta)}{1 - 2\mathcal{I}_{\hat{\Sigma}}^{\infty}(c_q, c_\eta)}\,, \qquad \Sigma = 1 - 2\mathcal{I}_{\hat{\Sigma}}^{\infty}(c_q, c_\eta)
$$

$$
\hat{m} = \frac{2\alpha\mathcal{I}_{\hat{m}}^{\infty}(c_q, c_\eta)}{1 - 2\mathcal{I}_{\hat{\Sigma}}^{\infty}(c_q, c_\eta)}\,, \qquad m = 2\alpha\mathcal{I}_{\hat{m}}^{\infty}(c_q, c_\eta) \tag{V.22}
$$

$$
\hat{q} = \frac{2\mathcal{I}_{\hat{q}}^{\infty}(c_q, c_\eta)}{\left(1 - 2\mathcal{I}_{\hat{\Sigma}}^{\infty}(c_q, c_\eta)\right)^2}\,, \quad q = 4\alpha^2\left(\mathcal{I}_{\hat{m}}^{\infty}(c_q, c_\eta)\right)^2 + 2\mathcal{I}_{\hat{q}}^{\infty}(c_q, c_\eta)\,,
$$

which can be closed by rewriting the equations eqs. (V.19):

$$
\eta = \frac{m^2}{q} \equiv 1 - \frac{c_\eta}{\alpha^2} = 1 - \frac{\mathcal{I}_{\hat{q}}^{\infty}(c_q, c_\eta)}{2\left(\mathcal{I}_{\hat{m}}^{\infty}(c_q, c_\eta)\right)^2}\frac{1}{\alpha^2}\,, \tag{V.23}
$$

$$
q = c_q\alpha^2 \simeq 4\alpha^2\left(\mathcal{I}_{\hat{m}}^{\infty}(c_q, c_\eta)\right)^2\,.
$$

Equivalently $(c_q^\star, c_\eta^\star)$ is the root of the set of non-linear fixed point equations $(F_\eta(c_q, c_\eta), F_q(c_q, c_\eta))$:

$$
F_\eta(c_q, c_\eta) \equiv \frac{\mathcal{I}_{\hat{q}}^{\infty}(c_q, c_\eta)}{2\left(\mathcal{I}_{\hat{m}}^{\infty}(c_q, c_\eta)\right)^2} - c_\eta\,, \qquad F_q(c_q, c_\eta) \equiv 4\left(\mathcal{I}_{\hat{m}}^{\infty}(c_q, c_\eta)\right)^2 - c_q\,, \tag{V.24}
$$

that cannot be solved analytically. However a unique numerical solution is found and lead to $(c_q^\star, c_\eta^\star) = (0.9911, 2.4722)$. Therefore the generalization error of the max-margin estimator in the large $\alpha$ regime is given by

$$
e_{\mathrm{g}}^{\mathrm{max-margin}}(\alpha) = \frac{1}{\pi}\arccos\left(\frac{m}{\sqrt{q}}\right) \underset{\alpha\to\infty}{\simeq} \frac{1}{\pi}\arccos\left(1 - \frac{c_\eta^\star}{\alpha^2}\right) \underset{\alpha\to\infty}{\simeq} \frac{K}{\alpha}\,, \tag{V.25}
$$

with $K = \frac{\sqrt{c_\eta^\star}}{\pi} \simeq 0.5005$, leading to

$$
e_{\mathrm{g}}^{\mathrm{max-margin}}(\alpha) \underset{\alpha\to\infty}{\simeq} \frac{0.5005}{\alpha}\,. \tag{V.26}
$$

## V.5 Logistic regression

The logistic loss is a combination of the cross entropy loss $l(y, z) = -y\log(\sigma(z)) - (1 - y)\log(1 - \sigma(z))$ with as sigmoid activation function $\sigma$, that simplifies for binary labels $y \pm 1$ to $l^{\mathrm{logistic}}(y, z) = \log(1 + \exp(-yz))$ with the two first derivatives given by

$$
\partial_z l^{\mathrm{logistic}}(y, z) = -\frac{y}{e^{zy} + 1}\,, \qquad \partial_z^2 l^{\mathrm{logistic}}(y, z) = \frac{y^2}{2(1 + \cosh(zy))} = \frac{y^2}{4\cosh\left(\frac{yz}{2}\right)}\,.
$$

Its proximal is not analytical, but it can be written as the solution of the implicit equation (I.28) providing the corresponding denoising functions (I.29). Solving the fixed point equations (IV.20), we obtain performances that approach closely the Bayes-optimal baseline as illustrated in Fig. 2 (**left**).

## V.6 Logistic with non-linearly separable data - A rectangle door teacher

The analysis of ERM for the linearly separable dataset generated by (V.1) reveals that logistic regression with $\ell_2$ regularization was able to approach very closely Bayes-optimal error. Therefore it seems us very interesting to investigate if logistic regression could perform as well on a more complicated non-linearly separable dataset obtained by a *rectangle door* channel

$$
\mathbf{y} = \mathrm{sign}\left(\left|\frac{1}{\sqrt{d}}\mathbf{X}\mathbf{w}^\star\right| - \kappa\right)\,. \tag{V.27}
$$

This channel has been already considered in [1] and we fix the width of the door to $\kappa = 0.6745$ to obtain labels $\pm 1$ with probability $0.5$. We then compare the ERM performances of logistic regression with $\ell_2$ regularization to the Bayes-optimal performances given by (V.2) with denoising functions derived in eq. (I.23). We show in Fig. 2 (**right**) the comparison only for an arbitrary hyper-parameter

Figure 2: (**Left**) Logistic regression - Generalization error as a function of $\alpha$ for different regularizations strength $\lambda$. Decreasing $\lambda$, the generalization error approaches very closely the Bayes-optimal error (black line). The difference with the Bayes error is shown as an inset. Logistic flirts with Bayes error but never achieves it exactly. The asymptotic behaviour is compared to numerical logistic regression with $d = 10^3$ and averaged over $n_s = 20$ samples, performed with the default method *LogisticRegression* of the `scikit-learn` package [8]. (**Right**) Rectangle door teacher with $\kappa = 0.6745$ - Bayes-optimal generalization error (black) compared to asymptotic generalization performances of $\ell_2$ logistic regression (dashed yellow line) and numerical ERM (crosses).

$\lambda = 10^{-2}$, as results are similar for any regularization. As we might expect, the logistic regression is not able to reach the Bayes-optimal generalization error. Both Bayes-optimal and ERM performances are stuck in the symmetric fixed point $m = 0$ up to $\alpha_{it} \simeq 1.393$. Above this threshold it becomes unstable and Bayes error decreases to zero in the $\alpha \to 0$ limit, while the logistic regression with arbitrary $\lambda$ remains stuck to its maximal generalization error, meaning that in this non-linearly separable case, the logistic regression largely underperforms Bayes-optimal performances.

## VI Reaching Bayes optimality

In this section, we propose a derivation inspired by [9–18] of the fine-tuned loss and regularizer (17) discussed in Sec. 4. We assume that the dataset is generated by a teacher (I.1) such that $\mathcal{Z}_{\mathrm{out}^\star}(., \omega, .)$ and $\mathcal{Z}_{\mathrm{w}^\star}(\gamma, .)$ are respectively log-concave in $\omega$ and $\gamma$. The derivation is based on the GAMP algorithm introduced in [2] for the model eq. (1), that we start by recalling.

### VI.1 Generalized Approximate Message Passing (GAMP) algorithm

The GAMP algorithm can be written as the following set of iterative equations that depend on the update functions (I.6):

$$
\begin{cases}
\hat{\mathbf{w}}^{t+1} = f_{\mathrm{w}}(\boldsymbol{\gamma}^t, \Lambda^t) \\[4pt]
\hat{\mathbf{c}}_{\mathrm{w}}^{t+1} = \partial_\gamma f_{\mathrm{w}}(\boldsymbol{\gamma}^t, \Lambda^t) \\[4pt]
\mathbf{f}_{\mathrm{out}}^t = f_{\mathrm{out}}(y, \boldsymbol{\omega}^t, V^t)
\end{cases}
\quad \text{and} \quad
\begin{cases}
\Lambda_i^t = -\frac{1}{d} \sum_{\mu=1}^n \mathrm{X}_{\mu i}^2 \partial_\omega f_{\mathrm{out}, \mu}^t \\[4pt]
\gamma_i^t = \frac{1}{\sqrt{d}} \sum_{\mu=1}^n \mathrm{X}_{\mu i} f_{\mathrm{out}, \mu}^t + \Lambda_i^t \hat{w}_i^t \\[4pt]
V_\mu^t = \frac{1}{d} \sum_{i=1}^d \mathrm{X}_{\mu i}^2 \hat{c}_{w,i}^t \\[4pt]
\omega_\mu^t = \frac{1}{\sqrt{d}} \sum_{i=1}^d \mathrm{X}_{\mu i} \hat{w}_i^t - V_\mu^t f_{\mathrm{out}, \mu}^{t-1}
\end{cases}.
\tag{VI.1}
$$

It has been proven in [19] that the GAMP algorithm with Bayes-optimal update functions $f_{\mathrm{w}} = f_{\mathrm{w}^\star}$ and $f_{\mathrm{out}} = f_{\mathrm{out}^\star}$ (I.8) converges to the Bayes-optimal performances in the large size limit. Yet the GAMP denoising functions are generic and can be chosen as will depending on the statistical estimation method. In particular we may choose the denoising functions for Bayes-optimal estimation (I.8) or the ones corresponding to ERM estimation (I.12)

$$
\begin{aligned}
f_{\mathrm{w}}^{\mathrm{bayes}}(\gamma, \Lambda) &= \partial_\gamma \log\left(\mathcal{Z}_{\mathrm{w}^\star}\right), \\
f_{\mathrm{out}}^{\mathrm{bayes}}(y, \omega, V) &= \partial_\omega \log\left(\mathcal{Z}_{\mathrm{out}^\star}\right), \\
f_{\mathrm{w}}^{\mathrm{erm}, r}(\gamma, \Lambda) &= \Lambda^{-1}\gamma - \Lambda^{-1}\partial_{\Lambda^{-1}\gamma} \mathcal{M}_{\Lambda^{-1}}\left[r(.)\right]\left(\Lambda^{-1}\gamma\right), \\
f_{\mathrm{out}}^{\mathrm{erm}, l}(y, \omega, V) &= -\partial_\omega \mathcal{M}_V[l(y, .)](\omega),
\end{aligned}
\tag{VI.2}
$$

whose corresponding GAMP algorithms (VI.1) will achieve potentially different fixed points and thus different performances. As it is proven that GAMP with Bayes-optimal updates lead to the optimal generalization error, so that ERM matches the same performances it is sufficient to enforce that at each time step $t$ the Bayes-optimal and ERM denoising functions are equal $f^{\mathrm{bayes}} = f^{\mathrm{erm}}$. Enforcing these two constraints will lead to the expressions for the optimal loss $l^{\mathrm{opt}}$ and regularizer $r^{\mathrm{opt}}$, so that ERM matches Bayes-optimal performances.

### VI.2 Matching Bayes-optimal and ERM performances

Imposing the equality on the channel updates we obtain

$$
f_{\mathrm{out}}^{\mathrm{bayes}}(y, \omega, V) = f_{\mathrm{out}}^{\mathrm{erm}, l}(y, \omega, V) \Leftrightarrow \partial_\omega \log\left(\mathcal{Z}_{\mathrm{out}^\star}\right)(y, \omega, V) = -\partial_\omega \mathcal{M}_V\left[l^{\mathrm{opt}}(y, .)\right](\omega).
$$

Integrating, leaving aside the constant that will not influence the final result, and taking the Moreau-Yosida regularization on both sides, we obtain:

$$
\mathcal{M}_V\left[\log \mathcal{Z}_{\mathrm{out}^\star}(y, ., V)\right](\omega) = \mathcal{M}_V\left[-\mathcal{M}_V\left[l^{\mathrm{opt}}(y, .)\right](\omega)\right] = -l^{\mathrm{opt}}(y, \omega),
$$

where we invert the Moreau-Yosida regularization in the last equality that is valid as long as $\mathcal{Z}_{\mathrm{out}^\star}(y, \omega, V)$ is assumed to be log-concave in $\omega$, (see [11] for a derivation). We finally obtain

$$
l^{\mathrm{opt}}(y, z) = -\mathcal{M}_V\left[\log\left(\mathcal{Z}_{\mathrm{out}^\star}\right)(y, ., V)\right](z) = -\min_\omega\left(\frac{(z-\omega)^2}{2V} + \log \mathcal{Z}_{\mathrm{out}^\star}(y, \omega, V)\right).
\tag{VI.3}
$$

Let us perform the same computation for the prior updates. First we introduce a rescaled denoising distribution:

$$
\begin{aligned}
\tilde{Q}_{\mathrm{w}^\star}(w; \gamma, \Lambda) &\equiv \frac{1}{\tilde{\mathcal{Z}}_{\mathrm{w}^\star}(\gamma, \Lambda)} P_{\mathrm{w}^\star}(w) e^{-\frac{1}{2}\Lambda\left(w - \Lambda^{-1}\gamma\right)^2}, \\
\log\left(\tilde{\mathcal{Z}}_{\mathrm{w}^\star}(\gamma, \Lambda)\right) &= \log\left(\mathcal{Z}_{\mathrm{w}^\star}(\gamma, \Lambda)\right) - \frac{1}{2}\Lambda^{-1}\gamma^2,
\end{aligned}
\tag{VI.4}
$$

so that the the prior updates read

$$
\begin{aligned}
f_{\mathrm{w}}^{\mathrm{bayes}}\left(\gamma, \Lambda\right) &= \partial_{\gamma} \log\left(\mathcal{Z}_{\mathrm{w}^{\star}}\right) = \Lambda^{-1}\gamma + \Lambda^{-1}\partial_{\Lambda^{-1}\gamma} \log\left(\tilde{\mathcal{Z}}_{\mathrm{w}^{\star}}\right), \\
f_{\mathrm{w}}^{\mathrm{erm},r}\left(\gamma, \Lambda\right) &= \mathcal{P}_{\Lambda^{-1}}\left[r\right]\left(\Lambda^{-1}\gamma\right) = \Lambda^{-1}\gamma - \Lambda^{-1}\partial_{\Lambda^{-1}\gamma}\mathcal{M}_{\Lambda^{-1}}\left[r\right]\left(\Lambda^{-1}\gamma\right).
\end{aligned}
\tag{VI.5}
$$

Imposing the equivalence of the Bayes-optimal and ERM prior update,

$$
f_{\mathrm{w}}^{\mathrm{bayes}}\left(\gamma, \Lambda\right) = f_{\mathrm{w}}^{\mathrm{erm},r}\left(\gamma, \Lambda\right) \Leftrightarrow \partial_{\Lambda^{-1}\gamma} \log\left(\tilde{\mathcal{Z}}_{\mathrm{w}^{\star}}\right) = -\partial_{\Lambda^{-1}\gamma}\mathcal{M}_{\Lambda^{-1}}\left[r^{\mathrm{opt}}\right]\left(\Lambda^{-1}\gamma\right), \tag{VI.6}
$$

and assuming that $\mathcal{Z}_{\mathrm{w}}(\gamma, \Lambda)$ is log-concave in $\gamma$, we may invert the Moreau-Yosida regularization, that leads to:

$$
\begin{aligned}
r^{\mathrm{opt}}\left(\Lambda^{-1}\gamma\right) &= -\mathcal{M}_{\Lambda^{-1}}\left[\log\left(\tilde{\mathcal{Z}}_{\mathrm{w}^{\star}}\right)\left(.,\Lambda^{-1}\right)\right](w) \\
&= -\min_{\Lambda^{-1}\gamma}\left(\frac{(w-\Lambda^{-1}\gamma)^2}{2\Lambda^{-1}} + \log\tilde{\mathcal{Z}}_{\mathrm{w}^{\star}}\left(\gamma, \Lambda\right)\right) = -\min_{\gamma}\left(\frac{1}{2}\Lambda w^2 - \gamma w + \log\mathcal{Z}_{\mathrm{w}^{\star}}\left(\gamma, \Lambda\right)\right).
\end{aligned}
\tag{VI.7}
$$

The last step, is to characterize the variances $V$ and $\Lambda$ involved in (VI.3) and (VI.7) that are so far undetermined. To achieve the Bayes-optimal performances, we therefore need to use the variances $V$ and $\Lambda$ solutions of the Bayes-optimal GAMP algorithm (VI.1). In the large size limit, these quantities concentrate and are given by the State Evolution (SE) of the GAMP algorithm, that we recall herein.

**State evolution of GAMP**   In the large size limit, the expectation of the parameter $V$ and $\Lambda$ over the ground truth $\mathbf{w}^{\star}$ and the input data X lead to [19]:

$$
\mathbb{E}_{\mathbf{w}^{\star},\mathrm{X}}\left[V\right] = \rho_{\mathrm{w}^{\star}} - q_{\mathrm{b}}, \qquad\qquad \mathbb{E}_{\mathbf{w}^{\star},\mathrm{X}}\left[\Lambda\right] = \hat{q}_{\mathrm{b}}, \tag{VI.8}
$$

where $q_{\mathrm{b}}$ and $\hat{q}_{\mathrm{b}}$ are solutions of the Bayes-optimal set of fixed point equations eq. (13).

## VI.3   Summary and numerical evidences

Choosing the fine-tuned (potentially non-convex depending on $\mathcal{Z}_{\mathrm{out}^{\star}}$ and $\mathcal{Z}_{\mathrm{w}^{\star}}$) loss and regularizer

$$
\begin{aligned}
l^{\mathrm{opt}}\left(y, z\right) &= -\min_{\omega}\left(\frac{(z-\omega)^2}{2(\rho_{\mathrm{w}^{\star}} - q_{\mathrm{b}})} + \log\mathcal{Z}_{\mathrm{out}^{\star}}\left(y, \omega, \rho_{\mathrm{w}^{\star}} - q_{\mathrm{b}}\right)\right) \\
r^{\mathrm{opt}}\left(w\right) &= -\min_{\gamma}\left(\frac{1}{2}\hat{q}_{\mathrm{b}}w^2 - \gamma w + \log\mathcal{Z}_{\mathrm{w}^{\star}}\left(\gamma, \hat{q}_{\mathrm{b}}\right)\right)
\end{aligned}
\tag{VI.9}
$$

with $q_{\mathrm{b}}$ and $\hat{q}_{\mathrm{b}}$ are solutions of the Bayes-optimal set of fixed point equations eq. (13), we showed that ERM can provably match the Bayes-optimal performances. In particular we illustrated the behaviour of the optimal loss and regularizer $\lambda^{\mathrm{opt}}$ and $r^{\mathrm{opt}}$ for the model (2) in Fig. 2 of the main text. Note in particular that even though the loss $l^{\mathrm{opt}}$ is not convex (but seems quasi-convex), numerical simulations of ERM with (VI.9) (black dots) presented in Fig. 3 show that ERM achieves indeed the Bayes-optimal performances (black line) even at finite dimension.

Figure 3: Generalization error obtained by optimization of the optimal loss $l^{\mathrm{opt}}$ and $r^{\mathrm{opt}}$ for the model (2), compared to $\ell_2$ logistic regression and Bayes-optimal performances. Numerics has been performed with `scipy.optimize.minimize` with the `L-BFGS-B` solver for $d = 10^3$ and averaged over $n_s = 10$ instances. The error bars are barely visible.