[Reviews · NeurIPS 2020]

Review 1

Summary and Contributions: 1) derived analytic expression (using CGMT) of the generalization error of linear classifiers in learning a single-index model with $\ell_2$ regularization; 2) evaluated various commonly-used losses and compared with the Bayes-optimal baseline; 3) determined the Bayes-optimal loss and regularization strength.

Strengths: The is a solid paper with a number of interesting findings. The theoretical analysis is complete and the results are nicely presented. I find the high-dimensional characterization of different loss functions interesting, and I believe the results will be relevant to the community (since the double descent phenomenon has attracted much attention).

Weaknesses: The setup (unit Gaussian data; binary model) is a bit limited, and the tool (CGMT) is relatively well-known. For additional comments see the questions section.

Correctness: To my knowledge yes.

Clarity: Yes.

Relation to Prior Work: The original Gordon's comparison theorem is not cited. In addition, for the double descent literature, I'm not sure why the "Dynamics of Generalization in Linear Perceptrons" paper is cited, instead of the 1992 paper "A Simple Weight Decay Can Improve Generalization".

Reproducibility: Yes

Additional Feedback: I have the following questions for the authors: 1. Is there a general criteria that tells me if a particular loss function achieves the 1/n fast rate for this problem? 2. to what extent does the analysis of generalization error extend beyond isotropic Gaussian data and $\ell_2$ regularization? My impression is that some rotation invariance is required in the current derivation, but previous CGMT-type analysis is also able to handle general covariances. 3. it is known that the SVM objective may also achieve a fast rate of 1/n, for instance: Sridharan, Karthik, Shai Shalev-Shwartz, and Nathan Srebro. "Fast rates for regularized objectives." Advances in neural information processing systems. 2009. Can the authors comment on the difference? 4. (minor) out of curiosity, my impression is the system of equations that describes the generalization error can often be derived by both AMP and CGMT. Is there a particular reason that CGMT is used in this work (i.e. limitation of the AMP approach)? ----------------------------------- Post-rebuttal: The authors addressed some of my concerns; I'm therefore keeping the positive score.


Review 2

Summary and Contributions: The paper discusses the performance of regularized empirical risk minimization for the high dimensional perceptron model. Specifically their rigorous results concern the model: for i=1,2,...,n Y_i=sign(<X_i,w^*>) for Gaussian prior on w^* and Gaussian X_i where they live on d dimensions. They assume n/d=\alpha (constant) and n,d grow to infinity. In [10] the Bayes optimal reconstruction error has been studied (verifying a stats physics prediction) and here they discuss about the performance of regularize ERM (potentially convex methods) to achieve it. Their first set of results is about the performance of any \ell_2 regularized convex loss and showing that their performance can be tracked using a fixed point equation. The result is based on Gordon's minimax theory and is shown then to be verifying also the replica (stats physics) prediction. Then the authors proceed by numerically checking the performance of the ridge loss, the logistic loss and the hinge loss for various regularization parameters. For ridge, they see the expected double descent behavior for small \lambda and they see always a gap between ridge and Bayes-optimal. Interestingly for hinge and logistic the difference from the Bayes-optimal error is smaller. Also there is some discussion on how Rademacher complexity fails to reveal the optimal rate of estimation for large \alpha. Finally there is some discussion about how the (poly-time) Approximate Message Passing (AMP) in this regime can give a *non-convex* loss and an appropriate regularization which gives an optimal regularized ERM for the problem. The authors mention in multiple places that there results should be generalizable to many more settings.

Strengths: The paper is definitely interesting in many ways. This is a regime where there is no computational-statistical gap (AMP reaches the Bayes-optimal performance) and therefore convex ERM could solve the problem to optimality - I think the authors should mention in the introduction. The authors provide very tight rigorous and non-trivial results on their asymptotic performance showing that also they match the replica predictions from statistical physics. The numerical simulation part is also very interesting and its add to the long line of work in ML which tries to understand the generalization performance of convex methods. It is also nice to see the double descent behavior appearing in the analysis of the square loss.

Weaknesses: The main weakness of the work is that in many ways it remains non-rigorous and many claims are observed through simulations. It would have been great to obtain provable analysis of the fixed point equations and what they imply in terms of success/failure for the various losses to achieve the Bayes-optimal performance. Also, I would *strongly* encourage the authors to be more open about which parts of the papers are rigorous and which are not: e.g. in line 145 the authors say that their results are valid for a wide range of regularizers, yet I am not sure which results they refer to and how do they know it as the statements are only from \ell_2. Please provide more details on this. Also the discussion with the Rademacher complexity seems puzzling to me. While I agree with it and I think it is interesting, I feel it has a lot of overlap with the work [49] from earlier this uear. I think the authors should mention in the Introduction when they describe their contributions, and recognize the similarities with this work early on.

Correctness: 1. I think some more details on the proof of Theorem 4.1. are needed. Why ERM given by (17) is identical with Bayes-optimal AMP? 2. By skimming through the proof of the important Theorem 2.2. (page 12 in the Appendix) I am puzzled by the following thing: Yes, Gordon's comparison inequality does say that \Phi and \tilde{\Phi} concentrate around the same optimal value, but it seems to me that the authors use that their *argmax* are also converging to each other, to replace completely the optimization problems and derive the desired conclusion. Am I reading this correctly? If yes, can more details be provided to convince me and the reader for the validity of this part? Gordon argmax also transfers?

Clarity: Yes, the paper is well-written.

Relation to Prior Work: I would suggest a better comparison with [49]. Also I would like to see a more extended literature review of rigorous analysis of ERM with regularization for the perceptron setting.

Reproducibility: Yes

Additional Feedback: In (1) the P_{out} should have been a conditional distribution, right?


Review 3

Summary and Contributions: In this article, the authors considered the high-dimensional classification problem in a teacher-student setting, and investigated the generalization performance of standard classifiers in the limit of a high (data) dimension d and the number of samples n, with a finite ratio alpha=n/d. Assuming the data x_mu to be i.i.d. standard Gaussian and the labels generated from a sign-teacher: y_mu = sign(x_mu * w^*) (that is a linearly separable model), for some Gaussian ground truth w^*, the classification generalization performances were given for popular convex losses (e.g., square, logistic and hinge loss) with positive l_2 regularization (Theorem 2.2). The proof is based on Gordon’s minimax theory and matches results from replica heuristics. The generalization performance of different classifiers was then compared to the Bayes-optimal performance. An optimal (non-convex) loss and the associated regularization were then proposed to achieve the Bayes-optimal performance (Theorem 4.1).

Strengths: The claims are, as far as I can check, correct, and match empirical evaluations on systems of moderate sizes. The author considered the classification setting and made an in-depth discussion on the impact of loss functions and regularization. The high-dimensional asymptotics was then compared to classical VC dimension and Rademacher statistical bounds, showing that the proposed analysis provides "sharper" results on this particular model.

Weaknesses: The contribution of this work is somehow incremental from a purely technical perspective since, as pointed out in the "Related works" paragraph, the same proof technique is applied to a new but somehow not very different model, i.e., sign-teacher with regularization. And the motivation for this sign-teacher is not very clear. Also, the Gaussian assumptions for both the data and the ground truth vector can be strong, it would be helpful to discuss how the conclusion might be extended beyond this setting, perhaps even empirically.

Correctness: The claims are, as far as I can check, correct.

Clarity: The paper is in general well written.

Relation to Prior Work: Previous efforts are clearly discussed.

Reproducibility: Yes

Additional Feedback: * line 155: "whose Moreau-Yosida regularization eq. (12) ..." bad reference here? =========================== After rebuttal: I've read the authors' response and my score remains the same.


Review 4

Summary and Contributions: This paper investigates the perceptron problem in a teacher-student scenario in which the output is generated from the teacher with i.i.d. Gaussian random inputs. Several learning rules, based on arbitrary convex losses with $\ell_2$ regularization and the Bayesian framework, are systematically compared in terms of the generalization error $\epsilon_g$. The contributions of the paper is three-fold. The first is to analytically rigorously derive the generalization error in generic convex losses. This resultantly justifies several known results derived by using the heuristic replica method from statistical physics. The second contribution is the detailed analysis of three convex losses: square, hinge, and logistic losses. This shows that the near Bayes-optimal error can be achieved by hinge and logistic losses with optimized regularization, including the learning rate scaling $\epsilon_g=O(\alpha^{-1})$ where $\alpha=n/d$ is the ratio between the dataset size $n$ and the dimensionality of the input $d$. It is also pointed out that bounds based on Rademacher complexity and Vapnik-Chervonenkis (VC) dimension lead to the scaling $O(\alpha^{-1/2})$ and thus are much looser. The third contribution is a systematic construction of a loss and an estimator achieving the Bayes-optimal error.

Strengths: I think the first contribution, the rigorous derivation of the generalization error, is appealing to people in NeurIPS community whose background is mathematics or physics. Especially for physicists, the replica method is a powerful utility but it is longly criticized due to the lack of rigorousness. The present method seems to be applied to a rather wide range of problems and thus can be a good alternative to the replica method. The second contribution is also considered to be important. The Bayes optimality is a nice theoretical baseline because it is considered that no better algorithm exist. The present result showing the near optimality of the hinge and logistic losses with optimized regularization provides a good support for using those losses. It is also interesting that the max-margin classifier also exhibits comparable performance with those cases. Comparing these results with the bounds based on the Rademacher complexity and the VC dimension in terms of the learning rate, the author(s) additionally discussed about the usefulness of the bounds based on the uniform convergence. Those bounds inevitably lead to looser scaling in the learning rate, implying the insufficiency of the uniform convergence framework for understanding generalization. I think this discussion would be useful for theoreticians interested in generalization issues in modern machine learning models.

Weaknesses: A weak point of the paper is the lack of direct consequence to practitioners. If any algorithm or theoretical knowledge relevant for practitioners are shown, it is really welcome. Another minor point is about the third contribution of the paper: the construction of the loss and estimator reaching the Bayes optimality. For me, it seems that the construction requires some information in the Bayesian framework such as $Z_{out^*}$, but if such information is available one seems to be able to directly work on the Bayesian framework. Is this incorrect? Is there any situation that the Bayesian cannot be used but the presented estimator can be? An explanation about this point is desired.

Correctness: I have checked several points in the theoretical computations and found they are correctly conducted. The numerical experiments well support the theoretical computations, and hence the results are considered to be correct.

Clarity: The paper is well organized and equipped with nice appendices well summarizing the very detailed computations. No big flaw is found. I only give the list of typos which I found: Line 72: technics -> techniques In references: Some references have typos: [27]: cdma -> CDMA [28]: bp -> BP, bayesian -> Bayesian [38]: &amp; -> & [50]: vapnic-chervonenkis -> Vapnic-Chervonenkis [52]: bayesian -> Bayesian In appendix: Line 123: must defined -> must define Line 199: $=$ is missed. Eq. (V.27): 1/\sqrt{d}Xw^* -> |1/\sqrt{d}Xw^*| (absolute value seems to be missed) Line 577: 1.10^(-2) -> 0.01 or 10^(-2) Line 615: need to used -> need to use # Comments after rebuttal The author rebuttal is satisfactory. I would like to keep my positive score.

Relation to Prior Work: The paper cites a broad range of references and its relevance is well explained in the light of them. The difference from previous contributions is clear.

Reproducibility: Yes

Additional Feedback:

[Author Response · NeurIPS 2020]

We thank the referees for their detailed work and their thoughtful comments that we gratefully use to improve our paper.

**Answer to referee 1:** We will add and adjust the two mentioned references, thanks for letting us know.

1. We do not know whether there is a general criteria that would distinguish when the decay is $1/\alpha$ or $1/\sqrt{\alpha}$. Providing
such a generic criteria is definitely a line of research we would like investigate in the future.

2. Our analysis in this paper focuses on i.i.d Gaussian data and L2 regularization, as this case was very extensively
studied in past works, in particular in statistical physics. As mentioned, the CGMT analysis used in this work can
be generalized to non-isotropic Gaussian and any convex loss and separable regularization. We will add a related
discussion in the final version to clarify.

3. We thank the referee for pointing out this very interesting work. First, note that the main focus of our work is on
the test error including the constants for any $\alpha$, not only on the rate at very large $\alpha$. Secondly, we focus here on the
generalization for a non Lipshitz function (namely the sign/misclassification loss) so that the suggested reference that
focus on smooth loss functions does not readily apply. We shall, however, include the discussion of these bounds in the
section where we compare to the Rademacher bounds, and investigate this interesting connection.

4. This is correct. The main reason why we use CGMT and not AMP is that we do not know how to prove that the state
evolution of the AMP corresponds to the solution of the ERM. Only after having the CGMT proof in hand, it follows
that the SE of AMP gives the same equations than the CGMT. We will add a comment to the paper.

**Answer to referee 2:** We shall indeed comment on the absence of the computational gap in the final version.

The set of fixed point equations is fully rigorous. The analytic solution of this set of equations is provided only in the
ridge case, and also for vanishing $\lambda \to 0$ that allows to obtain analytically the asymptotic generalization behaviour
of the max-margin estimator. Unfortunately, in the others situations, we did not find a closed form expression of the
fixed point, so that the generalization behaviour is evaluated numerically. We, however, note that these are fixed point
equations on scalar variables so their numerical resolution posed no problem. The non-trivial part of the rigorous
analysis is the reduction of the high-dimensional problem to the scalar fixed point equations.

We will clarify and distinguish the points of our work that are rigorously proven with this work, and the ones for which
the proof can be extended. As correctly pointed out, in order to not overload the paper and as most of the numerics has
been performed in this case, the set of fixed point equations (10) has been made rigorous only for L2 regularization
for binary classification, even though the entropy for regression is proven as well in (SM III.1.1). We stated that our
results are *valid* for a wide range of regularizers as the replica's prediction of the fixed point equations are provided for
generic convex and separable loss and regularizer. They are believed to hold true and the generic Gordon's mini-max
framework can be easily generalized to this case.

We only reproduced the results of [49] to bring to light interesting conclusions relating our work of the ERM estimations
and the discussion of [49]. We will clarify this in the final version.

1. As briefly explained in the proof below Theorem 4.1, the GAMP algorithm, recalled in (S.M VI.1), is valid for ERM
estimation with the corresponding updates $f_{\text{out}}^{\text{erm}}(l,r), f_{\text{w}}^{\text{erm}}(l,r)$ in (S.M VI.2). Thm 4.1 relies basically therefore on
the fact that the optimal loss $l^{\text{opt}}$ and regularizer $r^{\text{opt}}$ are designed in (S.M VI) such that at each time step the ERM
denoisers match the Bayes-optimal ones: $f_{\text{out}}^{\text{erm}}(l^{\text{opt}}, r^{\text{opt}}) = f_{\text{out}}^{\text{bayes}}$ and $f_{\text{w}}^{\text{erm}}(l^{\text{opt}}, r^{\text{opt}}) = f_{\text{w}}^{\text{bayes}}$. As these denoising
steps are the only difference between ERM-AMP and Bayes-AMP, and as according to [53] AMP algorithm with
Bayes-updates reaches Bayes-optimal performances, we obtain Theorem 4.1 and will clarify the discussion.

2. Indeed the Gordon analysis apply to $\Phi$ (SM III.9) and $\tilde{\Phi}$ (SM III.10). The free entropy of the problem is therefore
given by $\tilde{\Phi}$ that reads as an optimization problem over $\mu, \delta, \tau$ in (SM III.4). As $\lambda > 0$, it can be shown that this problem
has a unique solution $(\mu^\star, \delta^\star)$ so that $(\mu, \delta) \to (\mu^\star, \delta^\star)$. As suggested, we will add a clarification of this technical step.

**Answer to referee 3:** Our motivation to focus on the sign-teacher with Gaussian data and ground truth vector is that it
has been extensively studied in the past. Our analysis is valid more generically, and we will discuss this in the revised
manuscript more clearly, see answer to referee 2. We will correct the reference to the Moreau-Yosida regularization.

**Answer to referee 5:** We will correct the typos, thank you very much. Concerning consequences for practitioners, we
think that our result about approaching very closely the Bayes-optimal performance with simple regularized ERM is an
interesting message for practitioners. Of course, shoving this in more realistic settings would be desired.

As correctly pointed out, the construction of the optimal loss and regularizer requires the knowledge of the teacher
distribution related to $\mathcal{Z}_{\text{out}}^\star, \mathcal{Z}_{\text{w}}^\star$ in (17). Indeed, in the Bayes-optimal setting, we may directly use the Bayes-optimal
AMP algorithm to achieve optimal performances as proven in [53]. Nevertheless, it seems to us interesting to point out
that Bayes performances that require, in principle, to compute an intractable high-dimensional posterior sampling can
be obtained instead by the easier, more common and practical ERM estimation.

[Meta-Review · NeurIPS 2020]

All the reviewers agreed that the main results presented in this paper, the rigorous fixed-point equations for binary classification with generic loss and l2 regularizer, and more in-depth elucidation for three losses (ridge, hinge, and logistic), are sound and interesting. Although the problem setting may be thought as simple and limited, the findings in this paper are rigorous and non-trivial, which is the strength of this paper. In this regard, clarification on what statements are rigorous and what are not should be important. Some reviewers pointed out that it would be nicer if a general criterion telling if a particular loss would achieve the rate \propto \alpha^{-1} be provided. I think that this point would be worth mentioning in this paper. All the reviewers rated this paper favorably, which were kept after the author response. I would thus recommend acceptance of this paper. Minor points: Line 125: Corollary 2.3 should be referred to as corollary, not as theorem. acos (arccosine) should not be italicized.